# ONLINE MULTI-AGENT CONTROL
# WITH ADVERSARIAL DISTURBANCES

## ABSTRACT

Online multi-agent control problems, where many agents pursue competing and time-varying objectives, are widespread in domains such as autonomous robotics, economics, and energy systems. In these settings, robustness to adversarial disturbances is critical. In this paper, we study online control in multi-agent linear dynamical systems subject to such disturbances. In contrast to most prior work in multi-agent control, which typically assumes noiseless or stochastically perturbed dynamics, we consider an online setting where disturbances can be adversarial, and where each agent seeks to minimize its own sequence of convex losses. Under two feedback models, we analyze online gradient-based controllers with local policy updates. We prove per-agent regret bounds that are sublinear and near-optimal in the time horizon and that highlight different scalings with the number of agents. When agents' objectives are aligned, we further show that the multi-agent control problem induces a time-varying potential game for which we derive equilibrium tracking guarantees. Together, our results take a first step in bridging online control with online learning in games, establishing robust individual and collective performance guarantees in dynamic continuous-state environments.

## 1 INTRODUCTION

From energy grids and financial markets to autonomous driving fleets and online platforms, modern systems increasingly rely on many agents making independent decisions. These systems often operate in dynamic and uncertain environments that are vulnerable to *adversarial disturbances*. For instance, autonomous robots may suffer sensor failures or sudden disruptions from traffic and weather; financial markets may face adversarial price movements or shocks; and energy systems can be prone to demand spikes or strategic manipulation. In such settings, interacting agents pursue competing, time-varying objectives that may shift adversarially over time. Ensuring robustness in these environments requires online algorithms that adapt locally without relying on central coordination. Such algorithms are essential to ensure the safety, efficiency, and stability of large-scale multi-agent systems.

In this paper, we study online control in multi-agent linear dynamical systems subject to such adversarial disturbances. Specifically, we consider systems evolving as

$$x_{t+1} = Ax_t + B_1 u_t^1 + \cdots + B_N u_t^N + w_t , \tag{LDS}$$

where the global state $x_t$ depends simultaneously on the controls $(u_t^i)_{i \in \{1, \cdots, N\}}$ independently selected by $N$ agents, $A$ and $(B_i)_{i \in \{1, \cdots, N\}}$ are time-invariant transition matrices, and $w_t$ is an *adversarial* perturbation. At each time step $t$, every agent $i \in \{1, \cdots, N\}$ observes the state $x_t$, selects a control input $u_t^i$ according to a policy $\pi^i$ mapping states to controls, and subsequently incurs an individual time-varying cost $c_t^i(x_t, u_t^i)$.

In the *absence of adversarial disturbances*, multi-agent control with *quadratic* costs (linear quadratic games) is well-studied (Başar & Olsder, 1998; Mazumdar et al., 2020; Hambly et al., 2023). Applications span diverse domains including energy markets, formation control (Aghajani & Doustmohammadi, 2015; Han et al., 2019; Hosseinirad et al., 2023) and bioresource management (Mazalov et al., 2017), and we expand on these examples in Appendix B. However, most existing work on multi-agent control focuses on noiseless settings, or assumes Gaussian i.i.d. disturbances. Such assumptions are inadequate for modeling the *adversarial* disturbances that are increasingly present in modern multi-agent systems and which motivate our work.

In this adversarial and nonstationary setting, the natural performance measure is *individual regret*, which measures an agent's performance against a powerful class of counter-factual policies that have full knowledge of the future in hindsight. Formally, we define the individual regret of agent $i$ by

$$\text{Reg}_i^T(\mathcal{A}_i, \{u_t^{-i}\}, \Pi_i) = \sum_{t=0}^{T} c_t^i(x_t, u_t^i) - \min_{\pi^i \in \Pi_i} \sum_{t=0}^{T} c_t^i(x_t^{\pi^i}, u_t^{\pi^i}), \tag{1}$$

where $\mathcal{A}_i$ is the learning algorithm used by the $i$'th agent to select its control $u_t^i$, and $(x_t^{\pi^i}, u_t^{\pi^i})$ is the *counterfactual* state-control pair had policy $\pi^i$ been chosen by the agent starting from time $t = 0$, and where $\{u_t^{-i}\}$ are the fixed control inputs of other agents.

Achieving sublinear regret is the cornerstone of online learning, as it guarantees that an agent can adapt effectively to adversarial costs and disturbances. However, in a multi-agent system, this individual guarantee is only half the story. Because agents' costs are coupled through the shared state dynamics, the collective pursuit of low regret creates a complex decentralized dynamic. A fundamental insight from online game theory is that when all players achieve no-regret, their joint behavior can stabilize toward a collective equilibrium (Cesa-Bianchi & Lugosi, 2006; Nisan et al., 2007). Extending this powerful connection—from individual rationality to collective stability—to stateful, dynamical control systems is a major open challenge. This motivates our central question:

> *Can we design decentralized online control algorithms for (LDS) with adversarial disturbances that guarantee both uniform sublinear regret for each agent and stable equilibrium-tracking behavior for the system as a whole?*

This question introduces significant challenges not present in single-agent online control:

- **Decentralization:** Agents act locally without access to others' policies, so robust controllers cannot be computed centrally and broadcasted.

- **Scaling with number of agents:** The state coupling across all $N$ agents raises a key question: how do individual regret guarantees scale with the number of agents? Is sublinear regret even achievable?

- **Equilibrium behavior:** When agents have aligned objectives, it is unclear whether the dynamics driven by decentralized regret minimization can lead the system to track a global equilibrium.

### 1.1 OUR CONTRIBUTIONS

We provide an affirmative answer to our central question, establishing the first performance guarantees for online multi-agent control under adversarial disturbances. Our key results are:

**Individual Regret with Limited Information.** In an independent learning setting, where agents only observe the state, we prove a per-agent regret bound of $\widetilde{\mathcal{O}}(N^2\sqrt{T})$ using an online gradient-based controller (Algorithm 1). This result demonstrates robustness even with minimal feedback, while the quadratic dependence on $N$ quantifies a "price of decentralization" (Theorem 3.2). We also prove a matching lower bound of $\Omega(\sqrt{T})$, showing our time dependence is optimal (Theorem 3.3).

**Improved Regret with More Information.** In an aggregated control learning setting, where agents also observe the combined effect of others' actions, we improve the regret to $\widetilde{\mathcal{O}}(N\sqrt{T})$ (Theorem 3.4). With an additional Lipschitz assumption on the costs, we eliminate the dependence on $N$ entirely, achieving a near-optimal $\widetilde{\mathcal{O}}(\sqrt{T})$ regret (Theorem 3.5).

**Equilibrium Tracking.** In a common interest setting (a time-varying potential game), we prove that our no-regret dynamics successfully tracks the game's evolving Nash equilibria. The tracking error is bounded by the rate of change in the cost functions and disturbances, formally linking individual performance to collective stability (Theorem 4.1).

Together, these results bridge online non-stochastic control and learning in games, laying a foundation for robust and stable learning in dynamic, multi-agent environments and opening many avenues for future work and cross-fertilization between these two communities.

## 1.2 RELATED WORK

We give a brief discussion of related works and defer more details to Appendix A.

**Online non-stochastic control.** Our work builds on a recent and growing line of research focusing on the use of online learning techniques to address control problems with adversarially perturbed dynamical systems (Hardt et al., 2018; Abbasi-Yadkori & Szepesvári, 2011; Agarwal et al., 2019; Hazan et al., 2020; Foster & Simchowitz, 2020; Simchowitz et al., 2020; Simchowitz, 2020; Gradu et al., 2020; Ghai et al., 2023; Cai et al., 2024; Tsiamis et al., 2024; Golowich et al., 2024). On the one hand, when the dynamical system (LDS) involves only a single agent (i.e., $N = 1$), our setting collapses to (single-agent) *online non-stochastic control*. This problem has been thoroughly studied over the past years, see e.g. Hazan & Singh (2025) and the references therein. On the other hand, most of the works in this line of research are devoted to the control of linear dynamical systems influenced by a *single* controller. We discuss a few exceptions in the next section.

**Multi-agent control.** There is extensive research at the interface of control and game theory, see e.g. Marden & Shamma (2018); Chen & Ren (2019) for surveys. An important body of this literature has focused on linear-quadratic games (Başar & Olsder, 1998; Mazalov et al., 2017; Hosseinirad et al., 2023; Zhang et al., 2019; Bu et al., 2019; Zhang et al., 2021; Wu et al., 2023; uz Zaman et al., 2024; Mazumdar et al., 2020; Hambly et al., 2023). Some of these works typically consider the same (LDS) and assume quadratic costs for systems which are either deterministic ($w_t = 0$) or perturbed by a noise sequence $\{w_t\}$ which is i.i.d. Gaussian. Classical approaches to design robust controllers in optimal control rely either on using probabilistic models for disturbances or adopting a (worst-case) 'minimax' perspective (Başar & Bernhard, 2008).

A few recent works adopt an online learning approach for *distributed* control: Chang & Shahrampour (2023b;a) study a distributed online control problem over a multi-agent network of $m$ identical linear systems, where each agent seeks to compete with the best centralized control policy in hindsight. This is fundamentally different from our setting, where we consider *selfish strategic* agents influencing a *single* linear dynamical system, and where each agent attempts to minimize their own individual cost. Ghai et al. (2022) propose a reduction from any standard regret minimizing control method to a distributed algorithm implemented by several controllers, which is distinct from our setting of multiple, strategically competing agents. Recently, Golowich et al. (2024) proposed an online control approach for population dynamics where states are distributions in the simplex. We rather focus on the case of a finite and discrete large number of agents and discuss the influence of the total number of agents on individual regret.

**Online convex optimization and online learning in time-varying games.** Our regret analysis uses tools from online learning with memory (Anava et al., 2015; Kumar et al., 2023). Some of our results relate to the active research area of online learning in time-varying games (Cardoso et al., 2019; Duvocelle et al., 2023; Mertikopoulos & Staudigl, 2021; Fiez et al., 2021; Zhang et al., 2022a; Anagnostides et al., 2023; Feng et al., 2023; Yan et al., 2023b; Meng & Liu, 2024; Taha et al., 2024; Fujimoto et al., 2024; 2025; Crippa et al., 2025). However, these works do not address our multi-agent online control setting where time-varying costs depend on an underlying (LDS) with coupled state dynamics subject to adversarial disturbances.

## 2 PROBLEM FORMULATION: MULTI-AGENT ONLINE CONTROL

In this section, we formally introduce the multi-agent control setting over a finite time horizon $T$. The state process evolves as a linear dynamical system

$$x_{t+1} = Ax_t + \sum\nolimits_{i=1}^{N} B_i u_t^i + w_t, \quad t = 0, \cdots, T-1, \tag{LDS}$$

where $x_t \in \mathbb{R}^d$ is the state of the system initialized at a given (possibly random) state $x_0$, $u_t^i \in \mathbb{R}^{k_i}$ is the control of agent $i \in [N] := \{1, \cdots, N\}$, $w_t \in \mathbb{R}^d$ is an arbitrary system disturbance and $A \in \mathbb{R}^{d \times d}, B_i \in \mathbb{R}^{d \times k_i}$ are the system transition matrices defining the linear dynamical system.

### 2.1 ONLINE SETTING AND FEEDBACK MODELS

We consider the following online setting: at each time step $t$, all $N$ agents observe the state $x_t$ of the system. Then, each agent $i \in [N]$ selects a control input $u_t^i \in \mathbb{R}^{k_i}$ and incurs a loss $c_t^i(x_t, u_t^i)$,

where $c_t^i : \mathbb{R}^d \times \mathbb{R}^{k_i} \to \mathbb{R}$ is an adversarially chosen cost function. Finally, the system transitions to the next state according to the dynamics (LDS). The goal of each agent $i$ is to minimize their own cumulative cost over $T$ rounds.

We assume that each agent $i \in [N]$ knows the dynamics $(A, B_i)$. For each $i \in [N]$, the cost function $c_t^i$ is only locally accessible to agent $i$. The perturbation sequence $\{w_t\}$ is a priori unknown to agents. Moreover, we distinguish between the following two *information settings*:

**Information Setting 1** (**Independent Learning**). *At each time step $t$, agent $i \in [N]$ observes only the state $x_t$ (fully observable setting) and their own induced cost. In particular, agent $i$ has no access to the control inputs of other agents $j \neq i$.*

In the literature on multi-agent reinforcement learning, Information Setting 1 is commonly referred to as the *independent learning* setting (see, e.g., Daskalakis et al. (2020); Ozdaglar et al. (2021); Ding et al. (2022); Alatur et al. (2024)). We also consider a second setting where agents have access to more information about the other interacting agents in the system. This additional information revealed to every agent at each time step is naturally motivated by (LDS). Formally:

**Information Setting 2** (**Aggregated Control Learning**). *At each time step $t$, agent $i$ observes the state $x_t$ and their own induced cost, as well as the aggregated feedback $\sum_{j \neq i} B_j u_t^j$ that encodes information about other agents' control inputs. Each agent $i$ knows the total number of agents $N$.*

This stronger information setting is analogous to the standard setting of full-information feedback (hindsight observability) in the literature of online learning in games. This setting allows a player to evaluate their loss for any counterfactual action. Similarly, in our setting, observing the state and aggregated control lets each agent reconstruct the disturbance and thus compute their counterfactual loss for any alternative control they could have individually chosen, given others' actions.

## 2.2 REGRET FRAMEWORK FOR MULTI-AGENT ONLINE CONTROL

In this section, we give a more formal definition of our performance metric for multi-agent online control, inspired from both single-agent online control and online learning in games.

**Individual policy regret.** Since the system dynamics depend on unknown costs and possibly adversarial perturbations, determining an optimal controller a priori is not possible in general. Therefore, in contrast to classical and robust optimal control, we consider *regret* as a performance measure, following the recent line of works on (single-agent) online non-stochastic control (Hazan et al., 2020). For each agent $i \in [N]$, consider a benchmark policy class $\Pi_i \subset \{\pi^i : \mathcal{X} \to \mathcal{U}^i\}$. Each agent $i$ runs their online control algorithm $\mathcal{A}_i$ to determine their control input $u_t^i = \mathcal{A}_i(x_t)$, where $x_t$ is the state of the system described by (LDS). For any $T \geq H \geq 1$, we define the regret of agent $i$ w.r.t. policy class $\Pi_i$ when agent $i$ runs algorithm $\mathcal{A}_i$ and other agents use controls $\{u_t^{-i}\}$ as follows:

$$\text{Reg}_i^{H:T}(\mathcal{A}_i, \{u_t^{-i}\}, \Pi_i) = \max_{w_{1:T} : \|w_t\| \leq W} \left( \sum_{t=H}^T c_t^i(x_t, u_t^i) - \min_{\pi^i \in \Pi_i} \sum_{t=H}^T c_t^i(x_t^{\pi^i}, u_t^{\pi^i}) \right), \quad (2)$$

where $W > 0$ and $x_t^{\pi^i}, u_t^{\pi^i}$ are the *counterfactual state and controls* under the policy $\pi^i$ for agent $i$:

$$u_t^{\pi^i} = \pi^i(x_t^{\pi^i}), \quad x_{t+1}^{\pi^i} = A x_t^{\pi^i} + B_i u_t^{\pi^i} + \sum_{j \neq i} B_j u_t^j + w_t. \quad (3)$$

The counterfactual state sequence corresponds to the state sequence that would be observed if agent $i$ were to unilaterally deviate to using policy $\pi^i$, instead of their online control algorithm $\mathcal{A}_i$ (and where all other agents stick to their online control input sequence). Note that when $N = 1$, expression (2) recovers the regret definition for single-agent online control.

In this work, we consider two natural policy comparator classes, which we introduce as follows:

**Comparator policy class 1: Strongly stable linear controllers ($\Pi_i^{\text{lin}}$).** For agent $i$, a *linear controller* is defined by a matrix $K_i \in \mathbb{R}^{k_i \times d}$ s.t. $u_t^i = -K_i x_t$. We say that a linear policy $K_i$ is *stable* if $\rho(A - BK_i) < 1$ (where $\rho(\cdot)$ denotes the spectral radius), in which case the closed-loop state-feedback linear dynamical system is globally asymptotically stable. *Strong stability* of a controller is a quantitative version of stability which allows for deriving non-asymptotic guarantees.

**Definition 2.1** (Strong stability, e.g. Cohen et al. (2018)). *A linear policy $K$ is $(\kappa, \gamma)$-strongly stable (for $\kappa > 0$ and $0 < \gamma < 1$) for a linear dynamical system specified by $(A, B)$ if $\|K\| \leq \kappa$ and if there exists matrices $L, Q$ s.t. $A - BK = QLQ^{-1}$ with $\|L\| \leq 1 - \gamma$, and $\|Q\| \cdot \|Q^{-1}\| \leq \kappa$.*

Note that strong stability implies stability, and any stable policy is strongly-stable for some $(\kappa, \gamma)$. A natural policy comparator class is that of *strongly stable linear controllers* $\Pi_i^{\text{lin}}$, parameterized by:

$$\mathcal{K}_i := \left\{ K_i \in \mathbb{R}^{k_i \times d} : K_i \text{ is } (\kappa_i, \gamma_i)\text{-strongly stable for some } \kappa_i > 0, \gamma_i \in (0, 1) \right\} . \quad (4)$$

**Comparator policy class 2: Disturbance Action Controller (DAC) policies ($\Pi_i^{\text{DAC}}$).** The state sequence induced by a linear controller is not a linear function of its parameters. As a consequence, the induced cost is non-convex in the control parameters in general, even if the cost function is convex in both the state and the control input (see, e.g., Fazel et al. (2018)). Following prior work in single-agent control, we consider *Disturbance Action Controller* (DAC) policies. The system state induced by such policies is *linear* in the policy parameters and one can invoke tools from online convex optimization when the cost functions are convex in the state and control input. For a sequence of perturbations $\{w_t\}$, a DAC policy $\pi^i(M_i, K_i)$ for agent $i \in [N]$ is then specified by learnable matrix parameters $M_i = [M_i^{[0]}, M_i^{[1]}, \cdots, M_i^{[H-1]}]$ for a memory length $H \geq 1$, with a fixed given stabilizing controller $K_i$. The policy $\pi^i(M_i, K_i)$ selects action $u_t^i$ at a state $x_t$ as:

$$u_t^i = -K_i x_t + \sum\nolimits_{p=1}^{H} M_i^{[p-1]} w_{t-p} . \quad \text{(DAC-}i\text{)}$$

Note that for $p < 0$ we let $w_p = 0$, and moreover, the perturbations $w_t$ are not observed by the learners but rather computed online using the structure of (LDS) and the state observations (we discuss these points later). The policy can thus be implemented in an online fashion by agent $i$, and we henceforth use the notation $M_{i,t} = [M_{i,t}^{[p]}]_{0 \leq p \leq H-1}$ to reference the parameters of player $i$ at time $t$. For a fixed $H$ and stabilizing controller $K_i$, let $\mathcal{M}_i = \left\{ M_i = \{M_i^{[0]}, \cdots, M_i^{[H-1]}\} : \right.$ $\left. \|M_i^{[p-1]}\| \leq 2\kappa^2 (1-\gamma)^p, p = 1, \cdots, H \right\}$ denote the set of all DAC policy parameters for agent $i$, where $(\kappa, \gamma)$ are strong stability parameters of $K_i$ (with $(\kappa, \gamma) = (\kappa_i, \gamma_i)$ under Assumption 3 in information setting 1 and $(\kappa, \gamma) = (\bar{\kappa}, \bar{\gamma})$ under Assumption 4 in information setting 2).

## 3 INDIVIDUAL REGRET GUARANTEES

In this section, we present our results on individual regret guarantees. We analyze an *Online Gradient Perturbation Controller* algorithm, where each agent independently updates its DAC policy parameters via online gradient descent (Algorithm 1). In the single-agent setting ($N = 1$), this algorithm was introduced and analyzed by Agarwal et al. (2019). In our decentralized multi-agent setting, the coupling of state dynamics across all agents induces new obstacles to implementing and analyzing this gradient-based approach. We elaborate first on the computational challenge:

**Memory.** The cost $c_t^i(x_t, u_t^i)$ incurred by agent $i \in [N]$ at time step $t$ depends on the state $x_t$ of the system, which itself depends on all past states and control inputs from $t = 0$. However, to run the online gradient descent subroutine of Algorithm 1, agent $i$ must be able to evaluate its cost function $c_i^t$ on counterfactual state-action pairs. Unlike the single-agent case, counterfactual evaluation here depends not only on the agent's own past controls but also on the entire joint sequence of other agents' controls. This dependence breaks the straightforward counterfactual construction of the single-agent setting and requires a new memory-based approximation tailored to the multi-agent coupling.

Focusing on agent $i$'s perspective, suppose all other players use a given sequence of control inputs $\{u_t^{-i}\}$. Let $x_t^{K_i}(M_i, u_t^{-i})$ denote the (counterfactual) state reached by the system if agent $i$ were to execute a DAC-$i$ policy $\pi_i(M_i, K_i)$ with parameters $M_i$ and fixed matrix $K_i$ for all time steps from time zero. Evaluating the induced cost would require computations that scale linearly with time. Thus, for computational efficiency we endow agent $i$ with a memory of length $H$ that scales polylogarithmically with the time horizon $T$ and that will be carefully tuned to obtain our results. We denote by $y_t^{K_i}(M_i)$ the ideal state of the system that would have been reached if agent $i$ played the DAC-$i$ policy $\pi^i(M_i, K_i)$ from time $t-H$ to $t$, assuming that the state at time $t-H$ is zero, and while other agents use the control sequence $\{u_{t-H:t}^{-i}\}$. The idealized action to be executed at time $t$ at the state $y_t^{K_i}(M_i)$ observed at time $t$ is denoted by $v_t^{i, K_i}(M_i) = -K_i y_t^{K_i}(M_i) + \sum_{p=1}^{H} M_i^{[p-1]} w_{t-p}$. Let $\ell_t^i(M_i) = c_t^i(y_t^{K_i}(M_i), v_t^{i, K_i}(M_i))$ be agent $i$'s idealized cost function evaluated at the idealized state and action pair. The latter constitutes the counterfactual convex loss sequence for agent $i$ that can be evaluated efficiently, as in Algorithm 1.

**Algorithm variants.** Depending on the information setting (Settings 1 and 2), we define two variants of Algorithm 1, each described from the perspective of a fixed agent $i \in [N]$. These variants capture different levels of feedback and are essential for obtaining our regret guarantees.

---

**Algorithm 1** Online Gradient Perturbation Controller Algorithm (for agent $i \in [N]$)

---

1: Input: memory $H$, step size $\eta$, initialization $M_{i,1}^{[0:H-1]}$.
2: Compute a stabilizing linear controller $K_i$ knowing $(A, B_i)$.
3: **for** $t = 1 \ldots T$ **do**
4:     Observe state $x_t$.
5:

| /Update under Info. Setting 1: | /Update under Info. Setting 2: |
|---|---|
| Compute $\widetilde{w}_{t-1} = x_t - Ax_{t-1} - B_i u_{t-1}^i$ . | Observe $\sum_{j \neq i} B_j u_{t-1}^j$ . |
| Set $u_t^i = -K_i x_t + \sum_{p=1}^H M_{i,t}^{[p]} \widetilde{w}_{t-p}$ . | Compute $w_{t-1} = x_t - Ax_{t-1} - \sum_{k=1}^N B_k u_{t-1}^k$ . |
| | Set $u_t^i = -K_i x_t + \sum_{p=1}^H M_{i,t}^{[p]} w_{t-p}$ . |

6:     Record instantaneous cost $c_t^i(x_t, u_t^i)$
7:     Construct loss $\ell_t^i(M_i) = c_t^i(y_t^{K_i}(M_i), v_t^{i,K_i}(M_i))$.
8:     Update $M_{i,t+1} = \Pi_{\mathcal{M}_i}\left[ M_{i,t} - \eta \nabla \ell_t^i(M_{i,t}) \right]$.
9: **end for**

---

**Standing Assumptions.** Finally, before introducing our regret guarantees, we present our standing assumptions, all standard in the recent literature on online non-stochastic control:

**Assumption 1** (Cost functions). *The following assumptions hold for every $i \in [N]$:*

   *(i) The cost function $c_t^i : \mathcal{X} \times \mathcal{U}_i \to \mathbb{R}$ is convex w.r.t. both its arguments.*

   *(ii) There exists $\beta, G > 0$ s.t. for any $D > 0$ and every $(x, u^i) \in \mathcal{X} \times \mathcal{U}_i$ s.t. $\|x\| \leq D, \|u^i\| \leq D$, we have $|c_t^i(x, u)| \leq \beta D^2$ and $\|\nabla_x c_t^i(x, u^i)\|, \|\nabla_u c_t^i(x, u^i)\| \leq GD$ .*

**Lemma 3.1.** *Under Assumption 1, the loss function $\ell_t^i$ is convex w.r.t. $M_i$ for all $i \in [N]$.*

**Assumption 2** (Bounded disturbances). *There exists $W > 0$ s.t. for all $t \geq 0$, $\|w_t\| \leq W$ .*

### 3.1 INFORMATION SETTING 1: INDEPENDENT LEARNING

Under Information Setting 1, agents do not have access to other agents' control inputs. However, from the viewpoint of a given agent $i$, we observe that (LDS) can be re-expressed as follows:

$$x_{t+1} = Ax_t + B_i u_t^i + \widetilde{w}_t, \quad \widetilde{w}_t = \sum_{j \neq i} B_j u_t^j + w_t . \qquad (5)$$

In this view, in Algorithm 1, we naturally propose that agent $i$ executes a (DAC-$i$) policy with disturbance sequence $\widetilde{w}_t$. Given expression (5), note that $\widetilde{w}_t$ (unlike $w_t$) can be calculated by agent $i$ at each time step since $\widetilde{w}_t = x_{t+1} - Ax_t - B_i u_t^i$, and this computation only involves information observed under the information setting (state observations and the agent's own control input). Under this strategy, agent $i$ thus faces a linear dynamical system (5) controlled by its own, single control inputs, and for this we make a standard strong stability assumption adapted to the multi-agent setting:

**Assumption 3** (Agent-wise strong stability). *Each learner $i \in [N]$ knows a linear controller $K_i$ that is $(\kappa_i, \gamma_i)$-strongly stable for the linear dynamical system specified by $(A, B_i)$.*

Under this assumption, we present our first individual regret guarantees.

**Theorem 3.2** (**Individual Regret in Setting 1, Independent Learning**). *Let Assumptions 1, 2 and 3 hold. Suppose there exists $U > 0$ s.t. for all $t \geq 0, j \in [N], \|u_t^j\| \leq U$. If agent $i \in [N]$ runs Algorithm 1 under Setting 1 with (DAC-$i$) policy on perturbation sequence $\{\widetilde{w}_t\}$ and step size $\eta = \Theta(1/(G\widetilde{W}\sqrt{T}))$, where $\widetilde{W} = W + (N-1)U(\max_j \|B_j\|)$, and with $H \geq \log(\kappa_i T)/\gamma_i$, then for any $T \geq H + 1$, we have $\mathrm{Reg}_i^{H+1:T}(\mathcal{A}_i, \{u_t^{-i}\}, \Pi_i^{lin}) = \widetilde{\mathcal{O}}(U^2 N^2 \sqrt{T})$[1].*

---

[1]For readability, here and throughout, we use $\widetilde{\mathcal{O}}$ to hide polynomial factors in natural problem parameters and (poly)logarithmic factors in $T$ and $N$. We state the exact dependencies in the proofs of each result.

The proof of Theorem 3.2 consists of applying the single-agent regret guarantee for gradient perturbation controllers (Agarwal et al., 2019, Theorem 5.1) for each agent $i$ on the new perturbation sequence $\{\widetilde{w}_t\}$ in (5), and we give the full details in Section E. While the theorem highlights the robustness of gradient perturbation controllers to adversarial disturbances in this setting, the regret bound grows quadratically with both the number of agents $N$ and the magnitude $U$ of the control inputs. In the multi-agent setting, this scaling reflects the price of decentralization and indicates how performance can degrade when the number of agents in the system grows large.

**Regret lower bound.** In light of the regret guarantee of Theorem 3.2, it is also natural to ask whether the $\sqrt{T}$ dependence on the time horizon can be improved. In general, we prove that the answer is no. In particular, for any agent $i \in [N]$, we establish the following $\Omega(\sqrt{T})$ lower bound against the class of linear controllers that holds independently of the agent's algorithm $\mathcal{A}_i$:

**Theorem 3.3.** *For any agent $i \in [N]$, there exists an instance of* (LDS) *and cost functions $\{c_t^i\}$ such that, for any algorithm $\mathcal{A}_i$ and sequence $\{u_t^{-i}\}$, and any $T \geq 1$:* $\mathrm{Reg}_i^T(\mathcal{A}_i, \{u_t^{-i}\}, \Pi_i^{\mathrm{lin}}) = \Omega(\sqrt{T})$.

To prove the theorem, we construct a scalar-valued instance of (LDS) and a hard sequence of cost functions $\{c_t^i\}$ inspired by lower bounds for (single-agent) online linear optimization (see, e.g., Arora et al. (2012)). Importantly, we note that such $\Omega(\sqrt{T})$ lower bounds from online learning cannot be directly applied, as the incurred cost of the agent and the incurred cost of a comparator policy depend on different state evolution sequences. However, Theorem 3.3 implies that, due to the (possibly adversarially) time-varying nature of the cost sequence $\{c_t^i\}$, the individual regret of an agent in the present setting must in general have the same dependence on $T$ as in adversarial online learning. The proof is developed in Section H.

## 3.2 Information setting 2: Aggregated control learning

While the lower bound of Theorem 3.3 implies that a $\sqrt{T}$ dependence can not, in general, be improved upon, the regret in Theorem 3.2 under Setting 1 scales *quadratically* with the number of agents. In this section, we consider Information Setting 2 and analyze the case in which all agents run DAC policies. Under a global assumption on the resulting dynamical system, we prove that we can guarantee an individual regret bound with an improved dependence on the total number of agents $N$. We first make our global assumption which shall replace Assumption 3 in this section.

**Assumption 4** (Global strong stability). *Each learner $i \in [N]$ knows a linear controller $K_i$ such that $(K_1, \cdots, K_N)^T$ is $(\bar{\kappa}, \bar{\gamma})$-strongly stable for the LDS $(A, [B_1, \cdots, B_N])$.*

Assumption 4 is a natural global assumption which is relevant when each agent $i$ executes a (DAC-$i$) policy (with matrix $K_i$). Indeed, observe that the system state evolution of (LDS) in the absence of disturbances, and when all players use their linear controllers, can be written as $x_{t+1} = Ax_t - [B_1, \cdots, B_N](K_1, \cdots, K_N)^T x_t$. Each agent $i$ has access to the global parameters $\bar{\kappa}, \bar{\gamma}$ which can be centrally precomputed before each agent runs their Algorithm 1 independently. Recall that the matrices $K_i$ are not learning parameters and need to be precomputed even in the independent learning setting. Only the matrix parameters $M_i$ of (DAC-$i$) policies are learned by the agents.

Under Setting 2, all agents can compute the original disturbance $w_t$ at each time step (instead of $(\tilde{w}_t)$ as in Theorem 3.2). However, note that at every timestep $t$, each agent updates their own policy parameters independently and locally in an uncoupled fashion, without access to other agent's policy parameters at that round. After acting, each agent first incurs the loss according to their individual cost function, and *then* observes the aggregated feedback. This feedback is used to inform their next policy parameter update at round $t + 1$.

Our next result shows that when agent $i$ runs Algorithm 1 with (a) a conservative stepsize scaled by $N$ and (b) a larger memory which depends logarithmically on $N$ (compared to Theorem 3.2), they guarantee a regret w.r.t. the DAC policy class scaling only *linearly* in $N$ (not quadratically) . This result is also robust to other agents' strategies (as they can execute arbitrary (DAC-$i$) policies).

**Theorem 3.4** (**Individual Regret in Setting** 2). *Let Assumptions 1, 2, 4 hold. Then if agent $i \in [N]$ runs Algorithm 1 under Setting 2 with a (DAC-$i$) policy on perturbation sequence $\{w_t\}$, step size $\eta = \Theta(1/N\sqrt{T})$, and with $H \geq \log(2\bar{\kappa}N^2\sqrt{T})/\bar{\gamma}$, and when **all** other agents use a (DAC-$i$) policy with perturbation sequence $(w_t)$, then for any $T \geq H + 1$:* $\mathrm{Reg}_i^{H+1:T}(\mathcal{A}_i, \{u_t^{-i}\}, \Pi_i^{DAC}) = \widetilde{\mathcal{O}}(N\sqrt{T})$.

**Proof Overview.**    To prove the theorem, our analysis relies on a regret decomposition with two main terms: a counterfactual state-control error due to the use of a loss with limited memory $H$, and a regret term induced by the online gradient descent component of Algorithm 1. In summary, the main technical challenges we overcome are two-fold: first, states may grow unbounded with an undesirable scaling in $N$, and thus we control their magnitude by studying the state evolution when all agents use DAC policies (using Assumption 4), and while tracking the dependence on $N$. Second, we control both terms of the regret decomposition by carefully selecting the memory $H$, and with an adequate step size $\eta$ (optimal in terms of $N$). We present the full proof details in Appendix F.

We also remark that the linear dependence on $N$ in the regret bound is enabled by global stability (Assumption 4). By contrast, if only individual stability (Assumption 3) is assumed, even when agents can access aggregated control information, the dependence on $N$ deteriorates (see Appendix C.3 for a discussion). Moreover, under a stronger assumption on the cost functions (compared to Assumption 1-(ii)), we further prove a sublinear regret for agent $i$ that scales only *polylogarithmically* in $N$:

**Assumption 5** (Lipschitz costs). *There exists $\bar{L} > 0$ s.t. for any agent $i \in [N]$ and for all state-control pairs $(x, u^i), (\tilde{x}, \tilde{u}^i) \in \mathcal{X} \times \mathcal{U}_i, |c_t^i(x, u^i) - c_t^i(\tilde{x}, \tilde{u}^i)| \leq \bar{L}(\|x - \tilde{x}\| + \|u^i - \tilde{u}^i\|).$*

Note here that the Lipschitz constant does not scale with the state and control input magnitude. Under this assumption, we further obtain the following improved regret guarantee (proven in Appendix F):

**Theorem 3.5.** *Under the setting of Theorem 3.4, replace gradient boundedness in Assumption 1 -(ii) by Assumption 5. Set instead $\eta = \Theta(1/\sqrt{T})$ and $H \geq \log(2\bar{\kappa}N\sqrt{T})/\bar{\gamma}$. Then for any $T \geq H + 1$:*
$$\mathrm{Reg}_i^{H+1:T}(\mathcal{A}_i, \{u_t^{-i}\}, \Pi_i^{DAC}) = \widetilde{\mathcal{O}}(\sqrt{T}).$$

Note that using Assumption 5 in Theorem 3.2 does not result in the same improved dependence on $N$ as the regret will still scale with the magnitude of the modified disturbance $\widetilde{w}_t$, which is of order $N$. Finally, in Appendix H.2 we also show that the regret lower bound of Theorem 3.3 can be extended to hold against the DAC comparator class when the linear controller component is chosen adversarially. We state and prove this result formally in Theorem H.3 in Section H.2.

# 4    Equilibrium Tracking in the Common Interest Setting

In the previous section, we developed individual regret guarantees when other agents execute linear or DAC control policies with possibly misaligned or adversarially-chosen cost functions. In this section, we focus on the *common interest setting*, where the objectives of the agents are aligned and all cost functions are identical (i.e., $c_t^i = c_t^j := c_t$ for any $i, j \in [N]$ for every $t$). Our goal is to establish global equilibrium guarantees when *all* agents simultaneously and independently run Algorithm 1.

Since the cost functions are time-varying (not only via the strategies of the different players), our multi-agent control problem can be seen as a *time-varying game*. There have been considerable efforts endeavoring to extend the scope of traditional game-theoretic results to the time-varying setting and this is an active research area (see the related work in Section 1). In particular, our results in this section are inspired from recent developments for time-varying, normal-form, finite potential games in Anagnostides et al. (2023). In such games, agents participate in a potential game at each time step. We observe that the common interest multi-agent control problem can be seen as a stateful, time-varying potential *continuous* convex game where costs are functions of states driven by an underlying (LDS) influenced by multiple controllers. At each time step, the utility of each player is given by their cost function, and their strategy is defined by their DAC policy parameters.

Since our setting involves adversarial, time-varying costs depending on state dynamics influenced by adversarial (time-varying) disturbances, convergence to (static) Nash equilibria is irrelevant in general. Nevertheless, we establish equilibrium gap *tracking* guarantees for our dynamic setting. To state our result, we introduce notations for time-varying best responses and equilibrium gaps:

$$\mathrm{BR}_i^{(t)}(M_{-i,t}) := \max_{M_i \in \mathcal{M}_i} \ell_t(M_t) - \ell_t(M_i, M_{-i,t}); \quad \mathrm{EQGAP}^{(t)}(M_t) := \max_{i \in [N]} \mathrm{BR}_i^{(t)}(M_{-i,t}), \quad (6)$$

where, as previously defined, $\ell_t^i(M_t) = \ell_t^i(M_{t-1-H:t}) = c_t^i(y_t^K(M_{t-1-H:t-1}), v_t^{i,K}(M_{t-1-H:t}))$ and $K = (K_1, \cdots, K_N)$. Note that the equilibrium gap explicitly depends on time (as indicated by its superscript $^{(t)}$) due to the time dependence of the cost function and the disturbance sequence. We

now make regularity assumptions on the common cost function $c_t$ which are standard in the analysis of gradient methods in both optimization and learning in games.

**Assumption 6** (Uniform cost lower bound). *The cost function $c_t : \mathcal{X} \times \mathcal{U} \to \mathbb{R}$ is uniformly lower-bounded, i.e. there exists $c_{inf} > 0$ s.t. for all $x \in \mathcal{X}, u \in \mathcal{U}, t \geq 1, c_t(x, u) \geq c_{inf} > -\infty$.*

**Assumption 7** (Smoothness). *There exists $\zeta > 0$ s.t. the cost function $c_t : \mathcal{X} \times \mathcal{U} \to \mathbb{R}$ satisfies for every $t \geq 0$ and any $x, x' \in \mathcal{X}, u, u' \in \mathcal{U}$,*

$$\|\nabla_x c_t(x, u) - \nabla_x c_t(x', u')\| + \|\nabla_u c_t(x, u) - \nabla_u c_t(x', u')\| \leq \zeta(\|x - x'\| + \|u - u'\|). \quad (7)$$

Under these assumptions, when all agents run Algorithm 1, we bound the average equilibrium gap by the variation of both cost functions and disturbances.

**Theorem 4.1.** *Let Assumptions 1, 2, 4, 6 and 7 hold. Then if each agent $i \in [N]$ runs Algorithm 1 for $T$ steps with constant stepsize $\eta = 1/L$ (where $L$ is the smoothness constant in Lemma I.5), then*

$$\frac{1}{T} \sum_{t=1}^{T} \left( \text{EQGAP}^{(t)}(M_t) \right)^2 = \mathcal{O} \left( \frac{\ell_1(M_1) - c_{\inf}}{T} + \frac{1}{T} \sum_{t=1}^{T} \Delta_{c_t} + \frac{1}{T} \sum_{t=1}^{T} \|w_{t+1} - w_t\| \right), \quad (8)$$

*where $\Delta_{c_t} := \max_{\|x\|, \|u\| \leq D} \{c_{t+1}(x, u) - c_t(x, u)\}$ for every $t$, the $\mathcal{O}(\cdot)$ notation only hides polynomial dependence in the problem parameters $N, H, W, \bar{\kappa}, \bar{\gamma}^{-1}, \max_i \|B_i\|$ and $D$ depends polynomially on the same constants. All the constants are made explicit in the appendix.*

In a static setting, with time-independent costs in the absence of disturbances ($w_t = 0$ or constant), Theorem 4.1 translates into the existence of a time step $t \leq T$ s.t. the joint DAC policy $M_t$ is an $\epsilon$-approximate Nash equilibrium of the game induced by the loss functions $\ell^i, i \in [N]$ after $T$ iterations (typically $T = \mathcal{O}(1/\epsilon^2)$ for a $\mathcal{O}(1/T)$ rate). In this static case, the cumulative equilibrium gap is bounded by the initial cost optimality gap. If both the cost variability term and the cumulative variation in perturbations $\sum_{t=1}^{T} \|w_{t+1} - w_t\|$ are uniformly bounded by a constant, then the theorem results in a $\mathcal{O}(1/T)$ rate in terms of the average equilibrium gap squared. For example, this is clearly the case when the noise sequence $w_t$ converges towards a (not-necessarily vanishing) constant. If we only have $\sum_{t=1}^{T} \|w_{t+1} - w_t\| = o(T)$, then we still obtain a vanishing average equilibrium gap.

**Proof Overview.** To prove the theorem, we extend the approach of Anagnostides et al. (2023) (who considered time-varying *(finite) normal-form* potential games) to (a) cover *(continuous) convex games* and (b) account for state dynamics and adversarial disturbances in addition to the time-varying costs in our multi-agent control setting. We give an overview and details of the full proof in Appendix I.

## 5 CONCLUSION AND FUTURE WORK

This work initiates and makes progress on online multi-agent control in strategic environments subject to adversarial disturbances, taking a first step toward bridging online control with learning in games. In particular, we proved the first individual regret and global equilibrium tracking guarantees in the online multi-agent control setting with adversarial disturbances and time-varying costs.

Our results also open several directions for future research: on the technical side, it is interesting to investigate whether tighter regret bounds can be obtained with respect to the number of agents or under structural assumptions such as time-invariant costs. On the modeling side, important challenges include extending our analysis to settings with unknown or time-varying dynamics (Hazan et al., 2020; Minasyan et al., 2021; Gradu et al., 2023) and to feedback-limited regimes (Yan et al., 2023a), where learners can only access partially observed states and partially informed bandit costs. A broader challenge is to design decentralized multi-agent controllers that remain robust under adversarial disturbances beyond *linear* state dynamics. In conclusion, we view our work as a first step toward further advances at the interface of online control and learning in games in dynamical strategic environments.

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

CONTENTS

## A  EXTENDED RELATED WORK DISCUSSION

**Online non-stochastic control.** Our work builds on a recent and growing line of research focusing on the use of online learning techniques to address control problems with adversarially perturbed dynamical systems (Hardt et al., 2018; Abbasi-Yadkori & Szepesvári, 2011; Agarwal et al., 2019; Hazan et al., 2020; Foster & Simchowitz, 2020; Simchowitz et al., 2020; Simchowitz, 2020; Gradu et al., 2020; Ghai et al., 2023; Martin et al., 2024). We refer the reader to a nice introduction to the topic in the recent monograph of Hazan & Singh (2025) and the references therein for a survey. Recent follow-up works include studies on dynamic regret for online tracking (Tsiamis et al., 2024), performative control (Cai et al., 2024), online control in population dynamics (Golowich et al., 2024; Lu et al., 2025), simultaneous system identification and MPC with regret guarantees (Zhou & Tzoumas, 2024), online RL (Muehlebach et al., 2025; Ghai et al., 2023), partial feedback settings (Yan et al., 2023a) and bandit settings Sun & Lu (2024) to name a few. Most of the works in this line of research are devoted to the control of linear dynamical systems influenced by a *single* controller. We discuss a few exceptions in the next section.

**Multi-agent control.** The interface between game theory and control has given rise to a large body of work over the last decades to study settings involving multiple interacting controllers, see e.g. Marden & Shamma (2015; 2018); Chen & Ren (2019) for relevant surveys. Within the game-theoretic control literature, linear-quadratic (LQ) games is one of the canonical benchmark problems which has been studied in a variety of settings including LQ differential games (Başar & Olsder, 1998, Chap. 6), LQ potential games (Mazalov et al., 2017; Hosseinirad et al., 2023), zero-sum LQ games (Zhang et al., 2019; Bu et al., 2019; Zhang et al., 2021; Wu et al., 2023), static two-player quadratic games (Calderone & Oishi, 2024) and general-sum LQ games (uz Zaman et al., 2024; Mazumdar et al., 2020; Hambly et al., 2023; Chiu et al., 2024; Guan et al., 2024). Some of these works typically consider the same (LDS) and assume quadratic costs for systems which are either deterministic ($w_t = 0$) or perturbed by a noise sequence $\{w_t\}$ which is i.i.d. Gaussian. In particular they do not adopt the online learning perspective and do not address the case of arbitrary disturbances. Classical approaches to design robust controllers in the optimal control literature rely either on using statistical and probabilistic models for disturbances such as for linear quadratic Gaussian design, or adopting

a (worst-case) game theoretic perspective via designing 'minimax' controllers like in $H_\infty$ control (Başar & Bernhard, 2008). Only few recent works adopt an online learning perspective for *distributed* control (Ghai et al., 2022; Chang & Shahrampour, 2023b;a; Martinelli et al., 2024). Chang & Shahrampour (2023b;a) studied a distributed online control problem over a multi-agent network of $m$ identical linear time-invariant systems in the presence of adversarial perturbations. Each agent seeks to generate a control sequence that can compete with the best centralized control policy in hindsight. In contrast, we address a multi-agent setting involving *strategic* agents influencing a *single* linear dynamical system. Our state dynamics are not separable and are influenced by all the agents. The cost of each agent in our model is influenced by the (shared) observed state which is governed by all the agents' control inputs and the goal of each agent is to maximize their own individual cost.

**Markov Games.** Regret bounds have been previously established for discrete finite Markov games. Our multi-agent linear control setting can be seen as a continuous analog to Markov games. However, note that our linear dynamical system is fundamentally different from the usual Markov game (or stochastic game) setting involving an unknown state transition kernel outputting the next state probability distribution as a function of the current state and the (joint) actions of all players. When considering multi-agent potential games, there are three important distinctions with existing works on Markov potential games (e.g. Leonardos et al. (2022); Zhang et al. (2024); Ding et al. (2022); Zhang et al. (2022b); Sun et al. (2023)):

- In our work, the state and action spaces are continuous and are not mixed extensions of finite sets of states and actions. Most of the bounds scale with the cardinality of the action spaces of the players and are therefore vacuous in our continuous action space setting. In addition, our results use a suitable control policy for the linear dynamical system setting. The softmax policy used in e.g. Zhang et al. (2022b); Sun et al. (2023) is not immediately suitable for the continuous case, unless one puts a parametric probability distribution assumption on the disturbance sequence, which we want to avoid in order to consider adversarial disturbances.

- Our results consider adversarial disturbances, and hence the state transitions of the underlying system may not even be Markovian or stochastic, the disturbances can be chosen adversarially depending on the far past.

- Our work considers cost functions that are time-varying, which is in contrast with the standard fixed reward setting in the mentioned Markov potential games works. We also do not consider discounted rewards, and the potential assumption we use is with respect to the cost function itself, and not on the aggregate cost over a time horizon.

## B EXAMPLES

### B.1 DESCRIPTION

We provide a few concrete examples to illustrate our multi-agent control setting.

**(a) Smart grid markets.** In modern power grids, electricity is generated and distributed by a mix of independent energy producers such as traditional plants and renewable energy providers. These actors act selfishly and adapt to market conditions while they also jointly influence the grid. Let $x_t$ be the grid state defined by characteristics such as line loads and aggregate reserves, let $u_t^i$ be generator $i$'s power output decision (i.e. their control input) and let the sequence $w_t$ capture the demand fluctuation, the system noise and/or renewable energy shocks. Then, the system dynamics may evolve according to (LDS) (e.g. by linearization around an operating point). Each generator $i$ has their local cost function which accounts for the cost of production including e.g. fuel and a penalty for deviating from a target grid state.

**(b) Formation control.** Consider a multi-agent system consisting of $N$ vehicles or robots. The state (position, velocity) and control input of each agent $i$ at each time step $t$ are respectively given by $x_t^i$ and $u_t^i$. Suppose the (joint) state of the multi-agent system evolves according to (LDS). The formation of the multi-agent system is defined by specifying a desired distance to be maintained over time between the states of agents that are adjacent. The goal of each agent is to minimize their own formation error and energy consumption. A similar formation control problem has been studied in the control literature *in the absence of adversarial perturbations* ($w_t = 0$) using differential games (see

e.g. Aghajani & Doustmohammadi (2015); Han et al. (2019)) and discrete linear quadratic games (Hosseinirad et al., 2023).

**(c) Bioresource management.** A set of firms (or countries) exploit a set of renewable resources (e.g. a fish population) whose evolution is driven by (LDS) where $x_t \in \mathbb{R}^d$ denotes the vector of quantities of $d$ distinct resources, the matrix $A$ encodes their natural growth rate, the control $u_t^i$ models the exploitation rate of the firm $i$ and $w_t$ refers to perturbations due to exogenous factors such as weather conditions. Each firm $i$ has the goal to maximize their profit while minimizing their exploitation cost. See e.g. Mazalov et al. (2017) in the noiseless setting ($w_t = 0$).

### B.2 ABOUT ADVERSARIAL DISTURBANCES

In multi-agent systems, considering adversarial disturbances allows us to model a wide range of realistic, worst-case, or strategically motivated perturbations ranging from strategic behavior in energy markets to adversarial environments in robotics and ecological shocks in resource management, ensuring system robustness even under hostile or extreme scenarios.

We provide below examples of adversarial disturbances in each of the examples described in section B.1 above and comment on their importance:

- **Smart grid markets:** An adversarial disturbance could model sudden demand spikes, strategic demand manipulation by large consumers (i.e. major electricity buyers who have significant influence over the overall demand on the grid), malicious data injection attacks that falsify renewable generation forecasts or misreporting. For instance, an actor might manipulate demand predictions to influence market prices or grid loads in their favor.

- **Formation control:** Adversarial disturbances capture environmental disturbances with structured worst-case behavior, such as wind gusts or magnetic interference that affect formations in potentially harmful ways. It can also capture adversarial agents or spoofed sensor data to destabilize the formation. In hostile or uncertain environments (e.g., surveillance drones in contested airspace), agents must maintain formation despite external influences that could intentionally disrupt coordination.

- **Bioresource management:** Adversarial disturbances may reflect deliberate misinformation about resource levels, illegal over-harvesting by untracked actors, or policy shocks (e.g., sudden trade bans) that drastically affect the resource dynamics in a harmful way. Robust resource management must consider these disturbances to avoid collapse or irreversible damage.

## C FURTHER DISCUSSION OF ASSUMPTIONS

### C.1 ASSUMPTION 2

To the best of our knowledge, all prior works in the online control literature assume bounded adversarial disturbances. It would be interesting to relax this assumption further to model other scenarios involving catastrophic failures or highly irrational agents. As for the boundedness of the control inputs, note that this property is automatically satisfied using the gradient-based controllers considered via the projection of policy parameters.

### C.2 ASSUMPTION 3

As is standard in prior work on single-agent online control, we assume that agents have initial access to a stabilizing controller. Note that such controllers can be obtained offline using an SDP relaxation (e.g., using the method of Cohen et al. (2018)). Our main focus is on the challenging task of learning DAC policy parameters under adversarial disturbances.

### C.3 ASSUMPTION 4

Global stability is a key property enabling the linear dependence on N in the regret bound. There are two explanations for this depending on whether or not all agents in the population play DAC policies.

- First, without assuming the specific policies of other agents in the population, assume agent-wise strong stability holds (Assumption 3) in the Aggregated Control learning setting. Then, agent $i$

can locally compute the true disturbances and run their DAC policy w.r.t. this true disturbance sequence. However, bounding the individual regret of agent $i$ requires controlling the magnitude of the norm of the global state, and without any assumptions on the control policy of other agents, their "contributions" to the state evolution can only crudely be treated as an "error" term. With $(N-1)$ other agents in the population, the norm of the state will still scale linearly in $N$ in the worst case resulting in an $N^2$ dependence in the regret bound for agent $i$.

- On the other hand, suppose we assume all agents in the population play DAC policies. While it is possible to show that agent-wise strong stability (Assumption 3) implies global strong stability (Assumption 4), the resulting parameters for global strong stability will depend on the number of agents $N$ (note that it is natural that local strong stability does not imply global strong stability with the same constant parameter values, independently of $N$). Therefore, when applying the machinery of the proof of Theorem 3.4 using the resulting global strong stability parameters (which depend on $N$), the final regret bound will still have at least an $N^2$ dependence.

## D  PREPARATORY RESULTS FOR THE MAIN PROOFS

### D.1  NOTATION: COUNTERFACTUAL AND IDEALIZED STATES AND ACTIONS

We introduce a few useful notations in view of our regret analysis. We focus on agent $i$'s viewpoint and we suppose that other players are using a given sequence of control inputs $\{u_t^{-i}\}$. We will not highlight this dependence in the notation below to avoid overloaded notations as it will be clear from the context.

- *Counterfactual state and action:* We use the notation $x_t^{K_i}(M_{i,0:t-1})$ for the state reached by the system by execution of the non-stationary policy $\pi_i(M_{i,0:t-1}, K_i)$, and $u_t^{i,K_i}(M_{i,0:t-1})$ is the action executed at time $t$. If the same (stationary) policy $M_i$ is used by agent $i$ in all time steps, we use the more compact notation $x_t^{K_i}(M_i), u_t^{i,K_i}(M_i)$. We use the notation $x_t^{K_i}(0), u_t^{i,K_i}(0)$ for the linear control policy $K_i$.

- *Ideal state and action:* We denote by $y_{t+1}^{K_i}(M_{i,t-H:t})$ the ideal state of the system that would have been reached if agent $i$ played the non-stationary policy $M_{i,t-H:t}$ from time step $t-H$ to $t$ assuming that the state at time $t-H$ is zero while other agents use the control sequence $\{u_{t-H:t}^{-i}\}$. The ideal action to be executed at time $t+1$ if the state observed at time $t+1$ is $y_{t+1}^{i,K_i}(M_{i,t-h:t})$ will be denoted by $v_{t+1}^{i,K_i}(M_{i,t-H:t+1}) = -K_i y_{t+1}^{i,K_i}(M_{i,t-H:t}) + \sum_{p=1}^H M_{i,t+1}^{[p-1]} w_{t+1-p}$. We use the compact notations $y_{t+1}^{i,K_i}(M_i), v_{t+1}^{i,K_i}(M_i)$ when $M_i$ is constant across time steps $t-H$ to $t$.

- *Ideal cost:* Let $\ell_t^i(M_{i,t-1-H:t}) = c_t^i(y_t^{i,K_i}(M_{i,t-1-H:t-1}), v_t^{i,K_i}(M_{i,t-1-H:t}))$ be agent $i$'s cost function evaluated at the idealized state and action pair. Again we use the notation $\ell_t^i(M_i)$ when $M_i$ is constant across time steps $t-H$ to $t$. Importantly, for every agent $i \in [N]$, the function $\ell_t^i$ is a convex function of $M_{i,t-H-1:t}$ under assumption 1: This is because the cost function of agent $i$ is supposed to be convex w.r.t. both its arguments and both ideal state and action are linear transformations of $M_{i,t-H-1:t}$ (see Lemma 3.1 and its proof). Introducing and using this idealized cost which only involves the past $H$ controllers brings us to online convex optimization with memory (Anava et al., 2015).

### D.2  STATE EVOLUTION

In view of our analysis, we describe first the state evolution under (LDS). We introduce first some useful notations for any $i \in [N], t, h \le t, l \le H + h$:

$$\tilde{A}_{K_i} := A - B_i K_i, \quad \Psi_{t,l}^{i,h}(M_{i,t-h:t}) := \tilde{A}_{K_i}^l \mathbf{1}_{l \le h} + \sum_{k=0}^h \tilde{A}_{K_i}^k B_i M_{i,t-k}^{[l-k-1]} \mathbf{1}_{l-k \in [1,H]}, \quad (9)$$

$$\bar{A}_K := A - \sum_{i=1}^N B_i K_i, \quad \bar{\Psi}_{t,l}^h(M_{t-h:t}) := \bar{A}_K^l \mathbf{1}_{l \le h} + \sum_{k=0}^h \bar{A}_K^k \sum_{i=1}^N B_i M_{i,t-k}^{[l-k-1]} \mathbf{1}_{l-k \in [1,H]}. \quad (10)$$

Here, when player $i$ plays a DAC policy (DAC-$i$) and other players' control inputs are given by $\{u_t^{-i}\}$, the matrix $\tilde{A}_{K_i}$ describes the evolution of the state when agent $i$ executes the linear controller $K_i$ in the absence of disturbances and other players, and $\Psi_{t,l}^{i,h}(M_{i,t-h:t})$ is the disturbance-state transfer matrix for agent $i$ which will describe the influence of the perturbation term $w_{t-l}$ on the next state $x_{t+1}$ at time $t+1$. When all agents execute a DAC policy (DAC-$i$), the evolution of the state is driven by the matrix $\bar{A}_K$ and the influence of the perturbation term $w_{t-l}$ on the next state $x_{t+1}$ is captured by the disturbance-state transfer matrix $\bar{\Psi}_{t,l}^h(M_{t-h:t})$. Using these notations we have the following result describing the evolution of the states under (LDS) extending the single-agent result of Agarwal et al. (2019) (Lemma 4.3).

**Proposition D.1.** *(State evolution) Suppose all agents but $i \in [N]$ select their actions according to the sequence of control inputs $\{u_t^{-i}\}$ then for every time $t$ and every $h \geq 0$, if agent $i \in [N]$ executes a non-stationary DAC policy $\pi_i(M_{i,0:T}, K_i)$, the state of the system (LDS) is as follows:*

*(i) Under Setting 1, i.e. with perturbation sequence $\tilde{w}_t := w_t + \sum_{j \neq i} B_j u_{t-k}^j$,*

$$x_{t+1} = \tilde{A}_{K_i}^{h+1} x_{t-h} + \sum_{l=0}^{H+h} \Psi_{t,l}^{i,h}(M_{i,t-h:t})\tilde{w}_{t-l}\,. \tag{11}$$

*(ii) Under Setting 2, if in addition **all** the agents execute a DAC policy using the sequence $\{w_t\}$,*

$$x_{t+1} = \bar{A}_K^{h+1} x_{t-h} + \sum_{l=0}^{H+h} \bar{\Psi}_{t,l}^h(M_{t-h:t})w_{t-l}\,. \tag{12}$$

This result follows from unrolling the state dynamics for $h$ steps, injecting the DAC policy for agent $i$ (or all agents depending on the setting) and rewriting the state evolution to highlight the linear dependence of the state on the previous disturbances. We defer a complete constructive proof to Appendix D.2. Importantly, notice that $\Psi_{t,l}^{i,h}$ and $\bar{\Psi}_{t,l}^h$ are linear in the $h+1$ DAC policy parameters $M_{i,t-h:t}, i \in [N]$.

*Proof.* We prove the two claims of the Proposition separately:

**Proof of Claim (i).** The proof of the first part of the statement under Setting 1 is a direct application of the known single-agent result (Agarwal et al., 2019, Lemma 4.3) with the new disturbance sequence $\{\tilde{w}_t\}$ rather than the original disturbance sequence $\{w_t\}$ defining (LDS).

**Proof of Claim (ii).** We provide a full constructive proof which clarifies how we obtain our final state evolution expression. Observe first that

$$x_{t+1} = Ax_t + \sum_{i=1}^N B_i u_t^i + w_t \qquad \text{(using (LDS))}$$

$$= Ax_t + \sum_{i=1}^N B_i \left(-K_i x_t + \sum_{p=1}^H M_{i,t}^{[p-1]} w_{t-p}\right) + w_t \qquad \text{(using non-stat.(DAC-}i\text{))}$$

$$= \left(A - \sum_{i=1}^N B_i K_i\right) x_t + \sum_{i=1}^N \left(B_i \sum_{p=1}^H M_{i,t}^{[p-1]} w_{t-p}\right) + w_t\,,$$

$$= \bar{A}_K x_t + \tilde{\varphi}_{t,i}^0\,, \tag{13}$$

where we define: $\tilde{\varphi}_t^0 := \sum_{i=1}^N \left(B_i \sum_{p=1}^H M_{i,t}^{[p-1]} w_{t-p}\right) + w_t$. Expanding again the state $x_t$ yields:

$$x_{t+1} = \bar{A}_K x_t + \tilde{\varphi}_t^0 \qquad \text{(see (13))}$$

$$= \bar{A}_K \left(\bar{A}_K x_{t-1} + \sum_{i=1}^N \left(B_i \sum_{p=1}^H M_{i,t-1}^{[p-1]} w_{t-1-p}\right) + w_{t-1}\right) + \tilde{\varphi}_t^0 \quad \text{(same steps as in (13))}$$

$$= \bar{A}_K^2 x_{t-1} + \tilde{\varphi}_{t-1}^1 + \tilde{\varphi}_t^0\,, \tag{14}$$

where we define for every $k = 0, \cdots, h$:

$$\tilde{\varphi}_{t-k}^k := \bar{A}_K^k \sum_{i=1}^N \left( B_i \sum_{p=1}^H M_{i,t-k}^{[p-1]} w_{t-k-p} \right) + \bar{A}_K^k w_{t-k} , \qquad (15)$$

where we note for precision that the last term is not in the sum over $i$. Unrolling the recursion (14) for $h$ steps yields

$$x_{t+1} = \bar{A}_K^{h+1} x_{t-h} + \sum_{k=0}^h \tilde{\varphi}_{t-k}^k . \qquad (16)$$

It now remains to rewrite the second term in the above expression:

$$\sum_{k=0}^h \tilde{\varphi}_{t-k}^k = \sum_{k=0}^h \bar{A}_K^k \left( \sum_{i=1}^N B_i \sum_{p=1}^H M_{i,t-k}^{[p-1]} \right) w_{t-k-p} + \bar{A}_K^k w_{t-k} \qquad \text{(using definition (15))}$$

$$= \sum_{l=1}^{H+h} \left( \sum_{k=0}^h \bar{A}_K^k \left( \sum_{i=1}^N B_i M_{i,t-k}^{[l-k-1]} \right) \mathbf{1}_{l-k \in [1,H]} w_{t-l} + \bar{A}_K^k w_{t-k} \right)$$

$$\qquad \text{(index change } l = k+p, 0 \le k \le h, 1 \le p \le H)$$

$$= \sum_{l=0}^{H+h} \left( \bar{A}_K^l \mathbf{1}_{l \le h} + \sum_{k=0}^h \bar{A}_K^k \sum_{i=1}^N B_i M_{i,t-k}^{[l-k-1]} \mathbf{1}_{l-k \in [1,H]} \right) w_{t-l} \qquad \text{(simplifying 1st term)}$$

$$= \sum_{l=0}^{H+h} \bar{\Psi}_{t,l}^h(M_{t-h:t}) w_{t-l} . \qquad \text{(using definition of } \bar{\Psi}_{t,l}^h \text{ in (10)) .}$$

$$(17)$$

$\square$

## D.3 Transfer matrix bound

In view of our regret analysis, it will be useful to bound the norm of the states and actions. Given the expression of the state evolution shown in Proposition D.1-(ii), we will need to bound the norm of the state transfer matrix. This is the purpose of the next lemma which is similar to (Agarwal et al., 2019, Lemma 5.4).[2] However, our transfer matrix which is induced by all agents playing DAC-$i$ policies is different from their single-agent counterpart.

**Lemma D.2.** *Let the global strong stability assumption 4 hold, i.e. suppose that $K = (K_1, \cdots, K_N)^T$ is $(\bar{\kappa}, \bar{\gamma})$-strongly stable for $(A, [B_1, \cdots, B_N])$. Let $M_{i,t}$ be a sequence s.t. for all $t, p \in \{0, \cdots, H-1\}, \|M_{i,t}^{[p]}\| \le \tau(1-\bar{\gamma})^p$ where $\tau$ is some positive constant. Then for all $t \ge 1, h \le t$ and $l \le H + h$, we have*

$$\|\bar{\Psi}_{t,l}^h(M_{t-h:t})\| \le \bar{\kappa}(1-\bar{\gamma})^l \cdot \mathbf{1}_{l \le H} + H\bar{\kappa}\tau \left( \sum_{i=1}^N \|B_i\| \right) (1-\bar{\gamma})^{l-1} . \qquad (18)$$

*Proof.* Recall the definition of $\bar{\Psi}_{t,l}^h$ from (10):

$$\bar{\Psi}_{t,l}^h(M_{t-h:t}) := \bar{A}_K^l \mathbf{1}_{l \le h} + \sum_{k=0}^h \bar{A}_K^k \sum_{i=1}^N B_i M_{i,t-k}^{[l-k-1]} \mathbf{1}_{l-k \in [1,H]} . \qquad (19)$$

Using strong stability of $K$ (see definition 2.1), there exists matrices $L, Q$ s.t. $\bar{A}_K = A - \sum_{i=1}^N B_i K_i = QLQ^{-1}$ with $\|L\| \le 1 - \bar{\gamma}$, and $\|Q\| \cdot \|Q^{-1}\| \le \bar{\kappa}$. Therefore using the sub-multiplicativity of the norm we obtain for every $l = 0, \cdots, t$,

$$\|\bar{A}_K^l\| = \|(QLQ^{-1})^l\| = \|QL^lQ^{-1}\| \le \|Q\| \cdot \|Q^{-1}\| \cdot \|L\|^l \le \bar{\kappa}(1-\bar{\gamma})^l . \qquad (20)$$

---

[2]Note here that our powers of $\kappa$ are slightly different because we stick to the definition of $(\kappa, \gamma)$-strong stability introduced in Cohen et al. (2018) rather than the one later used in Agarwal et al. (2019) which is slightly different, this is without any loss of generality.

Therefore, we can bound the norm of the state transfer matrix in (19) as follows:

$$\|\bar{\Psi}_{t,l}^h(M_{t-h:t})\| \le \|\bar{A}_K^l\|\mathbf{1}_{l \le h} + \sum_{k=0}^{h} \|\bar{A}_K^k\| \cdot \sum_{i=1}^{N} \|B_i\| \cdot \|M_{i,t-k}^{[l-k-1]}\| \cdot \mathbf{1}_{l-k \in [1,H]}$$

$$\le \bar{\kappa}(1-\bar{\gamma})^l \cdot \mathbf{1}_{l \le h} + \bar{\kappa}\tau \sum_{i=1}^{N} \|B_i\| \sum_{k=0}^{h} (1-\bar{\gamma})^k (1-\bar{\gamma})^{l-k-1} \mathbf{1}_{l-k \in [1,H]}$$

$$\le \bar{\kappa}(1-\bar{\gamma})^l \cdot \mathbf{1}_{l \le H} + H\bar{\kappa}\tau \left( \sum_{i=1}^{N} \|B_i\| \right) (1-\bar{\gamma})^{l-1} , \tag{21}$$

where the second inequality stems from using strong stability (see (20)) and the assumed bound $\|M_{i,t}^{[p]}\| \le \tau(1-\bar{\gamma})^p$ for $p \in \{0, \cdots, H-1\}$. As for the last inequality, observe after simplification that the summand does not depend on the index $k$ of the sum apart from the indicator function and there are at most $H$ terms in the sum (since $l-H \le k \le l-1$ as $l-k \in \{1, \cdots, H\}$). $\qquad\square$

### D.4 STATE, ACTION AND DIFFERENCE OF STATE AND ACTION BOUNDS

The goal of the next proposition is to control the norms of states, actions and differences of states and actions. Note that we pay particular attention to the problem constants involved to elucidate the dependence of our bounds on the number of agents $N$ and the magnitude of the control inputs of all the agents. The result is a more refined version of (Agarwal et al., 2019, Lemma 5.5) which is adapted to our multi-agent control setting when each agent executes a (DAC-$i$) policy.

**Proposition D.3.** *Let Assumption 4 hold. Let the perturbation sequence $\{w_t\}$ in (LDS) satisfy Assumption 2. Let $M_{i,t}$ be a sequence s.t. for any time step $t$, for $p \in \{0, \cdots, H-1\}$, $\|M_{i,t}^{[p]}\| \le \tau(1-\bar{\gamma})^p$ for some $\tau > 0$. Let $K = (K_1, \cdots, K_N), K = (K_1^*, \cdots, K_N^*)$ be s.t. $K$ and $K^*$ are two $(\bar{\kappa}, \bar{\gamma})$-strongly stable matrices. Then the following holds:*

   *(i) **State under** (DAC-$i$): For every $t \ge H+1$,*

$$\|x_t^K(M_{0:t-1})\| \le \frac{\bar{\kappa}}{\bar{\gamma}} \cdot \frac{W(1 + \tau H \sum_{i=1}^{N} \|B_i\|)}{1 - \bar{\kappa}(1-\bar{\gamma})^{H+1}} . \tag{22}$$

   *(ii) **Ideal state under** (DAC-$i$): For every $t \ge H+1$,*

$$\|y_t^K(M_{t-1-H:t-1})\| \le \frac{\bar{\kappa}}{\bar{\gamma}} W \left( 1 + \tau H \sum_{i=1}^{N} \|B_i\| \right) . \tag{23}$$

   *(iii) **Linear controller state**: For every $t \ge 0$, $\|x_t^{K^*}(0)\| \le \frac{\bar{\kappa}}{\bar{\gamma}} W$.*

   *(iv) **Action under** (DAC-$i$): For every $t \ge H+1$,*

$$\|u_t^{i,K}(M_{0:t})\| \le \frac{\bar{\kappa}^2}{\bar{\gamma}} \cdot \frac{W(1 + \tau H \sum_{i=1}^{N} \|B_i\|)}{1 - \bar{\kappa}(1-\bar{\gamma})^{H+1}} + \frac{\tau}{\bar{\gamma}} W . \tag{24}$$

   *(v) **Ideal action under** (DAC-$i$): For every $t \ge H+1$,*

$$\|v_t^{i,K}(M_{t-1-H:t})\| \le \frac{\bar{\kappa}^2}{\bar{\gamma}} W \left( 1 + \tau H \sum_{i=1}^{N} \|B_i\| \right) + \frac{\tau}{\bar{\gamma}} W . \tag{25}$$

   *(vi) **State vs. ideal state comparison**: For every $t \ge H+1$,*

$$\|x_t^K(M_{0:t-1}) - y_t^K(M_{t-1-H:t-1})\| \le (1-\bar{\gamma})^H \frac{\bar{\kappa}^2}{\bar{\gamma}} \cdot \frac{W(1 + \tau H \sum_{i=1}^{N} \|B_i\|)}{1 - \bar{\kappa}(1-\bar{\gamma})^{H+1}} . \tag{26}$$

   *(vii) **Action vs ideal action comparison**: For every $t \ge H+1$,*

$$\|u_t^{i,K}(M_{0:t}) - v_t^{i,K}(M_{t-1-H:t})\| \le (1-\bar{\gamma})^H \frac{\bar{\kappa}^3}{\bar{\gamma}} \cdot \frac{W(1 + \tau H \sum_{i=1}^{N} \|B_i\|)}{1 - \bar{\kappa}(1-\bar{\gamma})^{H+1}} . \tag{27}$$

*(viii) Moreover, given all the above bounds, if $H + 1 \geq \frac{\ln(2\bar{\kappa})}{\bar{\gamma}}$ (where $\bar{\kappa} \geq 1$ without loss of generality), then we have the following simultaneous bounds:*

$$\max_{t \geq H+1} \left\{ \|x_t^K(M_{0:t-1})\|, \|y_t^K(M_{t-1-H:t-1})\|, \|x_t^{K^*}(0)\| \right\} \leq D, \qquad (28)$$

$$\max_{t \geq H+1} \left\{ \|u_t^{i,K}(M_{0:t})\|, \|v_t^{i,K}(M_{t-1-H:t})\| \right\} \leq D, \qquad (29)$$

$$\max_{t \geq H+1} \left\{ \|x_t^K(M_{0:t-1}) - y_t^K(M_{t-1-H:t-1})\|, \|u_t^{i,K}(M_{0:t}) - v_t^{i,K}(M_{t-1-H:t})\| \right\} \leq (1-\bar{\gamma})^H D, \qquad (30)$$

*where the constant $D$ is defined as follows as a function of the problem parameters:*

$$D := \frac{6\bar{\kappa}^3}{\bar{\gamma}} W \left( 1 + \bar{\kappa}^2 H \sum_{i=1}^N \|B_i\| \right). \qquad (31)$$

*Note in particular that $D = \mathcal{O}(N)$ where the notation $\mathcal{O}(\cdot)$ here hides all other constants which are independent of the number $N$ of agents.*

*Proof.* We prove each one of the statements of the proposition separately.

**Proof of Claim (i).** Using Proposition D.1-(ii) at time step $t - 1$ with $h = H$, we have

$$x_t^K(M_{0:t-1}) = \bar{A}_K^{H+1} x_{t-1-H}(M_{0:t-2-H}) + \sum_{l=0}^{2H} \bar{\Psi}_{t-1,l}^H(M_{t-1-h:t-1}) w_{t-1-l}. \qquad (32)$$

It follows from using the boundedness of the perturbation sequence $\{w_t\}$ by $W$, the $(\bar{\kappa}, \bar{\gamma})$-strong stability of the matrix $K$ (see Eq. (20)) that

$$\|x_t^K(M_{0:t-1})\| \leq \bar{\kappa}(1-\bar{\gamma})^{H+1}\|x_{t-1-H}(M_{0:t-2-H})\| + W \sum_{l=0}^{2H} \|\bar{\Psi}_{t-1,l}^H(M_{t-1-h:t-1})\|. \qquad (33)$$

Now invoking Lemma D.2 at time $t - 1$ with $h = H$ yields for every $l \leq 2H, t \geq 1$:

$$\|\bar{\Psi}_{t-1,l}^h(M_{t-1-h:t-1})\| \leq \bar{\kappa}(1-\bar{\gamma})^l \cdot \mathbf{1}_{l \leq H} + \bar{\kappa}\tau H \left( \sum_{i=1}^N \|B_i\| \right) (1-\bar{\gamma})^{l-1}. \qquad (34)$$

As a consequence, we have by summing these bounds over $l = 0, \cdots, 2H$,

$$\sum_{l=0}^{2H} \|\bar{\Psi}_{t-1,l}^h(M_{t-1-h:t-1})\| \leq \bar{\kappa} \sum_{l=0}^H (1-\bar{\gamma})^l + \bar{\kappa}\tau H \sum_{i=1}^N \|B_i\| \sum_{l=1}^{2H} (1-\bar{\gamma})^{l-1} \leq \frac{\bar{\kappa}}{\bar{\gamma}} \left( 1 + \tau H \sum_{i=1}^N \|B_i\| \right).$$

Therefore we obtain

$$\|x_t^K(M_{0:t-1})\| \leq \bar{\kappa}(1-\bar{\gamma})^{H+1}\|x_{t-1-H}(M_{0:t-2-H})\| + \frac{\bar{\kappa}}{\bar{\gamma}} W \left( 1 + \tau H \sum_{i=1}^N \|B_i\| \right). \qquad (35)$$

Unrolling the recursion results in the desired state norm bound:

$$\|x_t^K(M_{0:t-1})\| \leq \frac{\bar{\kappa}}{\bar{\gamma}} \cdot \frac{W(1 + \tau H \sum_{i=1}^N \|B_i\|)}{1 - \bar{\kappa}(1-\bar{\gamma})^{H+1}}. \qquad (36)$$

**Proof of Claim (ii).** Recall that $y_t^K(M_{t-1-H:t-1})$ is the ideal system state that would have been reached if each agent $i$ played the non-stationary policy $M_{i,t-1-H:t-1}$ from time step $t - 1 - H$ to $t - 1$ assuming that the state at time $t - 1 - H$ is zero. Therefore, similarly to (32) it follows that

$$y_t^K(M_{t-1-H:t-1}) = \sum_{l=0}^{2H} \bar{\Psi}_{t-1,l}^H(M_{t-1-h:t-1}) w_{t-1-l}. \qquad (37)$$

Using similar steps as for the proof of (i) results in the following desired bound:

$$\|y_t^K(M_{t-1-H:t-1})\| \le \frac{\bar{\kappa}}{\bar{\gamma}} W \left( 1 + \tau H \sum_{i=1}^{N} \|B_i\| \right) . \tag{38}$$

**Proof of Claim (iii).** Observe that for any time step $t \ge 1$, the state induced by linear controllers $K^* = (K_1^*, \cdots, K_N^*)$ is given by

$$x_t^{K^*}(0) = \sum_{l=0}^{t-1} \bar{A}_{K^*}^l w_{t-1-l} . \tag{39}$$

As a consequence, using $(\bar{\kappa}, \bar{\gamma})$-strongly stability of $K^*$ together with boundedness of the perturbation sequence $\{w_t\}$ and the sum of the geometric series by $1/\bar{\gamma}$, we have for every time step $t \ge 1$:

$$\|x_t^{K^*}(0)\| \le \frac{\bar{\kappa}}{\bar{\gamma}} W , \tag{40}$$

and this concludes the proof.

**Proof of Claim (iv).** Note first that action $u_t^{i,K_i}(M_{i,0:t})$ is computed using (DAC-$i$) policy as follows:

$$u_t^{i,K}(M_{0:t}) = -K_i x_t^K(M_{0:t-1}) + \sum_{p=1}^{H} M_{i,t}^{[p-1]} w_{t-p} . \tag{41}$$

Using the $(\bar{\kappa}, \bar{\gamma})$-strong stability of $K$ (and without loss of generality $\|K_i\| \le \bar{\kappa}$) and the bound assumption on $M_{i,t}$ together with the state bound already established in item (i), we obtain

$$\|u_t^{i,K}(M_{0:t})\| \le \bar{\kappa}\|x_t^K(M_{0:t-1})\| + W\frac{\tau}{\bar{\gamma}} \le \frac{\bar{\kappa}^2}{\bar{\gamma}} \cdot \frac{W(1 + \tau H \sum_{i=1}^{N} \|B_i\|)}{1 - \bar{\kappa}(1-\bar{\gamma})^{H+1}} + W\frac{\tau}{\bar{\gamma}} . \tag{42}$$

**Proof of Claim (v).** By definition of the ideal action $v_t^{i,K}(M_{t-1-H:t})$ given the ideal state $y_t^K(M_{t-1-H:t-1})$, we have:

$$v_t^{i,K}(M_{t-1-H:t}) = -K_i y_t^K(M_{t-1-H:t-1}) + \sum_{p=1}^{H} M_{i,t}^{[p-1]} w_{t-p} . \tag{43}$$

Therefore we can bound the ideal action as follows similarly to the proof of item (iv) using the ideal state bound already established in item (ii) to obtain

$$\|v_t^{i,K}(M_{t-1-H:t})\| \le \bar{\kappa}\|y_t^K(M_{t-1-H:t-1})\| + W\frac{\tau}{\bar{\gamma}} \le \frac{\bar{\kappa}^2}{\bar{\gamma}} W \left( 1 + \tau H \sum_{i=1}^{N} \|B_i\| \right) + W\frac{\tau}{\bar{\gamma}} . \tag{44}$$

**Proof of Claim (vi).** It follows from combining the state evolution expressions (32) and (37) that

$$\|x_t^K(M_{0:t-1}) - y_t^K(M_{t-1-H:t-1})\| = \|\bar{A}_K^{H+1} x_{t-1-H}(M_{0:t-2-H})\| \tag{45}$$

$$\le \bar{\kappa}(1-\bar{\gamma})^H \|x_{t-1-H}(M_{0:t-2-H})\| . \tag{46}$$

Plugging in again the state bound (item (i)-(22)) in the above inequality yields the desired inequality:

$$\|x_t^K(M_{0:t-1}) - y_t^K(M_{t-1-H:t-1})\| \le (1-\bar{\gamma})^H \frac{\bar{\kappa}^2}{\bar{\gamma}} \cdot \frac{W(1 + \tau H \sum_{i=1}^{N} \|B_i\|)}{1 - \bar{\kappa}(1-\bar{\gamma})^{H+1}} . \tag{47}$$

**Proof of Claim (vii).** Using the definitions of the actions $u_t^{i,K}(M_{0:t})$ and $v_t^{i,K}(M_{t-1-H:t})$ in (41)-(43), we immediately have:

$$\|u_t^{i,K}(M_{0:t}) - v_t^{i,K}(M_{t-1-H:t})\| = \|K_i(y_t^K(M_{t-1-H:t-1}) - x_t^K(M_{0:t-1}))\|$$

$$\le \bar{\kappa}\|y_t^K(M_{t-1-H:t-1}) - x_t^K(M_{0:t-1})\|$$

$$\le (1-\bar{\gamma})^H \frac{\bar{\kappa}^3}{\bar{\gamma}} \cdot \frac{W(1 + \tau H \sum_{i=1}^{N} \|B_i\|)}{1 - \bar{\kappa}(1-\bar{\gamma})^{H+1}} , \tag{48}$$

where the last inequality stems from using the inequality established in item (vii)-(47).

**Proof of Claim (viii).** Set $\tau = 2\kappa^2$. If $H + 1 \geq \frac{\ln(2\bar{\kappa})}{\bar{\gamma}}$, then $\bar{\kappa}(1 - \bar{\gamma})^{H+1} \leq \frac{1}{2}$. Using this bound and the fact that $\bar{\kappa} \geq 1$ without loss of generality (replace $\bar{\kappa}$ by $\max\{1, \bar{\kappa}\}$ otherwise), it is easy to see that we obtain the desired bounds with the same constant $D$ by taking the maximum of all the bounds appearing in the inequalities of Proposition D.3 . $\qquad\square$

## E    PROOF OF THEOREM 3.2

Here, we give the proof of Theorem 3.5, which we restate here:

**Theorem 3.2 (Individual Regret in Setting 1, Independent Learning).** *Let Assumptions 1, 2 and 3 hold. Suppose there exists $U > 0$ s.t. for all $t \geq 0, j \in [N], \|u_t^j\| \leq U$. If agent $i \in [N]$ runs Algorithm 1 under Setting 1 with (DAC-i) policy on perturbation sequence $\{\widetilde{w}_t\}$ and step size $\eta = \Theta(1/(G\widetilde{W}\sqrt{T}))$, where $\widetilde{W} = W + (N - 1)U(\max_j \|B_j\|)$, and with $H \geq \log(\kappa_i T)/\gamma_i$, then for any $T \geq H + 1$, we have $\mathrm{Reg}_i^{H+1:T}(\mathcal{A}_i, \{u_t^{-i}\}, \Pi_i^{lin}) = \widetilde{\mathcal{O}}(U^2 N^2 \sqrt{T})$.* [3]

**Remark E.1.** *The notation $\widetilde{\mathcal{O}}$ in Theorem 3.2 hides polynomial factors in $\gamma_i^{-1}, \kappa_i, \|B_i\|, G, d$ and logarithmic factors in $T$ .*

*Proof.* Under Assumptions 1, 2 and 3, we apply Agarwal et al. (2019, Theorem 5.1) for each agent $i \in [N]$. It remains to ensure that the considered perturbation sequence $\{\widetilde{w}_t\}$ in (5) also satisfies the boundedness condition of Assumption 2 using the boundedness of control inputs by $U$ as follows:

$$\|\widetilde{w}_t\| = \left\|\sum_{j\neq i} B_j u_t^j + w_t\right\| \leq \|w_t\| + \sum_{j\neq i} \|B_j\| \cdot \|u_t^j\| \leq W + (N-1)U(\max_j \|B_j\|), \quad (49)$$

where the last inequality follows from using boundedness of the control inputs of all the agents together with the bounded disturbances assumption (Assumption 2).

Selecting a step size $\eta = \Theta(1/(G\widetilde{W}\sqrt{T}))$, where $\widetilde{W} = W + (N - 1)U(\max_j \|B_j\|)$, and a (per-agent) memory length $H \geq \log(\kappa_i T)/\gamma_i$, we obtain the desired regret for any $T \geq H + 1$,

$$\mathrm{Reg}_i^{H+1:T}(\mathcal{A}_i, \{u_t^{-i}\}, \Pi_i^{\mathrm{lin}}) = \widetilde{\mathcal{O}}(U^2 N^2 \sqrt{T}). \quad (50)$$

This concludes the proof. $\qquad\square$

## F    PROOF OF THEOREM 3.4

This section is devoted to developing the proof of Theorem 3.4, which we restate here:

**Theorem 3.4 (Individual Regret in Setting 2).** *Let Assumptions 1, 2, 4 hold. Then if agent $i \in [N]$ runs Algorithm 1 under Setting 2 with a (DAC-i) policy on perturbation sequence $\{w_t\}$, step size $\eta = \Theta(1/N\sqrt{T})$, and with $H \geq \log(2\bar{\kappa}N^2\sqrt{T})/\bar{\gamma}$, and when **all** other agents use a (DAC-i) policy with perturbation sequence $(w_t)$, then for any $T \geq H + 1$: $\mathrm{Reg}_i^{H+1:T}(\mathcal{A}_i, \{u_t^{-i}\}, \Pi_i^{DAC}) = \widetilde{\mathcal{O}}(N\sqrt{T})$.*

**Remark F.1.** *The notation $\widetilde{\mathcal{O}}(\cdot)$ in Theorem 3.4 hides polynomial factors in $W, \bar{\gamma}^{-1}, \bar{\kappa}, \max_j \|B_j\|, G, d$, and only polylogarithmic factors in $T$ and $N$.*

The proof of the result is based on the regret decomposition that we outline in Section F.1. We start by making the following remark regarding the "burn-in" regret:

**Remark F.2.** *Under Assumption 1-(ii), the 'burn-in' regret $\mathrm{Reg}_i^{1:H}(\mathcal{A}_i, \{u_t^{-i}\}, \Pi_i^{DAC})$ can be bounded by $2H\beta D^2$ which only scales polylogarithmically in $T$ and can scale with $N^2$ in the worst case. This worst-case dependence can be offset by considering a sufficiently large $T$ . If the cost function is uniformly bounded by a constant $C$, then the bound becomes $2HC$, independently of $N$ .*

We now proceed to develop the main overview of the proof:

---

[3]For readability, here and throughout, we use $\widetilde{\mathcal{O}}$ to hide polynomial factors in natural problem parameters and (poly)logarithmic factors in $T$ and $N$. We state the exact dependencies in the proofs of each result.

## F.1 REGRET DECOMPOSITION AND PROOF OVERVIEW

Define the regret from time step $H$ to $T$ as follows:

$$\text{Reg}_i^{H:T}(\mathcal{A}_i, \mathcal{A}_{-i}, \Pi_i^{\text{DAC}}) := \sum_{t=H}^{T} c_t^i(x_t, u_t^i) - \min_{M_{i,\star} \in \mathcal{M}_i} \sum_{t=H}^{T} c_t^i(x_t^{K_i}(M_{i,\star}, M_{-i,t}), u_t^{i,K_i}(M_{i,\star}, M_{-i,t})).$$
(51)

In the rest of this proof we use the shorthand notation $\text{Reg}_i^{H:T}$ for $\text{Reg}_i^{H:T}(\mathcal{A}_i, \mathcal{A}_{-i}, \Pi^{i,\text{DAC}})$. First, it follows from Lemma J.2 that:

$$\text{Reg}_i^T \leq \text{Reg}_i^{0:H} + \text{Reg}_i^{H+1:T}.$$
(52)

Then we decompose the regret from time step $H + 1$ to $T$ as follows:

$$\text{Reg}_i^{H+1:T} = \sum_{t=H+1}^{T} c_t^i(x_t, u_t^i) - \min_{M_{i,\star} \in \mathcal{M}_i} \sum_{t=H+1}^{T} c_t^i(x_t^{K_i}(M_{i,\star}, M_{-i,t}), u_t^{i,K_i}(M_{i,\star}, M_{-i,t}))$$
(53)

$$= \underbrace{\sum_{t=H+1}^{T} (c_t^i(x_t, u_t^i) - l_t^i(M_{i,t-H-1:t}))}_{\text{Counterfactual state and action deviation error}}$$
(54)

$$+ \underbrace{\sum_{t=H+1}^{T} l_t^i(M_{i,t-H-1:t}) - \min_{M_{i,\star} \in \mathcal{M}_i} \sum_{t=H+1}^{T} l_t^i(M_{i,\star})}_{\text{Online gradient descent with memory regret}}$$
(55)

$$+ \underbrace{\min_{M_{i,\star} \in \mathcal{M}_i} \sum_{t=H+1}^{T} l_t^i(M_{i,\star}) - \min_{M_{i,\star} \in \mathcal{M}_i} \sum_{t=H+1}^{T} c_t^i(x_t^{K_i}(M_{i,\star}, M_{-i,t}), u_t^{i,K_i}(M_{i,\star}, M_{-i,t}))}_{\text{Counterfactual state and action deviation optimality error}}.$$
(56)

We conclude the proof of Theorem 3.4 by collecting the upper bounds of each one of the terms established in sections F.2 (see (62) with the choice $H \geq \frac{\log N^2 \sqrt{T}}{\bar{\gamma}}$) and F.3 (see (63) and (67)) below. In conclusion, we obtain

$$\text{Reg}_i^{H+1:T} = \tilde{\mathcal{O}}(N\sqrt{T}),$$
(57)

where $\tilde{O}$ hides polylogarithmic factors in $N$ and polynomial factors in all other problem parameters but $N$. Note that we pick $H \geq \frac{\log N^2 \sqrt{T}}{\bar{\gamma}} + \frac{\log 2\bar{\kappa}}{\bar{\gamma}} = \frac{\log 2\bar{\kappa} N^2 \sqrt{T}}{\bar{\gamma}}$ by combining the two conditions on the horizon length obtained in section F.2 and in Proposition D.3-(viii).

## F.2 COUNTERFACTUAL STATE AND ACTION DEVIATION ERROR

In this section, we upper bound the first and last error terms in the regret decomposition in (53), namely the error terms due to the difference between the realized incurred costs and the costs corresponding to the counterfactual states and actions.

For $t \geq H + 1$, each term in the first error sum term can be upper bounded as follows:

$$|c_t^i(x_t, u_t^i) - l_t^i(M_{i,t-H-1:t})|$$
$$= |c_t^i(x_t^K(M_{0:t-1}), u_t^{i,K}(M_{0:t})) - c_t^i(y_t^K(M_{t-1-H:t-1}), v_t^{i,K}(M_{t-1-H:t}))|$$
$$\leq GD(\|x_t^K(M_{0:t-1}) - y_t^K(M_{t-1-H:t-1})\| + \|u_t^{i,K}(M_{0:t}) - v_t^{i,K}(M_{t-1-H:t})\|)$$
$$\leq 2GD^2(1 - \bar{\gamma})^H,$$
(58)

where the first inequality stems from using Assumption 1-(ii) together with Proposition D.3 and the second inequality follows from using Proposition D.3-(viii), Eq. (30). Note that the constant $D$ is defined in (31).

Summing up the above inequality for $H + 1 \leq t \leq T$, we obtain

$$\sum_{t=H+1}^{T} (c_t^i(x_t, u_t^i) - l_t^i(M_{i,t-H-1:t})) \leq 2GD^2(T-H)(1-\bar{\gamma})^H . \quad (59)$$

The last counterfactual error term in the regret decomposition in (53) can be upper bounded the exact same way as in (59). Indeed pick a policy parameterization

$$\tilde{M}_{i,\star} \in \operatorname*{argmin}_{\tilde{M}_{i,\star} \in \mathcal{M}_i} \sum_{t=H+1}^{T} c_t^i(x_t^{K_i}(M_{i,\star}, M_{-i,t}), u_t^{i,K_i}(M_{i,\star}, M_{-i,t})) . \quad (60)$$

Then we can write

$$\min_{M_{i,\star} \in \mathcal{M}_i} \sum_{t=H+1}^{T} l_t^i(M_{i,\star}) - \min_{M_{i,\star} \in \mathcal{M}_i} \sum_{t=H+1}^{T} c_t^i(x_t^{K_i}(M_{i,\star}, M_{-i,t}), u_t^{i,K_i}(M_{i,\star}, M_{-i,t}))$$

$$= \min_{M_{i,\star} \in \mathcal{M}_i} \sum_{t=H+1}^{T} l_t^i(M_{i,\star}) - \sum_{t=H+1}^{T} c_t^i(x_t^{K_i}(\tilde{M}_{i,\star}, M_{-i,t}), u_t^{i,K_i}(\tilde{M}_{i,\star}, M_{-i,t}))$$

$$\leq \sum_{t=H+1}^{T} (l_t^i(\tilde{M}_{i,\star}) - c_t^i(x_t^{K_i}(\tilde{M}_{i,\star}, M_{-i,t}), u_t^{i,K_i}(M_{i,\star}, M_{-i,t}))) , \quad (61)$$

and the last sum is of the exact same form as the one we upper bounded in (59). Observe that Assumption 1-(ii) together with Proposition D.3 can be used again upon noticing that the results of Proposition D.3 are also valid when fixing player $i$'s matrix to be $\tilde{M}_{i,\star} \in \mathcal{M}_i$, it suffices to replace $M_{i,t-1-H:t}$ by the constant matrix $\tilde{M}_{i,\star}$ everywhere in the proof of Proposition D.3 and using the fact that $\tilde{M}_{i,\star} \in \mathcal{M}_i$, the proof remains unchanged.

In conclusion of this section, we have shown that

$$\sum_{t=H+1}^{T} (c_t^i(x_t, u_t^i) - l_t^i(M_{i,t-H-1:t}))$$

$$+ \min_{M_{i,\star} \in \mathcal{M}_i} \sum_{t=H+1}^{T} l_t^i(M_{i,\star}) - \min_{M_{i,\star} \in \mathcal{M}_i} \sum_{t=H+1}^{T} c_t^i(x_t^{K_i}(M_{i,\star}, M_{-i,t}), u_t^{i,K_i}(M_{i,\star}, M_{-i,t}))$$

$$\leq 4GD^2(T-H)(1-\bar{\gamma})^H . \quad (62)$$

Now, note from the definition of $D$ in (31) that $D = \mathcal{O}(N)$. Therefore, the above error term scales in $T$ and $N$ as $\mathcal{O}(N^2 T(1-\bar{\gamma})^H)$. Choosing $H \geq \frac{\log N^2 \sqrt{T}}{\bar{\gamma}}$ guarantees that the error term is of the order $\tilde{O}(\sqrt{T})$, where $\tilde{O}$ hides polylogarithmic factors in $N$ and polynomial factors in all other problem parameters but $N$.

### F.3 ONLINE GRADIENT DESCENT WITH MEMORY REGRET BOUND

Applying Theorem J.1 of Appendix J.1 in Anava et al. (2015) gives:

$$\sum_{t=H+1}^{T} l_t^i(M_{i,t-H-1:t}) - \min_{M_{i,\star} \in \mathcal{M}_i} \sum_{t=H+1}^{T} l_t^i(M_{i,\star}) \leq \frac{D_0^2}{\eta} + (G_0^2 + LH^2 G_0)\eta T . \quad (63)$$

It remains to check assumptions 1 to 3 of Theorem J.1 and specify the values of the diameter bound $D_0$, the coordinate-wise Lipschitz constant $L$ and the gradient bound constant $G_0$.

As for the diameter boundedness, we can set $D_0 = 4\sqrt{2}\bar{\kappa}^2/\bar{\gamma}$. This is because for any $M_1, M_2 \in \mathcal{M}_i$ (for any $i \in [N]$), we have

$$\|M_1 - M_2\| \leq \sqrt{2} \left( \sum_{p=1}^{H} \|M_1^{[p-1]}\| + \|M_2^{[p-1]}\| \right) \leq 4\sqrt{2} \sum_{p=1}^{H} \bar{\kappa}^2 (1-\bar{\gamma})^p \leq 4\sqrt{2}\bar{\kappa}^2/\bar{\gamma} . \quad (64)$$

Coordinatewise loss lipschitzness and gradient loss boundedness are respectively established in subsections F.3.1 (Lemma F.3) and F.3.2 (Lemma F.4) below.

Now in order to set the stepsize in the regret bound (63) above, we focus on optimizing the dependence on the time horizon $T$ as well as the total number $N$ of agents. Observe now from Lemma F.3 and Lemma (F.4) together with the definition of $D$ in (31) that

$$L = \mathcal{O}(D) = \mathcal{O}(N), \quad G_0 = \mathcal{O}(D) = \mathcal{O}(N), \tag{65}$$

where the big $\mathcal{O}(\cdot)$ notation hides problem parameters that are independent of $N$. Hence the regret bound in (63) is of the order

$$\mathcal{O}\left(\frac{1}{\eta} + N^2 \eta T\right), \tag{66}$$

where again the big $\mathcal{O}(\cdot)$ notation hides problem parameters that are independent of $N$. Therefore we set $\eta = \Theta(1/(N\sqrt{T}))$ and the final online gradient descent regret bound we obtain scales as

$$\mathcal{O}(N\sqrt{T}), \tag{67}$$

which concludes the proof. Note here that we have optimized the stepsize to obtain the best dependence on both the time horizon $T$ and notably the number $N$ of agents. In particular, using the standard optimal upper bound giving the smallest regret bound (without focusing on any parameter in particular) would result in a worse dependence on the number of agents.

### F.3.1 COORDINATE-WISE LOSS LIPSCHITZNESS

**Lemma F.3** (Coordinate-wise loss lipschitzness). *For any agent $i \in [N]$, let $(M_{i,t-1-H}, \cdots, M_{i,t-k}, \cdots, M_{i,t})$ and $(M_{i,t-1-H}, \cdots, \tilde{M}_{i,t-k}, \cdots, M_{i,t})$ be two policy parameter sequences for agent $i$ differing only in time step $t-k$ for $k \in 0, \cdots, H$ with $M_{i,t-k}$ replaced by $\tilde{M}_{i,t-k}$. Suppose that the policy parameters of other agents but $i$ are given by the same sequence $M_{-i,t-1-H:t}$ (i.e. the same for both joint policies, the difference is only in player $i$'s policy). Then we have for every $t \geq H+1$,*

$$|l_t^i(M_{i,t-1-H}, \cdots, M_{i,t-k}, \cdots, M_{i,t}) - l_t^i(M_{i,t-1-H}, \cdots, \tilde{M}_{i,t-k}, \cdots, M_{i,t})|$$

$$\leq L \sum_{p=1}^{H} \|M_{i,t-k}^{[p]} - \tilde{M}_{i,t-k}^{[p]}\|, \tag{68}$$

*where $L = 2GDW\bar{\kappa}^2 \max_{j=1,\cdots,N} \|B_j\|$ and $G, D$ are respectively defined in Assumption 1-(ii) and (31).*

*Proof.* The proof follows a similar approach to that of (Agarwal et al., 2019, Lemma 5.6). However, we provide a complete proof of this result since our multi-agent setting is different and induces a different state evolution given that all the agents run DAC-$i$ policies.

We introduce a few convenient notation for the rest of this proof. Define for every $t \geq H$,

$$y_t^K := y_t^K(M_{t-1-H}, \cdots, M_{t-k}, \cdots, M_{t-1}),$$

$$\tilde{y}_t^K := y_t^K(M_{t-1-H}, \cdots, \tilde{M}_{t-k}, \cdots, M_{t-1}),$$

$$v_t^{i,K} := v_t^{i,K}(M_{t-1-H:t}) = -K_i y_t^K + \sum_{p=1}^{H} M_{i,t}^{[p-1]} w_{t-p},$$

$$\tilde{v}_t^{i,K} := v_t^{i,K}(M_{t-1-H}, \cdots, \tilde{M}_{t-k}, \cdots, M_t)$$

$$= -K_i \tilde{y}_t^K + \sum_{p=1}^{H} (\tilde{M}_{i,t}^{[p-1]} - M_{i,t}^{[p-1]}) w_{t-p} \mathbf{1}_{k=0} + \sum_{p=1}^{H} M_{i,t}^{[p-1]} w_{t-p}. \tag{69}$$

Using this notation, we have

$$|l_t^i(M_{i,t-1-H}, \cdots, M_{i,t-k}, \cdots, M_{i,t}) - M_{i,t-1-H}, \cdots, \tilde{M}_{i,t-k}, \cdots, M_{i,t}|$$

$$= |c_t^i(y_t^K, v_t^{i,K}) - c_t^i(\tilde{y}_t^K, \tilde{v}_t^{i,K})|$$

$$\leq |c_t^i(y_t^K, v_t^{i,K}) - c_t^i(\tilde{y}_t^K, v_t^{i,K})| + |c_t^i(\tilde{y}_t^K, v_t^{i,K}) - c_t^i(\tilde{y}_t^K, \tilde{v}_t^{i,K})|$$

$$\leq GD(\|y_t^K - \tilde{y}_t^K\| + \|v_t^{i,K} - \tilde{v}_t^{i,K}\|), \tag{70}$$

where the last step uses Assumption 1-(ii).

Recall that we can write the counterfactual states $y_t^K, \tilde{y}_t^K$ using the transition matrix (see (37)):

$$y_t^K := \sum_{l=0}^{2H} \bar{\Psi}_{t-1,l}^H(M_{t-1-H:t-1})w_{t-1-l}, \tag{71}$$

$$\tilde{y}_t^K := \sum_{l=0}^{2H} \bar{\Psi}_{t-1,l}^H(M_{t-1-H}, \cdots, \tilde{M}_{t-k}, \cdots, M_{t-1})w_{t-1-l}. \tag{72}$$

Note for clarification that in the notation above $\tilde{M}_{t-k}$ is identical to $M_{t-k}$ except for its $i$-th matrix element, i.e. $\tilde{M}_{j,t-k} = M_{j,t-k}$ for every $j \neq i$. Therefore, using the definition of the state transfer matrix in (12) the difference of counterfactual states can be expressed as follows:

$$y_t^K - \tilde{y}_t^K = \sum_{l=0}^{2H} \bar{A}_K^k B_i(M_{i,t-k}^{[l-k-1]} - \tilde{M}_{i,t-k}^{[l-k-1]})\mathbf{1}_{l-k\in[1,H]}w_{t-l}. \tag{73}$$

We can now bound the difference of counterfactual states using $(\bar{\kappa}, \bar{\gamma})$-strong stability and boundedness of the disturbance sequence by $W$:

$$\|y_t^K - \tilde{y}_t^K\| \leq W\bar{\kappa}(1-\bar{\gamma})^k \cdot \|B_i\| \sum_{p=1}^{H} \|M_{i,t-k}^{[p-1]} - \tilde{M}_{i,t-k}^{[p-1]}\|, \tag{74}$$

where the bound uses a re-indexation of the sum in (73) with $p = l - k$. As for the difference of counterfactual actions, it stems from (69) that:

$$\tilde{v}_t^{i,K} - v_t^{i,K} = K_i(y_t^K - \tilde{y}_t^K)\mathbf{1}_{k\in[1:H]} + \sum_{p=1}^{H}(\tilde{M}_{i,t}^{[p-1]} - M_{i,t}^{[p-1]})w_{t-p}\mathbf{1}_{k=0}. \tag{75}$$

As a consequence, we have

$$\|\tilde{v}_t^{i,K} - v_t^{i,K}\| \leq \|K_i\| \cdot \|y_t^K - \tilde{y}_t^K\|\mathbf{1}_{k\in[1:H]} + W\sum_{p=1}^{H}\|\tilde{M}_{i,t}^{[p-1]} - M_{i,t}^{[p-1]}\|\mathbf{1}_{k=0} \tag{76}$$

$$\leq W\bar{\kappa}^2 \cdot \max_{j=1,\cdots,N}\|B_j\| \sum_{p=1}^{H} \|M_{i,t-k}^{[p-1]} - \tilde{M}_{i,t-k}^{[p-1]}\|, \tag{77}$$

where the last inequality stems from using the bound (74) together with the simplifying assumption that $\bar{\kappa}^2 \max_{j=1,\cdots,N}\|B_j\| \geq 1$ (without any loss of generality).

Combining (70) with the bounds (74) and (76) yields the desired inequality and concludes the proof:

$$|l_t^i(M_{i,t-1-H}, \cdots, M_{i,t-k}, \cdots, M_{i,t}) - M_{i,t-1-H}, \cdots, \tilde{M}_{i,t-k}, \cdots, M_{i,t}|$$

$$\leq 2GDW\bar{\kappa}^2 \max_{j=1,\cdots,N}\|B_j\| \sum_{p=1}^{H} \|M_{i,t-k}^{[p]} - \tilde{M}_{i,t-k}^{[p]}\|. \tag{78}$$

$\square$

### F.3.2 GRADIENT LOSS BOUNDEDNESS

**Lemma F.4.** *Let* $M = (M_i, M_{-i})$ *be s.t.* $\|M_i^{[p]}\| \leq \tau(1-\bar{\gamma})^p$ *for* $p \in \{0, \cdots, H-1\}$ *and for every* $i \in [N]$. *Then we have for any* $i \in [N]$,

$$\|\nabla_{M_i} l_t^i(M_i)\|_F \leq GD\sqrt{H}dW\left(1 + \frac{2\bar{\kappa}^2 \max_{j=1,\cdots,N}\|B_j\|}{\bar{\gamma}}\right), \tag{79}$$

*where* $G, D$ *are respectively defined in Assumption 1-(ii) and (31) whereas* $d$ *is the dimension of the state vector.*

*Proof.* The proof is similar to that of (Agarwal et al., 2019, Lemma 5.7) and is therefore omitted. $\square$

# G  PROOF OF THEOREM 3.5

Here, we develop the proof of Theorem 3.5, restated here:

**Theorem 3.5.** *Under the setting of Theorem 3.4, replace gradient boundedness in Assumption 1 -(ii) by Assumption 5. Set instead $\eta = \Theta(1/\sqrt{T})$ and $H \geq \log(2\bar{\kappa}N\sqrt{T})/\bar{\gamma}$. Then for any $T \geq H + 1$:*
$\text{Reg}_i^{H+1:T}(\mathcal{A}_i, \{u_t^{-i}\}, \Pi_i^{DAC}) = \tilde{\mathcal{O}}(\sqrt{T})$.

*Proof.* The proof of this refined result follows the same lines as the proof of Theorem 3.4. We indicate here the required modifications to establish the result of Theorem 3.5 using the uniform Lipschitz cost assumption 5 instead of gradient boundedness in Assumption 1 -(ii).

Recall the regret decomposition in (53) in Section F.1. We adapt the bounds in F.2 and F.3 to our new assumption.

- **Counterfactual state and action deviation error.** For this term, it suffices to observe that under the uniform Lipschitz cost assumption 5, we can replace GD in (58) by the uniform Lipschitz constant $\bar{L}$ (which is supposed to be independent of $N$). The rest of the proof is unchanged and the resulting counterfactual state-action deviation error is of the order:

$$\mathcal{O}(2\bar{L}D(1 - \bar{\gamma})^H), \tag{80}$$

where we recall that $D$ is defined in (31) and $D = \mathcal{O}(N)$.

- **Online gradient descent with memory regret bound.** We recall here from (63) that this regret term is bounded by

$$\frac{D_0^2}{\eta} + (G_0^2 + LH^2G_0)\eta T. \tag{81}$$

It suffices to reevaluate the coordinate-wise Lipschitzness constant $L$ and the gradient bound $G_0$ made explicit in Lemma F.3 and Lemma F.4 respectively. We now make the two following observations regarding these two constants and their dependence on the number $N$ of agents:

(i) Again using Assumption 5, we can replace $GD$ by $\bar{L}$ in (70) in the proof of Lemma F.3, the rest of the proof is unchanged. The result is that the coordinate-wise Lipschitz constant $L$ of Lemma F.3 becomes $L = 2\bar{L}W\bar{\kappa}^2 \max_{j=1,\cdots,N} \|B_j\|$ and therefore independent of the number of agents.

(ii) Similarly, the constant $GD$ in the gradient bound of Lemma F.4 can be replaced by $\bar{L}$ (which is independent of $N$), resulting in a gradient bound which is independent of the number of agents.

Combining the above insights, it suffices to choose $H \geq \log(2\bar{\kappa}N\sqrt{T})/\bar{\gamma}$ in (80) and $\eta = \Theta(1/\sqrt{T})$ in (81) to obtain the desired result for $T \geq H + 1$:

$$\text{Reg}_i^{H+1:T}(\mathcal{A}_i, \mathcal{A}_{-i}, \Pi^{i,\text{DAC}}) = \tilde{\mathcal{O}}\left(\sqrt{T}\right), \tag{82}$$

where $\tilde{\mathcal{O}}(\cdot)$ hides polynomial factors in $W, \bar{\gamma}^{-1}, \bar{\kappa}, \max_j \|B_j\|, G, d$ and only polylogarithmic factors in $T$ and $N$. This concludes the proof. □

# H  PROOFS OF REGRET LOWER BOUNDS

In this section, we develop the proof of Theorem 3.3, which we restate here:

**Theorem 3.3.** *For any agent $i \in [N]$, there exists an instance of* (LDS) *and cost functions $\{c_t^i\}$ such that, for any algorithm $\mathcal{A}_i$ and sequence $\{u_t^{-i}\}$, and any $T \geq 1$: $\text{Reg}_i^T(\mathcal{A}_i, \{u_t^{-i}\}, \Pi_i^{\text{lin}}) = \Omega(\sqrt{T})$.*

## H.1  PROOF OF THEOREM 3.3

Fix agent $i \in [N]$. To prove the theorem, we specify an LDS and a (randomized) sequence of cost functions $\{c_t^i\}$, and we will prove that the lower bound holds in expectation. By the probabilistic method, this implies the existence of a deterministic sequence of cost functions where the lower bound holds with probability 1. We begin by specifying the LDS instance and cost function constructions:

**Construction of LDS instance.** We specify a scalar-valued instance of (LDS), where all $A, B_j, w_t \in \mathbb{R}$. Specifically, we use the following settings which implies a state evolution of

$$
\begin{cases}
A = 0 \\
B_i = \frac{1}{2} \\
B_j = 0 \text{ for all } j \neq i \in [N] \\
w_t = 0 \text{ for all } t \in [T] \\
x_0 \in (0, 1]
\end{cases}
\implies \quad x_{t+1} = \frac{1}{2} u_t^i \quad \text{for all } t \geq 0.
\tag{83}
$$

In other words, due to the construction, the state $x_t$ is driven only by the control of the $i$'th agent. Observe also that for the scalar LDS $(0, 1/2)$ as specified in (83), we have by Definition 2.1 that a linear controller $K \in \mathbb{R}$ is $(\kappa, \gamma)$-strongly stable when $|K| \leq \kappa$ and $|K/2| \leq 1 - \gamma$, for $\gamma \in (0, 1)$.

**Construction of Agent $i$ cost functions.** We now construct a hard sequence of randomized cost functions for agent $i$, which are roughly inspired by lower bound constructions in (adversarial) online linear optimization settings (see e.g., Arora et al. (2012, Section 4)). Specifically, for all times $t \geq 0$ and $x, u \in \mathbb{R}$, let $c_t^i$ be given by

$$
c_t^i(x, u) = \left\langle \begin{pmatrix} u \\ 1 - u \end{pmatrix}, \begin{pmatrix} b_t \\ 1/2 \end{pmatrix} \right\rangle = u\left(b_t - \tfrac{1}{2}\right) + \tfrac{1}{2}
\tag{84}
$$

for all $x, u \in \mathbb{R}$, where each $b_t$ is an independent Bern$(1/2)$ random variable (i.e., each $b_t = 0$ with probability half and $b_t = 1$ with probability half).

Under the LDS of (83) and cost functions of (84), in show a expected lower bound on the regret of agent $i$, we establish bounds on (i) the expected cost of agent $i$, and (ii) the expected counterfactual cost of the best fixed linear controller in hindsight.

**Expected cost of agent $i$.** Under the cost functions of (84), it is straightforward to compute the total expected cost of agent $i$:

**Proposition H.1.** *Let $\{u_t^i\}$ be the sequence of controls of agent $i$ using any algorithm and with respect to the cost sequence $\{c_t^i\}$ from (84). Let $\{x_t\}$ be the resulting state evolution as in (83). Then over the randomness of $\{b_t\}$,*

$$
\mathbf{E}\left[ \sum_{t=0}^{T} c_t^i(x_t, u_t^i) \right] = \frac{T}{2}.
\tag{85}
$$

*Proof.* For any fixed $t \geq 0$, and any $x, u \in \mathbb{R}$, observe under the randomness of $b_t$ that

$$
\mathbf{E}\left[ c_t^i(x, u) \right] = \mathbf{E}\left[ u(b_t - \tfrac{1}{2}) + \tfrac{1}{2} \right] = \frac{1}{2}.
\tag{86}
$$

Then by linearity of expectation we have $\mathbf{E}[\sum_{t=1}^{T} c_t^i(x_t, u_t^i)] = \frac{T}{2}$. $\qquad\square$

**Expected cost of comparator.** Let $\mathcal{K}_i \subseteq \mathbb{R}$ be the set of strongly stable linear controllers. For a fixed $K \in \mathcal{K}_i$, let (by slight abuse of notation) $\widetilde{x}_t^K$ denote the counterfactual state evolution on the LDS in (83) using the fixed linear controller with (counterfactual) control sequence $\widetilde{u}_t^{i,K} = K\widetilde{x}_t^K$ at all times $t \geq 0$. Then for each $k \in \mathcal{K}$, let $\Phi(k)$ be the random variable

$$
\Phi(K) := \sum_{t=1}^{T} c_t^i(\widetilde{x}_t^K, \widetilde{u}_t^{i,K}) = \sum_{t=1}^{T} \left( K\widetilde{x}_t^K \left(b_t - \tfrac{1}{2}\right) + \tfrac{1}{2} \right).
\tag{87}
$$

Using a fixed linear controller $K$, and under the assumption that $x_0 \in (0, 1]$ observe from (83) that the counterfactual state evolution of $\widetilde{x}_t^K$ can be written as

$$
\widetilde{x}_t^K = \tfrac{1}{2} K \widetilde{x}_{t-1}^K = \left(\tfrac{1}{2} K\right)^t x_0.
$$

It follows that

$$
\Phi(K) := \sum_{t=1}^{T} \left( \frac{K^{t+1}}{2^t} \cdot x_0 \left(b_t - \frac{1}{2}\right) + \frac{1}{2} \right).
$$

Letting $\mathcal{K}_+ = \mathcal{K}_i \cap [0,1] \subset \mathcal{K}_i$, observe that (with probability 1)

$$\min_{K \in \mathcal{K}_i} \Phi(K) \ \leq \ \min_{K \in \mathcal{K}_+} \Phi(K) \ \leq \ \Phi(0) \ = \ \sum_{t=1}^{T} \frac{1}{2} \ = \ \frac{T}{2} \ . \tag{88}$$

Moreover, for $K \in [0,1]$ and $x_0 \in (0,1]$, and using the fact that $b_t \in \{0,1\}$ by definition, observe that we can bound (with probability 1)

$$\left| \frac{K^{t+1}}{2^t} \cdot x_0 (b_t - \tfrac{1}{2}) + \frac{1}{2} \right| \ \leq \ 1 \tag{89}$$

for all $t \in [T]$. Finally, for $x_0 \in (0,1]$, observe that the image of $\Phi$ over $\mathcal{K}_+$ is non-singleton.

**Tail bounds on cost of comparator.** It remains to derive an upper bound on the expected cost of the optimal comparator of the form $\mathbf{E}[\min_{K \in \mathcal{K}} \Phi(K)] \leq \frac{T}{2} - \Omega(\sqrt{T})$. For this, we will establish under the randomness of $\{b_t\}$ that the random variable $\min_{K \in \mathcal{K}_+} \Phi(K)$ is small with sufficiently large probability. Fix $K$ and define

$$\psi_t(b_t, K) \ = \ \frac{K^{t+1}}{2^t} \cdot x_0 (b_t - \tfrac{1}{2}) + \frac{1}{2} \ .$$

It follows that we can write

$$\Phi(K) \ = \ \sum_{t=1}^{T} \psi_t(b_t, K) \ ,$$

which by (89) means $\Phi(K)$ is the sum of $T$ independent and bounded random variables.

We now leverage the following lower bound on the tail of a sum of bounded random variables:

**Lemma H.2** (Zhang & Zhou (2020), Corollary 2). *Let $Z = Z_1 + \cdots + Z_T$ such that $\mathbf{E}[Z_t] = 0$ and $|Z_t| \leq C$ for all $t \in [T]$ and some absolute constant $C > 0$. Then there exist absolute constants $0 < a < 1$ and $p > 0$ such that*

$$\Pr\left( Z \leq -a \cdot \sqrt{T} \right) \ \geq \ p \ .$$

By centering $\psi'_t = \psi_t(b_t, K) - \frac{1}{2}$, we have $\mathbf{E}[\psi'_t] = 0$ and each $|\psi'_t|$ bounded (which follows from expression (89)). Then applying Lemma H.2 to the sum $\sum_{t=1}^{T} \psi'_t$, we conclude that there exist absolute constants $a, p > 0$ such that

$$\Pr\left( \Phi(K) \leq \frac{T}{2} - a \cdot \sqrt{T} \right) \ \geq \ p \ . \tag{90}$$

Moreover, as $\phi(K) \leq \frac{T}{2} - a \cdot \sqrt{T} \implies \min_{K \in \mathcal{K}_+} \Phi(K) \leq \frac{T}{2} - a \cdot \sqrt{T}$, we further have

$$\Pr\left( \min_{K \in \mathcal{K}_+} \Phi(K) \leq \frac{T}{2} - a \cdot \sqrt{T} \right) \ \geq \ \Pr\left( \Phi(K) \leq \frac{T}{2} - a \cdot \sqrt{T} \right) \ \geq \ p \ . \tag{91}$$

Finally, since by expression (88) we have $\min_{K \in \mathcal{K}} \Phi(K) \leq \frac{T}{2}$ with probability 1, it follows that

$$\mathbf{E}\left[ \min_{K \in \mathcal{K}} \Phi(K) \right] \ \leq \ \mathbf{E}\left[ \min_{K \in \mathcal{K}_+} \Phi(K) \right] \ \leq \ -pa\sqrt{T} + \frac{T}{2} \tag{92}$$

Combining expressions (85) and (92), we conclude that over the randomness of $\{b_t\}$

$$\mathbf{E}\left[ \sum_{t=1}^{T} c_t^i(x_t, u_t^i) \ - \ \min_{k \in \mathcal{K}} \Phi(K) \right] \ \geq \ \frac{T}{2} - \left( pa\sqrt{T} + \frac{T}{2} \right) \ = \ pa\sqrt{T} \ .$$

Thus in expectation over the sequence $\{b_t\}$, $\mathrm{Reg}_T^i$ is at least $\Omega(\sqrt{T})$, which implies that for some realization of $\{b_t\}$, the same lower bound holds. $\qquad\square$

### H.2 Lower Bound Against DAC Policies

In this section, we extend the regret lower bound against linear policies from Theorem 3.3 to also hold for the DAC comparator class. Note that as the class of DAC policies contains the class of linear policies, a regret lower bound against linear policies does not immediately imply a lower bound against DAC policies. However, by slightly modifying the hard LDS construction from (83), and under the assumption that the linear controller component of the DAC policy is chosen adversarially, then a similar lower bound can be established following the proof of Theorem 3.3. Formally:

**Theorem H.3.** *Fix $i \in [N]$, and let $\Pi^{i,\mathrm{DAC}}$ denote the set of DAC policies for agent $i$. Then there exists an instance of* (LDS) *and cost functions $\{c_t^i\}$ such that, for any algorithm $\mathcal{A}_i$ and control sequence $\{u_t^{-i}\}$, and any $T \geq 1$, when the linear DAC component $K_i$ is chosen adversarially:*

$$\mathrm{Reg}_T^i(\mathcal{A}_i, \{u_t^{-i}\}, \Pi_i^{\mathrm{DAC}}) = \Omega\left(\sqrt{T}\right) .$$

*Proof.* Similar to the proof of Theorem 3.3, we specify a scaler-value instance of (LDS). Now we use settings with corresponding state evolution as follows:

$$\begin{cases} A = 0 \\ B_i = 1 \\ B_j = 0 \text{ for all } j \neq i \in [N] \\ w_t = 1 \text{ for all } t \in [T] \\ x_0 = 0 \end{cases} \implies x_{t+1} = u_t^i + 1 \quad \text{for all } t \geq 0. \tag{93}$$

We use the same construction of costs $\{c_t^i\}$ from expression (84) in the proof of Theorem 3.3. By Proposition H.1, this implies

$$\mathbf{E}\left[\sum_{t=0}^{T} c_t^i(x_t, u_t^i)\right] = \frac{T}{2} .$$

Next, we control the expected (counterfactual) cost of the optimal comparator policy. For this, let $\mathcal{M}_+$ denote the subset of DAC parameters in $\mathcal{M}_i$ such that $M_i^{[p]} = M_i^{[h]}$ for all $p, h \in [H]$. In other words, for a DAC policy parameter in $\mathcal{M}_+$, all $H$ parameter values are equal. We denote such a policy in $\mathcal{M}_+$ by a scalar $M \in \mathbb{R}$. As clearly $\mathcal{M}_+ \subset \mathcal{M}_i$, it follows that

$$\min_{M \in \mathcal{M}_i} \sum_{t=1}^{T} c_t^i(\widetilde{x}_t^M, \widetilde{u}_t^{i,M}) \leq \min_{M \in \mathcal{M}_+} \sum_{t=1}^{T} c_t^i(\widetilde{x}_t^M, \widetilde{u}_t^{i,M})$$

where (by slight abuse of notation) $\widetilde{x}_t^M$ and $\widetilde{u}_t^{i,M}$ denote counterfactual state and control sequences under a fixed comparator policy parameter $M$. Thus for the purposes of a regret lower bound, it suffices to derive an upper bound on the optimal comparator cost with respect to the class $\mathcal{M}_+$.

For this, using similar notation as in the proof of Theorem 3.3, for $M \in \mathcal{M}_+$, define $\Phi(M)$ as

$$\Phi(M) = \sum_{t=1}^{T} c_t^i(\widetilde{x}_t^M, \widetilde{u}_t^{i,M}) .$$

Under an adversarial choice of linear controller $K_i = 0$, and using the LDS settings of (93), it follows by definition of DAC policies in $\mathcal{M}_+$ that

$$\widetilde{u}_t^{i,M} = Kx_{t-1} + \sum_{p=1}^{H} M^{[p-1]} w_{t-p} = HM . \tag{94}$$

Then using the definition of $c_t^i$ from expression (84), we have

$$\Phi(M) = \sum_{t=1}^{T} c_t^i(\widetilde{x}_t^M, \widetilde{u}_t^{i,M}) = \sum_{t=1}^{T} HM(b_t - \tfrac{1}{2}) + \tfrac{1}{2} .$$

Then clearly $\Phi(0) = \frac{T}{2}$, and thus also

$$\min_{M \in \mathcal{M}_+} \Phi(M) \leq \Phi(0) = \frac{T}{2}$$

Now using the fact that, under the randomness of $\{b_t\}$, for each $M \in \mathcal{M}$ $\Phi(M)$ is the sum of $T$, independent random variables bounded by $H \geq 1$, we apply the tail bound of Lemma H.2 (as in the proof of Theorem 3.3) to find

$$\Pr\left(\Phi(M) \leq \frac{T}{2} - a\sqrt{T}\right) \geq p$$

for absolute constants $a, p > 0$. Then following identical calculations as in expressions (91) and (92), we conclude that

$$\mathbf{E}\left[\sum_{t=1}^{T} c_t^i(x_t, u_t^i) - \min_{M \in \mathcal{M}_+} \Phi(M)\right] \geq pa\sqrt{T} \,,$$

which by the probabilistic method implies the lower bound of the theorem statement. $\qquad \square$

## I  PROOF OF THEOREM 4.1

We first recall the theorem:

**Theorem 4.1.** *Let Assumptions 1, 2, 4, 6 and 7 hold. Then if each agent $i \in [N]$ runs Algorithm 1 for $T$ steps with constant stepsize $\eta = 1/L$ (where $L$ is the smoothness constant in Lemma I.5), then*

$$\frac{1}{T}\sum_{t=1}^{T}\left(\mathrm{EQGAP}^{(t)}(M_t)\right)^2 = \mathcal{O}\left(\frac{\ell_1(M_1) - c_{\inf}}{T} + \frac{1}{T}\sum_{t=1}^{T}\Delta_{c_t} + \frac{1}{T}\sum_{t=1}^{T}\|w_{t+1} - w_t\|\right), \quad (8)$$

*where $\Delta_{c_t} := \max_{\|x\|, \|u\| \leq D}\{c_{t+1}(x, u) - c_t(x, u)\}$ for every $t$, the $\mathcal{O}(\cdot)$ notation only hides polynomial dependence in the problem parameters $N, H, W, \bar{\kappa}, \bar{\gamma}^{-1}, \max_i \|B_i\|$ and $D$ depends polynomially on the same constants. All the constants are made explicit in the appendix.*

**Outline of the proof.** The proof of Theorem 4.1 can be divided into three main steps that are recorded in the following three propositions:

1. Proposition I.1 upperbounds the sum of equilibrium gaps by the sum of policy parameter deviations across time and players.

2. Proposition I.2 upperbounds the latter policy parameter deviations by the sum of loss deviations along time.

3. Finally, Proposition I.3 upperbounds the sum of loss deviations by the initial distance to the infimal cost value, the cost function variability and the sum of disturbance variations.

The proof of Theorem 4.1 follows from combining Proposition I.1 with Propositions I.2 and I.3 by chaining them. The rest of this section I is devoted to proving each one of Propositions I.1, I.3 and I.3 separately.

**Proposition I.1.** *Let Assumption 1 hold. Then for every time horizon $T \geq 1$,*

$$\sum_{t=1}^{T}\left(\mathrm{EQGAP}^{(t)}(M^{(t)})\right)^2 \leq C_{\mathcal{M}}\sum_{t=1}^{T}\sum_{i=1}^{N}\|M_{i,t+1} - M_{i,t}\|^2 \,, \quad (95)$$

*where $C_{\mathcal{M}} := \sum_{i=1}^{N}\left(\frac{diam(\mathcal{M}_i)}{\eta} + GD\right)^2$ and $G, D$ are the constants in Assumption 1 .*

**Proposition I.2.** *Let Assumptions 1, 2 and 7 hold. Then running Algorithm 1 for $T$ steps with step size $\eta = 1/L$ where $L$ is the smoothness constant in Lemma I.5 yields:*

$$\sum_{t=1}^{T}\sum_{i=1}^{N}\|M_{i,t+1} - M_{i,t}\|^2 \leq \eta\sum_{t=1}^{T} l_t(M_t) - l_t(M_{t+1}) \,. \quad (96)$$

**Proposition I.3.** *Let Assumptions 2, 4 hold. For every $T \geq 1$,*

$$\sum_{t=1}^{T} \ell_t(M_t) - \ell_t(M_{t+1}) = \mathcal{O}\left(\ell_1(M_1) - c_{\inf} + \sum_{t=1}^{T} \Delta_{c_t} + \sum_{t=1}^{T} \|w_{t+1} - w_t\|\right), \quad (97)$$

*where $\Delta_{c_t} := \max_{\|x\|,\|u\|\leq D}\{c_{t+1}(x,u) - c_t(x,u)\}$ for every $t$, the $\mathcal{O}(\cdot)$ notation only hides polynomial dependence in the problem parameters $N, H, W, \bar{\kappa}, \bar{\gamma}^{-1}, \max_i \|B_i\|$ and $D$ depends polynomially on the same constants.*

I.1    PROOF OF PROPOSITION I.1

First, recall the following notations of the best response and equilibrium gap for every $i \in [N], t \geq 1$:

$$\mathrm{BR}_i^{(t)}(M_{-i,t}) := \max_{M_i \in \mathcal{M}_i} \ell_t^i(M_t) - \ell_t^i(M_i, M_{-i,t}) \quad (98)$$

$$\text{and} \quad \mathrm{EQGAP}^{(t)}(M_t) := \max_{i \in [N]} \mathrm{BR}_i^{(t)}(M_{-i,t}). \quad (99)$$

Observe in particular that $\mathrm{BR}_i^{(t)}(M_{-i,t}) \geq 0$ (use $M_i = M_{i,t}$). Using the definition of the equilibrium gap, it immediately follows that

$$\sum_{t=1}^{T} \mathrm{EQGAP}^{(t)}(M_t)^2 = \sum_{t=1}^{T} \left(\max_{i \in [N]} \mathrm{BR}_i^{(t)}(M_{-i,t})\right)^2 \leq \sum_{t=1}^{T}\left(\sum_{i=1}^{N} \mathrm{BR}_i^{(t)}(M_{-i,t})\right)^2. \quad (100)$$

We now relate the best response quantities to the deviation of DAC policy parameters via the following proposition whose proof is deferred to section I.4.

**Proposition I.4.** *Let Assumption 1 hold. Then for every $i \in [N], M_i \in \mathcal{M}_i, t \geq 1$, we have*

$$\ell_t^i(M_i, M_{-i,t}) - \ell_t^i(M_t) \geq -\left(\frac{diam(\mathcal{M}_i)}{\eta} + GD\right)\|M_{i,t+1} - M_{i,t}\|, \quad (101)$$

*where $diam(\mathcal{M}_i) = \max_{M,M' \in \mathcal{M}_i}\|M' - M\|$ and $G, D$ are the constants in Assumption 1.*

Invoking Proposition I.4 gives the following inequality

$$0 \leq \mathrm{BR}_i^{(t)}(M_{-i,t}) \leq \left(\frac{diam(\mathcal{M}_i)}{\eta} + GD\right)\|M_{i,t+1} - M_{i,t}\|. \quad (102)$$

Summing up this inequality across all the $N$ players yields:

$$0 \leq \sum_{i=1}^{N} \mathrm{BR}_i^{(t)}(M_{-i,t}) \leq \sum_{i=1}^{N}\left(\frac{diam(\mathcal{M}_i)}{\eta} + GD\right)\|M_{i,t+1} - M_{i,t}\|. \quad (103)$$

Using now the Cauchy-Schwarz inequality on the squared sum of best responses gives

$$\left(\sum_{i=1}^{N} \mathrm{BR}_i^{(t)}(M_{-i,t})\right)^2 \leq \sum_{i=1}^{N}\left(\frac{diam(\mathcal{M}_i)}{\eta} + GD\right)^2 \cdot \sum_{i=1}^{N}\|M_{i,t+1} - M_{i,t}\|^2. \quad (104)$$

Finally, we obtain the desired inequality by summing up the above inequality over the time steps $t = 1, \cdots, T$ and using (100),

$$\sum_{t=1}^{T} \mathrm{EQGAP}^{(t)}(M_t)^2 \leq C_{\mathcal{M}} \sum_{t=1}^{T}\sum_{i=1}^{N}\|M_{i,t+1} - M_{i,t}\|^2, \quad (105)$$

where $C_{\mathcal{M}} = \sum_{i=1}^{N}\left(\frac{diam(\mathcal{M}_i)}{\eta} + GD\right)^2$.

## I.2  PROOF OF PROPOSITION I.2

The proof of Proposition I.2 follows from using the smoothness of the potential function together with the update rule of the multi-agent gradient perturbation controller algorithm.

**Lemma I.5** (Cai et al. (2024), Lemma B.6). *Under Assumptions 2 and 7, the loss function $l_t$ is $L$-smooth where $L$ is a constant depending on $H, W, \zeta, d, \kappa$.*

Using the smoothness of the loss function $l_t$ (see Lemma I.5) which plays the role of a (time-varying) potential function, we have

$$\ell_t(M_{t+1}) \le \ell_t(M_t) + \langle \nabla_M \ell_t(M_t), M_{t+1} - M_t \rangle + \frac{L}{2} \|M_{t+1} - M_t\|^2 . \tag{106}$$

Define now the product set $\mathcal{M} := \prod_{i=1}^{N} \mathcal{M}_i$ which is the space of joint policy parameters. Observe that for any $M = (M_1, \cdots, M_N) \in \mathcal{M}$, we have

$$\Pi_{\mathcal{M}}(M) := (\Pi_{\mathcal{M}_1}(M_1), \cdots, \Pi_{\mathcal{M}_N}(M_N)) . \tag{107}$$

Given the potential structure of the game, observe in addition that

$$\nabla_M \ell_t(M_t) = \left[ \nabla_i \ell_t^i(M^t) \right]_{i=1,\cdots,N} , \tag{108}$$

$$\ell_t^i = \ell_t , \tag{109}$$

$$\text{and} \quad M_t = [M_{i,t}]_{i=1,\cdots,N} , \tag{110}$$

where we recall that $\nabla_i \ell_t^i$ denotes the gradient of $\ell_t^i$ w.r.t. its variable $M_i$. As a consequence, the update rules of all the players in Algorithm 1 can be compactly written as follows:

$$M_{t+1} = \Pi_{\mathcal{M}}(M_t - \eta \nabla_M l_t(M_t)) , \tag{111}$$

where $M_{t+1} = [M_{i,t+1}]_{i=1,\cdots,N}$. Using the characterization of the projection operator, we have:

$$\forall M \in \mathcal{M}, \quad \langle M - M_{t+1}, M_t - \eta \nabla_M l_t(M_t) - M_{t+1} \rangle \le 0 . \tag{112}$$

Setting $M = M_t$ and rearranging the inequality gives:

$$\langle \nabla_M \ell_t(M_t), M_{t+1} - M_t \rangle \le -\frac{1}{\eta} \|M_{t+1} - M_t\|^2 . \tag{113}$$

It follows from injecting (113) into (106) that

$$\ell_t(M_{t+1}) \le \ell_t(M_t) + \left( \frac{L}{2} - \frac{1}{\eta} \right) \sum_{i=1}^{N} \|M_{i,t+1} - M_{i,t}\|^2 . \tag{114}$$

Setting $\eta = 1/L$, rearranging and summing up the above inequality yields the desired result, namely for all $t \ge 1$,

$$\sum_{t=1}^{T} \sum_{i=1}^{N} \|M_{i,t+1} - M_{i,t}\|^2 \le 2\eta \sum_{t=1}^{T} \ell_t(M_t) - \ell_t(M_{t+1}) . \tag{115}$$

## I.3  PROOF OF PROPOSITION I.3

First, we decompose the sum of difference of losses as follows:

$$\sum_{t=1}^{T} \ell_t(M_t) - \ell_t(M_{t+1}) = \sum_{t=1}^{T} \ell_t(M_t) - \ell_{t+1}(M_{t+1}) + \sum_{t=1}^{T} \ell_{t+1}(M_{t+1}) - \ell_t(M_{t+1})$$

$$= \ell_1(M_1) - \ell_{T+1}(M_{T+1}) + \sum_{t=1}^{T} \ell_{t+1}(M_{t+1}) - \ell_t(M_{t+1})$$

$$\le \ell_1(M_1) - c_{\inf} + \sum_{t=1}^{T} \ell_{t+1}(M_{t+1}) - \ell_t(M_{t+1}) , \tag{116}$$

where the second identity follows from simplifying the telescoping sum and the last inequality uses our uniform lower bound assumption on the cost function.

Now we control each term of the last sum above. Recall that for any $M \in \prod_{i=1}^{N} \mathcal{M}_i$,

$$\ell_t(M) := c_t(y_t^K(M), v_t^{i,K_i}(M_i)), \tag{117}$$

where $K := (K_i, K_{-i})$ and $y_t^K(M), v_t^{i,K_i}(M_i)$ are the counterfactual state and action induced by the (DAC-$i$) policy with the matrix $K$ and the policy parameters $M$ as previously defined.

We start with the following decomposition:

$$\ell_{t+1}(M_{t+1}) - \ell_t(M_{t+1}) = c_{t+1}(y_{t+1}^K(M_{t+1})), v_{t+1}^{i,K_i}(M_{i,t+1})) - c_t(y_{t+1}^K(M_{t+1})), v_{t+1}^{i,K_i}(M_{i,t+1}))$$
$$+ c_t(y_{t+1}^K(M_{t+1})), v_{t+1}^{i,K_i}(M_{i,t+1})) - c_t(y_t^K(M_{t+1})), v_t^{i,K_i}(M_{i,t+1})). \tag{118}$$

For the first term, we have

$$c_{t+1}(y_{t+1}^K(M_{t+1})), v_{t+1}^{i,K_i}(M_{i,t+1})) - c_t(y_{t+1}^K(M_{t+1})), v_{t+1}^{i,K_i}(M_{i,t+1})) \leq \max_{\|x\|, \|u\| \leq D} c_{t+1}(x, u) - c_t(x, u). \tag{119}$$

For the second term, we use Assumption 1 to write

$$c_t(y_{t+1}^K(M_{t+1}), v_{t+1}^{i,K_i}(M_{i,t+1})) - c_t(y_t^K(M_{t+1}), v_t^{i,K_i}(M_{i,t+1}))$$
$$\leq GD \cdot (\|y_{t+1}^K(M_{t+1}) - y_t^K(M_{t+1})\| + \|v_{t+1}^{i,K_i}(M_{i,t+1}) - v_t^{i,K_i}(M_{i,t+1})\|). \tag{120}$$

Define the following convenient notations for the counterfactual state and control differences for the rest of this proof:

$$\Delta_{t+1}^y := y_{t+1}^K(M_{t+1}) - y_t^K(M_{t+1}),$$
$$\Delta_{t+1}^v := v_{t+1}^{i,K_i}(M_{i,t+1}) - v_t^{i,K_i}(M_{i,t+1}). \tag{121}$$

Using these notations together with (120) and (119) in (116), it follows that:

$$\sum_{t=1}^{T} \ell_t(M_t) - \ell_t(M_{t+1}) \leq \ell_1(M_1) - c_{\inf} + \sum_{t=1}^{T} \max_{\|x\|, \|u\| \leq D} c_{t+1}(x, u) - c_t(x, u) + GD \sum_{t=1}^{T} (\|\Delta_{t+1}^y\| + \|\Delta_{t+1}^v\|). \tag{122}$$

It remains to bound $\sum_{t=1}^{T} \|\Delta_{t+1}^y\| + \|\Delta_{t+1}^v\|$ to conclude the proof of Proposition I.3. We upper bound each one of the terms separately starting with the first one ($\sum_{t=1}^{T} \|\Delta_{t+1}^y\|$) which will be useful for bounding the second one ($\sum_{t=1}^{T} \|\Delta_{t+1}^v\|$).

**Bound of $\sum_{t=1}^{T} \|\Delta_{t+1}^y\|$.** We split the sum into two sums by isolating the first burn-in period of time length $2H + 1$ for $T \geq 2H + 1$,

$$\sum_{t=1}^{T} \|\Delta_{t+1}^y\| = \sum_{t=1}^{2H} \|\Delta_{t+1}^y\| + \sum_{t=2H+1}^{T} \|\Delta_{t+1}^y\|. \tag{123}$$

The first sum can be bounded as follows using the boundedness of the counterfactual states by $D$,

$$\sum_{t=1}^{2H} \|\Delta_{t+1}^y\| \leq 2HD. \tag{124}$$

The second sum requires a special treatment using the expression of the evolution of the counterfactual state involving the state transfer matrix which gives:

$$\Delta_{t+1}^y = \sum_{l=0}^{2H} \bar{\Psi}_{t+1,l}^H(M_{t+1}) \xi_{t-l}, \quad \xi_{t-l} := w_{t+1-l} - w_{t-l}, \tag{125}$$

$$\bar{\Psi}_{t+1,l}^H(M_{t+1}) := \bar{A}_K^l \mathbf{1}_{l \leq H} + \sum_{k=0}^{H} \bar{A}_K^k \sum_{i=1}^{N} B_i M_{i,t+1-k}^{[l-k-1]} \mathbf{1}_{l-k \in [1,H]}, \tag{126}$$

where the last transfer matrix was previously introduced in (10) and the first identity follows from using Proposition D.1. For $t \geq 2H + 1$, we have

$$\sum_{l=0}^{2H} \bar{\Psi}_{t+1,l}^H (M_{t+1}) \xi_{t-l} = \sum_{l=0}^{H} \bar{A}_K^l \xi_{t-l} + \sum_{l=0}^{2H} \sum_{k=0}^{H} \bar{A}_K^k \sum_{i=1}^{N} B_i M_{i,t+1-k}^{[l-k-1]} \mathbf{1}_{l-k \in [1,H]} \xi_{t-l} \qquad (127)$$

$$= \sum_{l=0}^{H} \bar{A}_K^l \xi_{t-l} + \sum_{l=0}^{2H} \sum_{p=1}^{l} \bar{A}_K^{l-p} \sum_{i=1}^{N} B_i M_{i,t+1-k}^{[p-1]} \xi_{t-l} \,, \qquad (128)$$

where the last identity follows from a change of index $p = l - k$ and the fact that $p \in [1 : H], k \geq 0$. Using now $(\bar{\kappa}, \bar{\gamma})$-strong stability together with the bound on matrices $M_{i,t+1-k}^{[p]}$ specified by the projection sets $\mathcal{M}_i$ (see Algorithm 1), we obtain

$$\sum_{t=2H+1}^{T} \|\Delta_{t+1}^y\| \leq \sum_{t=2H+1}^{T} \sum_{l=0}^{H} \bar{\kappa}(1-\bar{\gamma})^l \|\xi_{t-l}\| + \sum_{t=2H+1}^{T} \sum_{l=0}^{2H} \sum_{p=1}^{l} \bar{\kappa}(1-\bar{\gamma})^{l-p} \sum_{i=1}^{N} \|B_i\| 2\bar{\kappa}^2 (1-\bar{\gamma})^p \|\xi_{t-l}\|$$

$$\leq \sum_{t=2H+1}^{T} \sum_{l=0}^{H} \bar{\kappa}(1-\bar{\gamma})^l \|\xi_{t-l}\| + \left(2\bar{\kappa}^3 \sum_{i=1}^{N} \|B_i\|\right) \sum_{t=2H+1}^{T} \sum_{l=0}^{2H} l(1-\bar{\gamma})^l \|\xi_{t-l}\|$$

$$\leq \sum_{t=2H+1}^{T} \sum_{l=0}^{H} \bar{\kappa}(1-\bar{\gamma})^l \|\xi_{t-l}\| + \left(2\bar{\kappa}^3 \sum_{i=1}^{N} \|B_i\|\right)(2H+1) \sum_{t=2H+1}^{T} \sum_{l=0}^{2H} (1-\bar{\gamma})^l \|\xi_{t-l}\|$$

$$= \left(\bar{\kappa} + 2(2H+1)\bar{\kappa}^3 \sum_{i=1}^{N} \|B_i\|\right) \sum_{t=2H+1}^{T} \sum_{l=0}^{H} (1-\bar{\gamma})^l \|\xi_{t-l}\|$$

$$= \left(\bar{\kappa} + 2(2H+1)\bar{\kappa}^3 \sum_{i=1}^{N} \|B_i\|\right) \sum_{s=H+1}^{T} \sum_{l=0}^{H} (1-\bar{\gamma})^l \|\xi_s\|$$

$$\leq \frac{\bar{\kappa} + 2(2H+1)\bar{\kappa}^3 \sum_{i=1}^{N} \|B_i\|}{\bar{\gamma}} \sum_{s=H+1}^{T} \|\xi_s\| \,, \qquad (129)$$

where the last equality follows from re-indexing the sum ($s = t - l$) and using $2H + 1 \leq t \leq T$ and $0 \leq l \leq H$. In conclusion, we obtain by combining (129) and (124) that

$$\sum_{t=1}^{T} \|\Delta_{t+1}^y\| \leq 2HD + \frac{\bar{\kappa} + 2(2H+1)\bar{\kappa}^3 \sum_{i=1}^{N} \|B_i\|}{\bar{\gamma}} \sum_{s=H+1}^{T} \|w_{s+1} - w_s\| \,. \qquad (130)$$

**Bound of $\sum_{t=1}^{T} \|\Delta_{t+1}^v\|$.** For this term, we use the definition of the counterfactual state to obtain for every $t \geq H$:

$$\|\Delta_{t+1}^v\| = \left\| K_i \Delta_{t+1}^y + \sum_{p=1}^{H} M_{i,t+1}^{[p-1]} (w_{t+1-p} - w_{t-p}) \right\|$$

$$\leq \bar{\kappa} \|\Delta_{t+1}^y\| + \sum_{p=1}^{H} \bar{\kappa}(1-\bar{\gamma})^p \|w_{t+1-p} - w_{t-p}\| \,. \qquad (131)$$

Therefore summing up these inequalities for $2H + 1 \leq t \leq T$ yields:

$$\sum_{t=2H+1}^{T} \|\Delta_{t+1}^v\| \leq \bar{\kappa} \sum_{t=2H+1}^{T} \|\Delta_{t+1}^y\| + \bar{\kappa} \sum_{t=2H+1}^{T} \sum_{p=1}^{H} (1-\bar{\gamma})^p \|w_{t+1-p} - w_{t-p}\|$$

$$= \bar{\kappa} \sum_{t=2H+1}^{T} \|\Delta_{t+1}^y\| + \bar{\kappa} \sum_{s=H+1}^{T-1} \sum_{p=1}^{H} (1-\bar{\gamma})^p \|w_{s+1} - w_s\|$$

$$\leq \bar{\kappa} \sum_{t=2H+1}^{T} \|\Delta_{t+1}^y\| + \frac{\bar{\kappa}}{\bar{\gamma}} \sum_{s=H+1}^{T-1} \|w_{s+1} - w_s\| \,. \qquad (132)$$

Similarly to (124), using boundedness of the counterfactual actions, we get

$$\sum_{t=1}^{2H} \|\Delta_{t+1}^v\| \leq 2HD. \tag{133}$$

Combining (133) with (132) and (129), we obtain

$$\sum_{t=1}^{T} \|\Delta_{t+1}^v\| \leq 2HD + \left( \frac{\bar{\kappa}^2 + 2(2H+1)\bar{\kappa}^4 \sum_{i=1}^{N} \|B_i\|}{\bar{\gamma}} + \frac{\bar{\kappa}}{\bar{\gamma}} \right) \sum_{s=H+1}^{T} \|w_{s+1} - w_s\|. \tag{134}$$

Finally to conclude the proof of Proposition I.3, we inject (134) and (130) into (122) to obtain the desired result:

$$\sum_{t=1}^{T} l_t(M_t) - l_t(M_{t+1}) \leq l_1(M_1) - c_{\inf} + \sum_{t=1}^{T} \max_{\|x\|, \|u\| \leq D} c_{t+1}(x, u) - c_t(x, u)$$

$$+ GD \left( 4HD + \frac{\bar{\kappa} + 2\bar{\kappa}^2 + 4(2H+1)\bar{\kappa}^4 \sum_{i=1}^{N} \|B_i\|}{\bar{\gamma}} \right) \sum_{s=H+1}^{T} \|w_{s+1} - w_s\|. \tag{135}$$

This concludes the proof of Proposition I.3. We have shown that

$$\sum_{t=1}^{T} l_t(M_t) - l_t(M_{t+1}) = \mathcal{O}\left( l_1(M_1) - c_{\inf} + \sum_{t=1}^{T} \Delta_{c_t} + \sum_{t=1}^{T} \|w_{t+1} - w_t\| \right), \tag{136}$$

where $\Delta_{c_t} := \max_{\|x\|, \|u\| \leq D} \{c_{t+1}(x, u) - c_t(x, u)\}$ for every $t$ and the $\mathcal{O}(\cdot)$ notation only hides polynomial dependence in the problem parameters $N, H, W, \bar{\kappa}, \bar{\gamma}^{-1}, \max_i \|B_i\|$ where $D$ also depends polynomially on the same constants.

## I.4 PROOF OF PROPOSITION I.4

The proof proceeds in several steps as follows:

**(i) Convexity.** Using convexity of the loss function $l_t^i$ w.r.t. $M_i$ (see Lemma 3.1), we have for every player $i \in [N]$ and every time step $t \geq 1$,

$$\ell_t^i(M_i, M_{-i,t}) - \ell_t^i(M_t) \geq \langle \nabla_i \ell_t^i(M_t), M_i - M_{i,t} \rangle$$
$$= \langle \nabla_i \ell_t^i(M_t), M_i - M_{i,t+1} \rangle + \langle \nabla_i \ell_t^i(M_t), M_{i,t+1} - M_{i,t} \rangle. \tag{137}$$

**(ii) Lower-bound of the first inner product in** (137)**.** Recall now the gradient update rule of Algorithm 1:

$$M_{i,t+1} = \text{Proj}_{\mathcal{M}_i} \left( M_{i,t} - \eta \nabla_i \ell_t^i(M_t) \right). \tag{138}$$

Using the characterization of the projection yields:

$$\forall M_i \in \mathcal{M}_i, \langle M_i - M_{i,t+1}, M_{i,t} - M_{i,t+1} - \eta \nabla_i \ell_t^i(M_t) \rangle \leq 0. \tag{139}$$

Rearranging this inequality and using the Cauchy-Schwarz inequality, we obtain:

$$\langle M_i - M_{i,t+1}, \nabla_i \ell_t^i(M_t) \rangle \geq \frac{1}{\eta} \langle M_i - M_{i,t+1}, M_{i,t} - M_{i,t+1} \rangle$$

$$\geq -\frac{1}{\eta} \|M_i - M_{i,t+1}\| \cdot \|M_{i,t} - M_{i,t+1}\|$$

$$\geq -\frac{\text{diam}(\mathcal{M}_i)}{\eta} \|M_{i,t} - M_{i,t+1}\|, \tag{140}$$

where $\text{diam}(\mathcal{M}_i) := \max_{M, M' \in \mathcal{M}_i} \|M' - M\|$.

**(iii) Lower-bound of the second inner product in** (137)**.** Using again the Cauchy-Schwarz inequality gives

$$\langle \nabla_i \ell_t^i(M_t), M_{i,t+1} - M_{i,t} \rangle \geq -\|\nabla_i \ell_t^i(M_t)\| \cdot \|M_{i,t+1} - M_{i,t}\|. \tag{141}$$

Then, using the boundedness of the gradients following from Assumption 1, there exists a constant $GD > 0$ (independent of $t$ and $i$) s.t. $\|\nabla_i l_t^i(M_t)\| \leq GD$. Therefore, we obtain

$$\langle \nabla_i \ell_t^i(M_t), M_{i,t+1} - M_{i,t}\rangle \geq -GD\|M_{i,t+1} - M_{i,t}\| . \tag{142}$$

**(iv) Combining all the steps.** Using (140) and (142) in (137), we have for all $i \in [N], M_i \in \mathcal{M}_i$, and $t \geq 1$

$$\ell_t^i(M_i, M_{-i,t}) - \ell_t^i(M_t) \geq -\left(\frac{\mathrm{diam}(\mathcal{M}_i)}{\eta} + GD\right)\|M_{i,t+1} - M_{i,t}\| , \tag{143}$$

where $\mathrm{diam}(\mathcal{M}_i) := \max_{M,M' \in \mathcal{M}_i} \|M' - M\|$ and $G, D$ are the constants defined in Assumption 1. This concludes the proof of Proposition I.4.

### I.5  PROOF OF LEMMA 3.1

Recall that the loss function $\ell_t^i$ is defined for every $M_i \in \mathcal{M}_i$ by

$$\ell_t^i(M_i) = c_t^i(y_t^{i,K_i}(M_i), v_t^{i,K_i}(M_i)) , \tag{144}$$

where the counterfactual idealized state $y_t^{i,K_i}(M_i)$ and action $v_t^{i,K_i}(M_i)$ are defined in section D.1.

By Assumption 1, the loss function $c_t^i$ is convex w.r.t. both its variables. It suffices to show that $y_t^{i,K_i}(M_i)$ and $v_t^{i,K_i}(M_i)$ are both affine in $M_i = M_i^{[1:H]}$ to obtain the desired result as the composition of a convex function and an affine function is also convex. This is clearly the case given the state evolution unfolding using the transfer matrix, see section D.2, (9)-(10) for the transfer matrices which are linear in the policy parameter $M_i$ of agent $i$ and (37)-(43) for the unrolled expressions of $y_t^{i,K_i}(M_i)$ and $v_t^{i,K_i}(M_i)$. Note that this result holds in both cases where other agents but $i$ use either arbitrary control inputs or DAC policies throughout time.

## J  TOOLS FROM ONLINE CONVEX OPTIMIZATION

### J.1  ONLINE CONVEX OPTIMIZATION WITH MEMORY

---
**Algorithm 2** Online Gradient Descent with Memory
---
1: **Input:** step size $\eta$, loss functions $\{\ell_t\}_{t=1}^T$.
2: Initialize $x_0, \ldots, x_{H-1} \in \mathcal{K}$ arbitrarily.
3: **for** $t = H \ldots T$ **do**
4:    Play $x_t \in \mathcal{K}$, suffer loss $\ell_t(x_{t-H}, \ldots, x_t)$.
5:    Set $x_{t+1} = \Pi_{\mathcal{K}}(x_t - \eta\nabla\ell_t(x_t, \ldots, x_t))$.
6: **end for**
---

**Theorem J.1** (Anava et al. (2015)). *Let $\{\ell_t\}_{t=1}^T$ be a sequence of loss functions where $\ell_t : \mathcal{X}^{H+1} \to \mathbb{R}$ for each $t \in [T]$. Moreover, suppose the following hold:*

*1. (Coordinate-wise Lipschitzness): There exists $L > 0$ s.t. for any $x_1, \ldots, x_H, \tilde{x}_j \in \mathcal{X}$,*

$$\left|\ell_t(x_1, \ldots, x_j, \ldots, x_H) - \ell_t(x_1, \ldots, \tilde{x}_j, \ldots, x_H)\right| \leq L\|x_j - \tilde{x}_j\| .$$

*2. (Bounded gradients) There exists $G_0 > 0$ s.t. for all $x \in \mathcal{X}$ and $t \in [T]$, $\|\nabla f_t(x, \ldots, x)\| \leq G_0$.*

*3. (Bounded diameter) There exists $D_0 > 0$ s.t. for all $x, y \in \mathcal{X}$, $\|x - y\| \leq D_0$.*

*Then running Algorithm 2 for $T$ iterations with any positive stepsize $\eta$ yields:*

$$\sum_{t=H}^T \ell_t(x_{t-H}, \ldots, x_t) - \min_{x \in \mathcal{X}} \sum_{t=H}^T \ell_t(x, \ldots, x) \leq \frac{D_0^2}{\eta} + (G_0^2 + LH^2 G_0)\eta T . \tag{145}$$

*Running Algorithm 2 for $T$ iterations with stepsize $\eta := D_0/\sqrt{G_0(G_0 + LH^2)T}$ guarantees:*

$$\sum_{t=H}^T \ell_t(x_{t-H}, \ldots, x_t) - \min_{x \in \mathcal{X}} \sum_{t=H}^T \ell_t(x, \ldots, x) \leq 3D_0\sqrt{G_0(G_0 + LH^2)T} .$$

We provide a few remarks regarding this result and its use in our work:

- This result has been used in single-agent online control.

- Note that we are using here the notations $D_0, G_0$ to avoid confusion with the constant $D$ defined in (31) and $G$ as introduced in Assumption 1-(ii).

- The specification of the constants $G_0, L$ and the stepsize $\eta$ in our setting will be important to elucidate the dependence of our final regret bound on the number $N$ of agents involved in our multi-agent setting.

## J.2 TIME REGRET DECOMPOSITION

**Lemma J.2.** *For every agent* $i \in [N]$, *every horizon* $H \geq 1$ *and every time* $T \geq H$, *we have:*

$$\text{Reg}_i^T(\mathcal{A}_i, \{u_t^{-i}\}, \Pi_i) \leq \text{Reg}_i^{0:H-1}(\mathcal{A}_i, \{u_t^{-i}\}, \Pi_i) + \text{Reg}_i^{H:T}(\mathcal{A}_i, \{u_t^{-i}\}, \Pi_i), \qquad (146)$$

*where we recall that* $\text{Reg}_i^{H:T}(\mathcal{A}_i, \{u_t^{-i}\}, \Pi_i)$ *is defined in* (51) *and* $\{u_t^{-i}\}$ *is an arbitrary sequence.*

*Proof.* From the definition of the regret of agent $i$, we can write

$$\text{Reg}_i^T(\mathcal{A}_i, \{u_t^{-i}\}, \Pi_i) \;=\; \sum_{t=0}^{T} c_t^i(x_t, u_t^i) - \min_{\pi^i \in \Pi_i} \sum_{t=0}^{T} c_t^i(x_t^{\pi^i}, u_t^{\pi^i})$$

$$=\; \sum_{t=0}^{H-1} c_t^i(x_t, u_t^i) + \sum_{t=H}^{T} c_t^i(x_t, u_t^i) - \min_{\pi^i \in \Pi_i} \sum_{t=0}^{T} c_t^i(x_t^{\pi^i}, u_t^{\pi^i}). \qquad (147)$$

Now observe that

$$\min_{\pi^i \in \Pi_i} \sum_{t=0}^{T} c_t^i(x_t^{\pi^i}, u_t^{\pi^i}) \geq \min_{\pi^i \in \Pi_i} \sum_{t=0}^{H-1} c_t^i(x_t^{\pi^i}, u_t^{\pi^i}) + \min_{\pi^i \in \Pi_i} \sum_{t=H}^{T} c_t^i(x_t^{\pi^i}, u_t^{\pi^i}). \qquad (148)$$

The desired result follows from combining (147) and (148). $\qquad\square$

# K SIMULATIONS

## K.1 SETTING

We consider a 2-dimensional ($d = 2$) LDS with $N = 3$ agents and scalar control inputs ($k_i = 1$ for $i \in \{1, 2, 3\}$) with:

$$A = \begin{bmatrix} 0.95 & 0.1 \\ 0 & 0.9 \end{bmatrix}, \quad B_1 = B_3 = \begin{bmatrix} 1 \\ 0 \end{bmatrix}, \quad B_2 = \begin{bmatrix} 0 \\ 1 \end{bmatrix}, \quad x_0 = \begin{bmatrix} 6 \\ 6 \end{bmatrix}. \quad (149)$$

Each agent $i \in \{1, 2, 3\}$ has their quadratic cost function $c_t^i(x, u^i) = x^\top Q_i x + r_i(u^i)^2$ where:

$$Q_i = \begin{bmatrix} 1 + 0.8i & 0 \\ 0 & 1 + 0.4(N - 1 - i) \end{bmatrix}, \quad r_i = 0.1(1 + 0.4i), \quad u^i \in \mathbb{R}. \quad (150)$$

The cost functions reflect different distances to the origin goal state $(0, 0)$, see figures 1, 2, 3 below (bottom right subplots).

We test Algorithm 1 with three different kinds of disturbances $w_t \in \mathbb{R}^2$:

(1) Constant disturbance: $w_t = 0.7$ (see Fig. 1),

(2) Sinusoidal disturbance: $w_{t,1} = \sin(0.1t), w_{t,2} = \sin(0.1t)$ where $w_{t,1}, w_{t,2}$ are the coordinates of $w_t$ (see Fig. 2),

(3) Independent and identically distributed Gaussian: $w_t \sim \mathcal{N}(0, \sigma^2)$ with $\sigma = 0.5$ (see Fig. 3).

For the hyperparameters of Algorithm 1, we set $T = 500$ for the time horizon, $\eta = 10^{-4}$ for the step size, $H = 5$ for the memory parameter and DAC policy parameters are initialized with zero values.

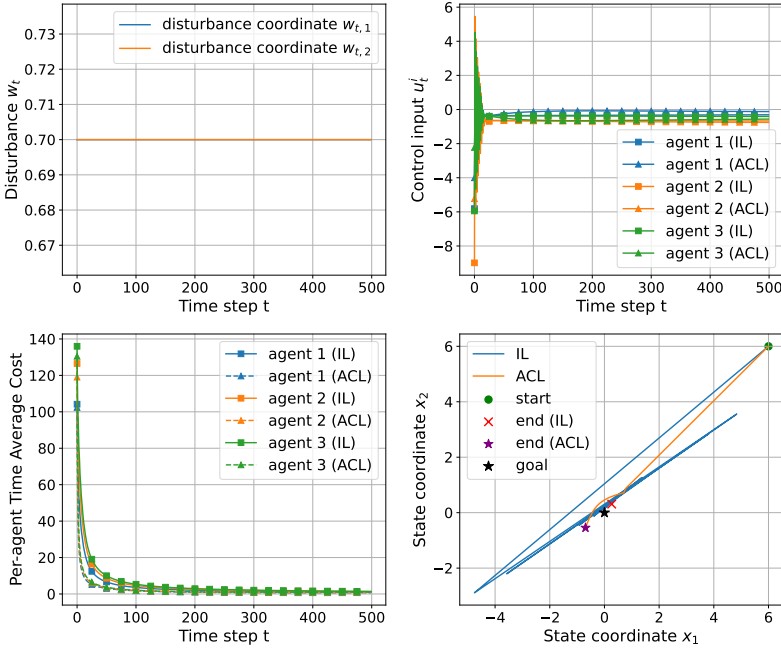

Figure 1: Illustration of the performance of Algorithm 1 on a simple multi-agent LDS with constant disturbance sequence. 'IL' stands for Independent Learning (see Information Setting 1, 'ACL' for Aggregated Control Learning (see Information Setting 2), 'start' refers to the initial state $x_0$, 'end' to the state at the last time step for $t = T$.

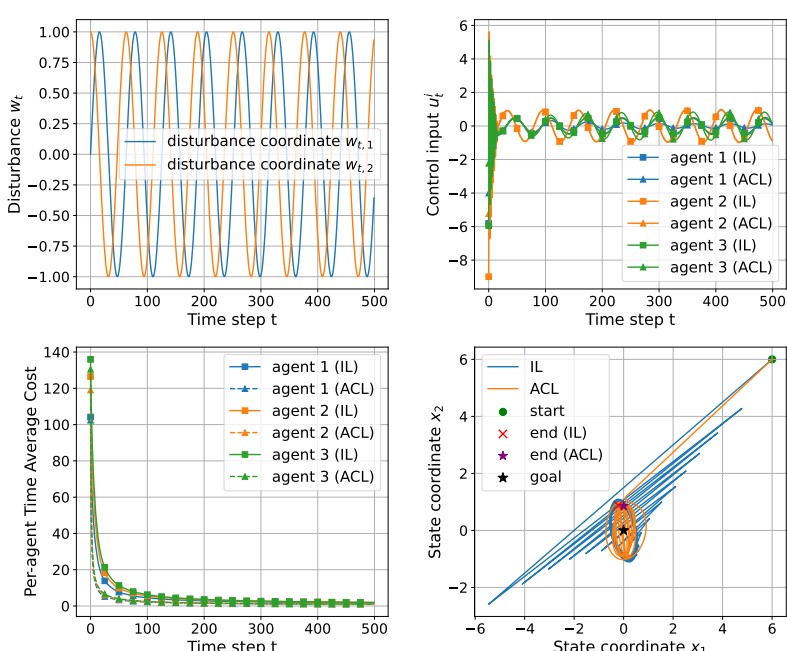

Figure 2: Illustration of the performance of Algorithm 1 on a simple multi-agent LDS with sinusoidal disturbance sequence. 'IL' stands for Independent Learning (see Information Setting 1, 'ACL' for Aggregated Control Learning (see Information Setting 2), 'start' refers to the initial state $x_0$, 'end' to the state at the last time step for $t = T$.

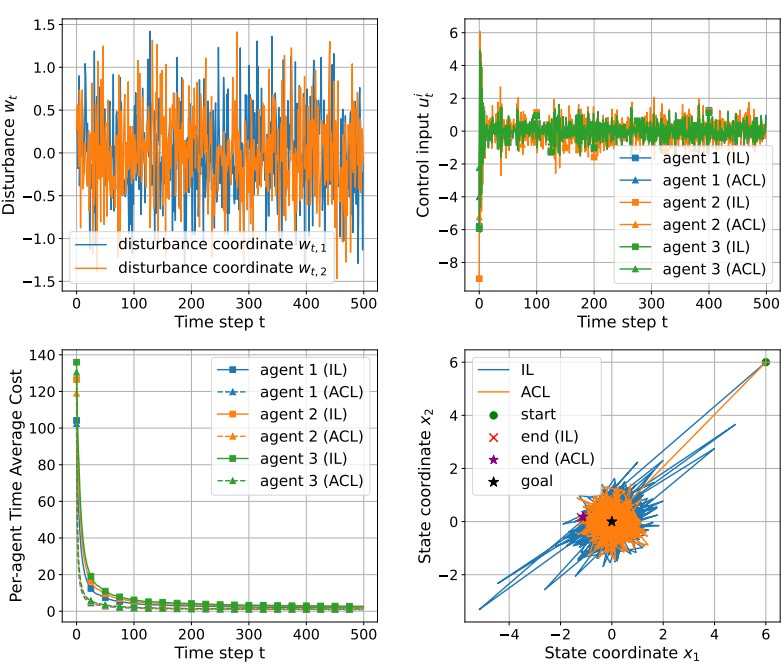

Figure 3: Illustration of the performance of Algorithm 1 on a simple multi-agent LDS with Gaussian disturbance sequence. 'IL' stands for Independent Learning (see Information Setting 1, 'ACL' for Aggregated Control Learning (see Information Setting 2), 'start' refers to the initial state $x_0$, 'end' to the state at the last time step for $t = T$.

## K.2 COMMENTS

We make a few remarks on the results of the simulations (see Figs. 1, 2, 3 above):

- Starting from the initial state $x_0$ (see bottom right subplots in all the figures), the state evolves quickly towards the goal origin state minimizing the costs by strong stability of the controllers. In particular, this quick phase corresponds to an application of higher control inputs by all agents in all three disturbance scenarios (see Figs. 1 to 3, top right subplots). Then the control inputs have a similar shape to the disturbances themselves to stabilize the system (almost constant in the first case, sinusoidal in the second and random Gaussian in the third case).

- Remark that in all figures (bottom left subplots), per-agent time-average costs vanish over time as expected. This corroborates our theoretical guarantees regarding the behavior of Algorithm 1 and our individual regret guarantees.

- It can be seen in all the figures that there is a slight advantage to the ACL setting (which can infer the disturbance values) compared to the independent learning setting in our simple simulation setting. Compare for instance the dotted per-agent time-average cost curves to the plain ones in bottom left subplots of all three figures.

- We can also observe from all the state trajectories (bottom right subplots) that Algorithm 1 in the ACL setting is more stable than in the IL setting. For instance, in the sinusoidal case (see Fig. 2, bottom right subplot), there are less oscillations and their magnitude is smaller in the ACL setting as expected from our theory. In particular, the state trajectory converges to a neighborhood of the goal state defined by the amplitude of the disturbance sequence. The same observation can be made in the case of the Gaussian noise disturbance where the state trajectory in the ACL setting concentrates more around the origin than in the IL setting as expected. The region of concentration is controlled by the standard deviation of the Gaussian noise disturbance sequence.

