# OpenReview forum: "Online Multi-Agent Control with Adversarial Disturbances"
_ICLR.cc/2026/Conference — Submitted to ICLR 2026_

### Official Review · Reviewer_jhEz · 2025-10-15

**Soundness:** 2
**Presentation:** 3
**Contribution:** 2
**Rating:** 6
**Confidence:** 3

**Summary:**

This paper focus on online control in multi-agent linear dynamical systems with adversarial disturbances.

**Strengths:**

1. The presentation is very clear, with careful organization and precise mathematical exposition.

2. The proofs are detailed and technically solid. Although I did not check every appendix proof line by line, the arguments in the main text appear correct and internally consistent.

3. The multi-agent setting is both natural and valuable — it extends the single-agent adversarial control framework to a broader and practically relevant domain.

**Weaknesses:**

1. The algorithmic framework and proof techniques largely follow Agarwal et al. (2019), with relatively limited methodological novelty beyond extending to multiple agents.

2. The paper lacks any empirical validation, even simple synthetic experiments. Given that multi-agent setups are especially suitable for simulation, this absence weakens the practical impact.

3. Some assumptions (e.g., global strong stability) are relatively strong, and it is unclear how restrictive they are in practice.

**Questions:**

1. Could the authors highlight more explicitly what are the key technical differences and challenges compared to Agarwal et al. (2019)? For example, what specific parts of the regret analysis or equilibrium proof required new ideas due to the multi-agent coupling?

2. In the Aggregated Control Learning setting, the improved regret bound relies on a global strong stability assumption. Could the authors give a simple counterexample showing that without this assumption, the result may fail?

3. It would significantly strengthen the paper to include simple multi-agent simulations (e.g., 2–3 agents in a linear system) to illustrate how the proposed algorithm performs and whether the regret empirically scales as predicted.

---

> ### Author Response · Authors · 2025-11-20
> **Rebuttal**
>
> We thank the reviewer for their valuable feedback, and we appreciate their time in reviewing our manuscript. We answer each one of their questions below.
>
> **1. Technical novelty with respect to prior work**
> > Could the authors highlight more explicitly what are the key technical differences and challenges compared to Agarwal et al. (2019)? For example, what specific parts of the regret analysis or equilibrium proof required new ideas due to the multi-agent coupling?
>
>
> The scope and technical aspects of our results depart substantially from those of Agarwal et al. (2019). Our work studies an online multi-agent control problem,  while the work of Agarwal et al. deals only with a single-agent setting. Thus, the question on the price of decentralization is not relevant in the work of Agarwal et al. From a more technical viewpoint, except for Theorem 3.2 (as we mention in lines 324-326 of the manuscript), none of our results follow from an application of those in Agarwal et al. Specifically:
> - Theorem 3.3 is a lower bound not present in Agarwal et al.
> - Theorems 3.4 and 3.5 use similar bounding techniques but do not follow from applying prior work. In our multi-agent control setting, as we briefly explain in l. 378-385, the main technical challenges we overcome are two-fold: first, assuming that all the agents play DAC policies, we need to carefully rework the state evolution in order to be able to bound the magnitude of the state norms appropriately in terms of $N$ (see app. D.2, p. 18-23) using our natural global stability Assumption 4. Second, we need to control both terms of the regret decomposition (the counterfactual state and action deviation errors) by carefully selecting the memory H and an adequate step size $\eta$ (optimally in terms of N), see app. F, p. 24-28.
> - Section 4 is purely about the game-theoretic multi-agent system behavior, for which the single-agent results of Agarwal et al. are not relevant. In particular, Theorem 4.1 provides a new equilibrium tracking guarantee for the collective behavior of the system of $N$ agents that complements but is independent from individual regret guarantees.
>
> We further provide more details below on the novelty of Theorem 4.1:
>
> **Details on Theorem 4.1:** This result establishes a new global equilibrium tracking guarantee about the collective stability of the system of $N$ agents, complementing our individual regret guarantees. The proof of this result does not rely on any of the prior results of Agarwal et al. 2019 (which provides an individual regret guarantee in the single agent setting) and does not build on the regret guarantees we establish. As we mention in l. 464-467, Theorem 4.1 provides a novel equilibrium gap guarantee in our multi-agent control setting for continuous convex games where cost functions depend on state dynamics and adversarial disturbances in addition to the exogenous time-variability of costs, extending prior recent work in time-varying (stateless) normal-form games.
>
> The coupling comes from the fact that agents’ cost functions are coupled through the state evolution which depends on the controls of all agents. We upperbound the sum of equilibrium gaps by the sum of policy parameter deviations across time and players (Prop. I.1). Then we upperbound the latter policy parameter deviations by the sum of loss deviations along time (Prop. I.2). Finally, Proposition I.3 upperbounds the sum of loss deviations by the initial distance to the infimal cost value, the cost function variability and the sum of disturbance variations. For more details, please see a proof sketch and a complete proof in App. I, p. 33-39.
>
> **2. Global strong stability assumption**
> > In the Aggregated Control Learning setting, the improved regret bound relies on a global strong stability assumption. Could the authors give a simple counterexample showing that without this assumption, the result may fail?
>
> Suppose $w_t = 0$ and agents hence use linear controllers (as DAC policies degenerate in that case). As we mention in l. 359-360, the multi-agent LDS can then be written as: $x_{t+1} = A x_t - [B_1, \cdots, B_N](K_1, \cdots, K_N)^T x_t = (A - \sum_{i=1}^N B_i K_i) x_t$. Without global stability (Assumption 4), if for instance the spectral radius of $A - \sum_{i=1}^N B_i K_i$ is not strictly smaller than 1,  the states can go unbounded and their magnitude can diverge as a consequence. This hinders any regret analysis (which relies on state norm bounds).

---

> ### Author Response · Authors · 2025-11-20
> **Rebuttal end**
>
> **3. Simulations**
> > It would significantly strengthen the paper to include simple multi-agent simulations (e.g., 2–3 agents in a linear system) to illustrate how the proposed algorithm performs and whether the regret empirically scales as predicted.
>
> While our contributions are mainly theoretical, we have performed and included simple simulations corroborating our theoretical findings as suggested by the reviewer. We have implemented Algorithm 1 in both settings (independent learning and aggregated control learning) for $N=3$ agents with different quadratic cost functions on a 2 dimensional LDS with three different kinds of disturbance sequences: (1) constant nonzero disturbance, (2) sinusoidal disturbances (different for each coordinate) and (3) Gaussian noise.
> Please see section K in the last pages of the revised pdf (p. 41-43) for a more detailed description of the setting, figures (including vanishing per-agent time average costs curves, input control values and 2d state trajectories toward a goal origin state) and comments on simulations.

---

> > ### Comment · Reviewer_jhEz · 2025-11-20
> >
> > I have read the rebuttal and additional simulations provided by the authors. These results are quite interesting, and I believe they would further strengthen the paper if included in the main text rather than only in the appendix. I am happy to raise my score accordingly.

---

> > > ### Author Response · Authors · 2025-11-20
> > >
> > > We thank the reviewer for their prompt response and encouraging feedback. We appreciate their suggestion and will incorporate these results to the main part if space permits.

---

### Official Review · Reviewer_JSuY · 2025-10-16

**Soundness:** 4
**Presentation:** 4
**Contribution:** 2
**Rating:** 6
**Confidence:** 4

**Summary:**

This work is a follow up to Agarwal 2019 in the multi-agent setup.  In particular, the authors study two setups:
1. Each agent observes their own actions;
2. Each agent additionally observes the accumulated actions.

The authors show that the second setup lead to better N dependence in the regret.

Further, the authors consider a special setup when all agent has the same cost function. Here the approximation error of nash equilibrium can be bounded as in a online convex optimization problem, and hence the average action converges to the "nonstationary" Nash equilibrium.

**Strengths:**

This work extends [Agarwal 2019] in an interesting direction. Analyzing regret / convergence / stability for multi-agent systems can always be challenging. To address the challenge, the authors identify nontrivial multi-agent setups in which learning can happen.

Further, the paper is techinically sound and easy to follow. The authors may benifit from a better comparison of their analysis to the original analysis in Agarwal, and highlight the technical difficulty.

**Weaknesses:**

My concern is about how nontrivial the analyses are. For the three results:

1. When each individual only observe their own actions: it seems one doesn't need to care about other actions because the perturbation is already adversarial. Further, the disturbance caused by other agents' action is bounded due to the system being stable. Therefore, it seems one can just apply [Agarwal] result directly with N different copies.

2. When the individual agent can observe actions from other agents, they can infer the perturbation and get better "problem coefficients" (e.g. disturbance size) when apply Agarwal.

3. When all agents share the same cost, the problem is just online learning, except that its not exactly convex. However, when use DAC class, the problem is not very far away from a convex surrogate with larger decision space.

I am not 100% sure my understanding is accurate, but after reading the results, all the bounds seem expected, and relatively straight-forward.

**Questions:**

See weakness

---

> ### Author Response · Authors · 2025-11-20
> **Rebuttal**
>
> We thank the reviewer for their time and feedback.
>
>
> Aside from Theorem 3.2 (independent learning regret upper bound), which follows from applying the result of Agarwal et al. 2019 (as we mention in lines 324-326 of the manuscript), the technical aspects of our results depart substantially from those of Agarwal et al. Specifically:
> - Theorem 3.3 is a new regret lower bound which was not derived in prior work.
> - Theorems 3.4 and 3.5 (aggregated learning regret upper bounds) do not follow from an immediate application of existing work. While the $\sqrt{T}$ dependence in these upper bounds may be expected, and while we use similar proof techniques as in prior work, our results  quantify the price of decentralization in terms of scaling with $N$.
> Our analysis requires new technical developments, including controlling state evolution and counterfactual cost deviation errors in our multi-agent setting (see details below).
> - The results of section 4 are about collective game-theoretic behavior. As such, these results are not covered by any prior work on single-agent online control that deal with **individual regret bounds**.
>
> Below, we give more details on the technical novelty of our results in response to the reviewer's comments:
>
> **1. Theorem 3.2 (regret upper bound, independent learning setting)**
> > When each individual only observe their own actions: it seems one doesn't need to care about other actions because the perturbation is already adversarial. Further, the disturbance caused by other agents' action is bounded due to the system being stable. Therefore, it seems one can just apply [Agarwal] result directly with N different copies.
>
> As we explain in section 3.1 and lines 324-326, for Theorem 3.2 (and only for this theorem), this is exactly what we do: we apply the single-agent regret bound from Agarwal et al. However this leads to a quadratic dependence on the number $N$ of agents in the individual regret bound. We improve this dependence in section 3.2 and the proofs do not follow from applying known single agent results. Note also that we provide a lower bound in terms of the dependence on the time horizon $T$ that is not present in Agarwal et al. 2019. See also the response to reviewer nX9c for further details.
>
> **2. Theorem 3.4 (regret upper bound, ACL setting)**
> > When the individual agent can observe actions from other agents, they can infer the perturbation and get better "problem coefficients" (e.g. disturbance size) when apply Agarwal.
>
> Theorem 3.4 does not follow from any application of results from Agarwal et al.
> While in the setting of the theorem, the perturbation can be inferred (as observed by the reviewer and as we explain in the manuscript), we do not apply the result of Agarwal et al. to obtain Theorem 3.4, even if our proof technique for this theorem is similar. In particular, while the $\sqrt{T}$ scaling in the bound is expected in our regret upper bounds, the improvement in $N$ and the exact linear dependence are not straightforward. In our multi-agent control setting, as we briefly explain in l. 378-385, the main technical challenges we overcome are two-fold: first, assuming that all the agents play DAC policies, we need to carefully rework the state evolution in order to be able to bound the magnitude of the state norms appropriately in terms of $N$ (see app. D.2, p. 18-23) using our natural global stability Assumption 4. Second, we need to control both terms of the regret decomposition (the counterfactual state and action deviation errors) by carefully selecting the memory H and an adequate step size $\eta$ (optimally in terms of N), see app. F, p. 24-28.
>
> **3. Section 4 and Theorem 4.1 (Equilibrium gap guarantee)**
>
> > When all agents share the same cost, the problem is just online learning, except that its not exactly convex. However, when use DAC class, the problem is not very far away from a convex surrogate with larger decision space.
>
> In section 4, we establish equilibrium tracking guarantees that are new to the online learning in games literature. In contrast to standard results in online learning (even in the common interest setting), our setting is different in many aspects:
> - Our cost functions are still time-varying, not only via the dependence on other players’ strategies, but via an exogenous time dependence in the costs that shows up in our bound.
> - Our cost functions also depend on states whose evolution as an LDS is influenced by all the agents’ control inputs and by time-dependent disturbances. This is our novel online multi-agent control setting.
>
> From the technical viewpoint, as we mention in l. 463-466, we cast our setting as a time-varying game and we extend recent work considering time-varying **(finite) normal-form potential games** to (a) cover continuous convex games and (b) account for state dynamics and adversarial disturbances in addition to the time-varying costs in our multi-agent control setting.

---

> > ### Comment · Reviewer_JSuY · 2025-11-21
> > **Response**
> >
> > I thank the authors for the reply. I will keep my score, which is already positive.
> >
> > I agree that the proofs can be very involved, and one need to close certain gaps upon the analysis in [Agarwal] which was already quite complicated. However, I don't see anything unexpected. For example
> >
> > 1. In thm 3.3, it seem one can use a diagonal linear system update to decouple the interactions between the agent and then apply standard online lower bounds.
> >
> > 2. In Thm 4.1, for a collaborative game, the regret bound can naturally translate into equilibrium bound, then everything follows from the existing regret bounds.

---

> > > ### Author Response · Authors · 2025-12-04
> > > **Response**
> > >
> > > We thank the reviewer for their updated response and for acknowledging our rebuttal. We would like to clarify that Theorem 4.1 provides a *new equilibrium-tracking guarantee with a fully independent proof that does not rely on our regret bounds*. This result does not follow from standard regret-to-equilibrium arguments, which apply to fixed normal-form or static convex games and are therefore substantially weaker in comparison to our dynamic, state-dependent setting.
> > >
> > > We address the reviewer’s two comments below.
> > >
> > > ### 1. On Theorem 3.3 (lower bound)
> > >
> > > > 1. In thm 3.3, it seem one can use a diagonal linear system update to decouple the interactions between the agent and then apply standard online lower bounds.
> > >
> > > As discussed in our response to reviewer nX9c (point 3), one can indeed recover a lower bound by reducing to a trivial static system with $A = B = 0$ and fixed disturbances ($w_t = 1$). Theorem 3.3 complements this by demonstrating that the same order of lower bound also holds for nontrivial LDS, where
> > > $B \neq 0$ and the state evolves nontrivially. Our construction (Appendix H) provides such a nontrivial hard instance, establishing the lower bound in a more robust and meaningful dynamical setting.
> > >
> > > ### 2. On Theorem 4.1 (equilibrium tracking)
> > >
> > > > 2. In Thm 4.1, for a collaborative game, the regret bound can naturally translate into equilibrium bound, then everything follows from the existing regret bounds.
> > >
> > > **TL;DR:** Theorem 4.1 does *not* follow from our regret bounds or from standard no-regret to equilibrium arguments for time-invariant games. It is a new equilibrium-tracking guarantee for a dynamic, stateful potential game, proved via a dedicated potential-based analysis.
> > >
> > > We elaborate briefly below:
> > >
> > > - **Dynamic, stateful game:** the game at time $t$ is induced by the LDS, the DAC policies, and adversarial disturbances. As a result, the potential $\ell_t$ is *state-dependent and time-varying*, not a fixed normal-form payoff. Classical no-regret to equilibrium results assume a fixed game and do not apply directly to this setting.
> > >
> > > - **Best response vs fixed comparator policies:** The regret bounds in Section 3 are defined with respect to a fixed comparator policy over the full horizon (i.e. a fixed policy minimizing the sum of counterfactual costs over the horizon $T$). In contrast, the equilibrium gap $\text{EQGAP}^{(t)}(M_t)$ involves *instantaneous best-response DAC policies* at each time $t$. Controlling a sum of time-varying equilibrium gaps does not follow from external regret defined against a single static comparator policy.
> > >
> > > - **Independent analysis:** The proof of Theorem 4.1 uses a dedicated potential-based argument tailored to this dynamic game. It does not invoke the Section 3 regret bounds nor standard results from repeated static games. A proof sketch and full details are provided in Appendix I.

---

### Official Review · Reviewer_tTR7 · 2025-10-30

**Soundness:** 3
**Presentation:** 3
**Contribution:** 2
**Rating:** 6
**Confidence:** 4

**Summary:**

This paper addresses online control in multi-agent linear dynamical systems (LDS) with time-varying convex costs and adversarial disturbances, contrasting with prior work on noiseless or stochastic systems. The authors propose a decentralized, gradient-based algorithm (Algorithm 1) using Disturbance Action Controller (DAC) policies, aiming to achieve both sublinear individual regret and collective equilibrium tracking. The analysis is conducted under two information models. First, in an independent learning setting (observing only the state), the paper establishes a per-agent regret bound of $\tilde{\mathcal{O}}(N^2 \sqrt{T})$. Second, in an aggregated control learning setting (observing state and others' aggregated inputs), the bound is improved to $\tilde{\mathcal{O}}(N \sqrt{T})$, and further to a near-optimal $\tilde{\mathcal{O}}(\sqrt{T})$ with an additional Lipschitz assumption, removing the dependence on $N$. Finally, in a common interest setting (a time-varying potential game), the paper demonstrates that the no-regret algorithm successfully tracks the evolving Nash equilibria.

**Strengths:**

1. The paper tackles a challenging and relevant problem at the intersection of online non-stochastic control and online learning in games.
2. The paper provides a strong theoretical analysis. The analysis under two different information structures (Settings 1 and 2) provides a clear understanding of the value of information and the price of decentralization. The $\Omega(\sqrt{T})$ lower bound confirms the optimality of the bounds with respect to $T$.
3. The equilibrium tracking result (Theorem 4.1) is a valuable contribution.

**Weaknesses:**

1. The algorithm assumes that agents have perfect knowledge of the system dynamics ($A$ and their own $B_i$). This is a common but significant limitation. While unknown dynamics is mentioned as future work, a brief discussion in the main text about the specific challenges this would introduce (e.g., the need for system identification competing with the adversarial disturbances) would be beneficial.

2. The algorithm requires agents to possess a stabilizing linear controller $K_i$ a priori. More critically, the improved results in Setting 2 (Theorem 3.4) rely on Assumption 4, which is a global strong stability condition on the joint controller $(K_1, \dots, K_N)$. Moreover, the paper (below Assumption 4) suggests this can be centrally precomputed, which weakens the decentralized claim for Setting 2 and should be stated more explicitly in the main text. How practical is it to obtain this global controller in a multi-agent setting?

3. The regret bound in Theorem 3.2 depends on $U^2$, where $U$ is an assumed upper bound on the control inputs of other agents. This seems somewhat circular. While the algorithm's projection step bounds the agent's own DAC parameters, it's not immediately obvious that this guarantees $||u_t^j|| \le U$ for all $j \neq i$, especially when other agents are also learning. Do you need to prove that ``if all agents run Algorithm 1, then the control output $u_t$ for all will be bounded. Thus, the assumption $||u_t^j||\le U$ for all $j\in[N]$ needs more justification.

4.  The paper does not consider a fully decentralized control setting in that (i) each agent has access to the global state and (ii) each agent observes the aggregated control input from all the other agents. In a fully decentralized setting, each agent may only have access to its local state, and the communication among the agents is through a general graph structure. The authors should more clearly describe this point in the paper and also point to references in the literature that study fully decentralized control setting with local states and communication graph.

5. Theorem 3.2 provides regret against the class of strongly stable linear controllers ($\Pi_i^{\text{lin}}$), whereas the main results for Setting 2 (Theorems 3.4 and 3.5) are against the DAC policy class ($\Pi_i^{\text{DAC}}$). While DAC is a standard comparator class used to ensure convexity in online control, this is a weaker benchmark than $\Pi_i^{\text{lin}}$. This distinction should be explained and discussed more clearly in the paper.

**Questions:**

Greatly appreciated if the authors could address the weaknesses mentioned above.

---

> ### Author Response · Authors · 2025-11-20
> **Rebuttal**
>
> We thank the reviewer for their thorough review and valuable feedback. We appreciate the constructive suggestions for improvement that we will follow. We respond below to each one of the comments and questions regarding weaknesses.
>
>
> **1. Knowledge of the system dynamics $(A, B_i)$.**
>
> > The algorithm assumes that agents have perfect knowledge of the system dynamics ($A$ and their own $B_i$). This is a common but significant limitation. While unknown dynamics is mentioned as future work, a brief discussion in the main text about the specific challenges this would introduce (e.g., the need for system identification competing with the adversarial disturbances) would be beneficial.
>
> As suggested, we will add more details in the main part. Indeed, this more challenging setting requires to perform system identification as mentioned by the reviewer. A promising approach consists in developing an explore-then-commit strategy inspired by the work of Hazan et al. 2020. Here, one first estimates the unknown system dynamics before using the estimations for exploitation. Some of the challenges to address then are: (i) how to construct estimates of the dynamics using noise injection in the multi-agent setting, (ii) how to aggregate different estimates of the same matrix $A$ estimated by each agent, and (iii) how to control the state estimation error propagation due to system dynamics errors in the multi-agent LDS.
>
> **2. Stabilizing linear controllers computation.**
> > The algorithm requires agents to possess a stabilizing linear controller $K_i$ a priori. More critically, the improved results in Setting 2 (Theorem 3.4) rely on Assumption 4, which is a global strong stability condition on the joint controller $(K_1, \cdots, K_N)$ . Moreover, the paper (below Assumption 4) suggests this can be centrally precomputed, which weakens the decentralized claim for Setting 2 and should be stated more explicitly in the main text. How practical is it to obtain this global controller in a multi-agent setting?
>
> We will expand our discussion regarding this point as suggested. Indeed, we require access to a globally strongly stable joint controller for our improved results, and it would be interesting to try to relax this requirement. Nevertheless, we would like to mention a couple of points:
> - While this is a stronger requirement,  the need for a stabilizing linear controller $K_i$ is a requirement even in the single agent setting in most prior work on single-agent online control.
> - Only the DAC policy parameters are learned, even in prior work in the single agent setting. The main purpose of our online control algorithm is to obtain a controller which is robust to adversarial system disturbances ($w_t$) assuming access to stabilizing controllers. In particular, the matrices $K_i$ are not learned by the algorithm. This is a prerequisite which is certainly limiting and that would be interesting to relax.
> - Regarding computation, we agree that the need for centralized computation of the controllers $K_i$ makes the full algorithm less decentralized. However these parameters are not learned, and under their knowledge, the algorithm can be run in a decentralized way to learn the DAC parameters.
> - The stability of the joint controllers is needed to guarantee that the states remain bounded (see response to reviewer jhEz Q.2 regarding this point).
>
> **3. Boundedness of control inputs in Theorem 3.2.**
> > The regret bound in Theorem 3.2 depends on $U^2$, where $U$ is an assumed upper bound on the control inputs of other agents. This seems somewhat circular. While the algorithm's projection step bounds the agent's own DAC parameters, it's not immediately obvious that this guarantees $\|u_t^j\| \leq U$ for all $j \neq i$, especially when other agents are also learning. Do you need to prove that ``if all agents run Algorithm 1, then the control output $u_t$ for all will be bounded. Thus, the assumption $\|u_t^j\| \leq U$ for all $j \in [N]$ needs more justification.
>
>
> In the statement of Theorem 3.2, we make the assumption that all the controls of all the agents are bounded without specifying which algorithms are used by players other than player $i$. Note that if this is not the case, then the states can also grow unbounded if one of the control inputs of some agent cannot be bounded. An example where our boundedness assumption on all controls is satisfied is when all agents run Algorithm 1. In this case their DAC policy parameters are uniformly bounded by the projection step (see step 8) and it follows by the definition of DAC policies (see DAC-i) that the control inputs will also be uniformly bounded since the states are shown to be bounded (see Proposition D.3 p. 21 in the appendix) and the disturbance sequence $(w_t)$ is bounded(by Assumption 2). Then one can pick $U$ to be the maximum bound over all agents control input bounds.

---

> > ### Author Response · Authors · 2025-11-20
> > **Rebuttal end**
> >
> > **4. Fully decentralized control setting.**
> > > The paper does not consider a fully decentralized control setting in that (i) each agent has access to the global state and (ii) each agent observes the aggregated control input from all the other agents. In a fully decentralized setting, each agent may only have access to its local state, and the communication among the agents is through a general graph structure. The authors should more clearly describe this point in the paper and also point to references in the literature that study fully decentralized control setting with local states and communication graph.
> >
> >
> > We thank the reviewer for this comment, we will mention the possible extension to this challenging setting in the paper. We believe this is an interesting direction for future work. In our setting, we use the adjective ‘decentralized’ to refer to the fact that: at every timestep $t$, each agent updates their own policy parameters independently and locally in an uncoupled fashion, without any access to other agent’s policy parameters at that round. After acting, each agent first incurs the loss according to their individual cost function, and then observes the aggregated feedback used to inform their next updates. This term was similarly used in multi-agent reinforcement learning where a single state space is also shared and not necessarily a product space. As suggested, we will mention the possibility to consider the setting in which only local states can be observed by agents, in addition to the extension to the partially observable setting that is mentioned in the conclusion.
> >
> > **5. Policy class comparators.**
> > > Theorem 3.2 provides regret against the class of strongly stable linear controllers ($\Pi_i^{lin}$) whereas the main results for Setting 2 (Theorems 3.4 and 3.5) are against the DAC policy class ($\Pi_i^{DAC}$). While DAC is a standard comparator class used to ensure convexity in online control, this is a weaker benchmark than $\Pi_ i^{lin}$. This distinction should be explained and discussed more clearly in the paper.
> >
> > While standard, the DAC policy class is indeed a weaker comparator. Nevertheless, we believe it is natural to consider DAC policies as a comparator class if other players are *also* playing DAC policies (we consider that others are playing DAC to be robust to the perturbation sequence $w_t$). We will further highlight this distinction in the paper as suggested. If player $i$ compares to a linear controller, then the counterfactual state evolution will be influenced by a linear controller for player $i$’s contribution and by DAC policies for other players. This induced mismatch between the contributions of players to the state evolution makes it difficult to compare to the class of linear policies. We will add more details as suggested.

---

### Official Review · Reviewer_nX9c · 2025-11-10

**Soundness:** 3
**Presentation:** 3
**Contribution:** 2
**Rating:** 4
**Confidence:** 4

**Summary:**

This paper considers a linear dynamical system that evolves via inputs of N agents each with a different cost function, subject to adversarial disturbances. In this setting, the paper provides (A) a O(N^2T^0.5) regret bound when the agents can observe the state against linear policies, (B) a O(NT^0.5) regret against recently introduced disturbance-action policies, (c) and a bound on the sub optimality with respect to the best response produced due to individual regret minimization.

**Strengths:**

The paper puts forward and interesting setup, and it is an interesting question as to what equilibria arise due to self-centered learning behaviors in this setting.

The closest work (Ghai et al) I am aware of only considers the cooperative setting: where the feedback is limited, but the cost function is shared. This paper on the other hand models disparate costs per agent.

**Weaknesses:**

In my reading, a major weakness is that many results (Theorem 3.2. 3.3, 3.4) in the paper follow blackbox (entirely or almost) from known results, and hence the contribution of the present work in these contexts is restricted to framing.

Note that the regret is defined as treating other agents' actions fixed. Thus, Theorem 3.4 is in fact a corollary of earlier work on online non-stochastic control, purely by including the other players' actions in the "disturbances". The size of disturbances inflated by a factor of N, and this can substituted in the earlier results blackbox.

This also extends to Theorem 3.4, which primarily is a Lipschitz constant computation. Additionally, notice that here a DAC policy may not be super meaningful since ideally the agent should also respond to the other agents actions (which the linear policy does allow the agent to do).

The authors note that Theorem 3.3 does not follow from existing online convex regret lower bounds, due to policy regret differences. But consider any LDS with (A,B) = (0,0), this reduces to an online convex game over controls, and now standard T^0.5 lower bounds apply. So, I don't see the barrier here.

In contrast to the above, I really like the set of questions in Section 4. However, Theorem 4.1 provides a non-varnishing bound, thus there is no implication of convergence to an equilibrium-like notion.

**Questions:**

Can the authors comment on points I might have misunderstood above in the weakness section? Setting aside subjective judgements, I think we should be able to agree on facts.

---

> ### Author Response · Authors · 2025-11-20
> **Rebuttal (updated)**
>
> We thank the reviewer for their time and feedback. We appreciate the reviewer’s genuine engagement to understand and position our technical contributions compared to prior work.
>
> Our main goal in section 3 is to discuss the price of decentralization in our online multi-agent control setting, which is not addressed in prior work on online control. Aside from Theorem 3.2 (independent learning regret upper bound), which follows from applying the result of Agarwal et al. 2019 (as we mention in lines 324-326 of the manuscript), our proofs do not immediately apply known results of prior work. We first provide a few general comments in response to the reviewer:
> - Theorems 3.4 and 3.5 (aggregated learning regret upper bounds) do not follow from an application of existing bounds even if our proof techniques are inspired from the single-agent result of Agarwal et al. 2019.
> - Regarding Theorem 3.3 (regret lower bound), we agree that a $\Omega(\sqrt{T})$ lower bound on policy regret with respect to $\Pi_i^{\mathrm{lin}}$ can follow from a reduction to classical online convex optimization lower bounds using the trivial one-dimensional LDS instance ($A= B = 0$), see our more detailed response below for some nuances about how. In the lower bound of Theorem 3.3, we construct a nontrivial LDS with $B_i \neq 0$ and our result shows that the lower bound also holds for **nontrivial** LDS which are more relevant for our upper bounds and our control setting.
> - The results of section 4 are about collective game-theoretic behavior. As such, these results are not covered by any prior work on single-agent online control that deal with individual regret bounds. The equilibrium tracking upper bound is time dependent (without further assumptions) due to the inherent time-varying nature of the induced potential game. This equilibrium tracking performance metric is standard in the recent literature on online learning in time-varying games. We discuss conditions under which this bound is vanishing after the statement of the result.
>
> We provide a more detailed discussion below.
>
> **1. Theorem 3.2 (regret upper bound, independent learning)**
>
> > Note that the regret is defined as treating other agents' actions fixed. Thus, Theorem 3.4 is in fact a corollary of earlier work on online non-stochastic control, purely by including the other players' actions in the "disturbances". The size of disturbances inflated by a factor of N, and this can substituted in the earlier results blackbox.
>
> This is the approach we adopt for Theorem 3.2 (we guess this is a typo) as we explain in l. 324-236. As for Theorem 3.4, we do not treat other players’ actions as part of disturbances, as this leads to the worse $N^2$ dependence on the number of players that we obtain in Theorem 3.2. Note that in this setting the agents run DAC policies w.r.t. the original disturbances rather than modified disturbances as in Theorem 3.2.
>
> **2. Theorem 3.4 (regret upper bound, ACL)**
>
> > This also extends to Theorem 3.4, which primarily is a Lipschitz constant computation. Additionally, notice that here a DAC policy may not be super meaningful since ideally the agent should also respond to the other agents actions (which the linear policy does allow the agent to do).
>
> Theorem 3.4 is not primarily a Lipschitz constant computation and the result does not follow from applying prior work results. Given that other players are also using DAC policies, we do not include their contributions to the multi-agent LDS in the disturbances. Rather, we rework the state evolution (see app. D.2, Prop. D.1 (ii), eq. (13)) and we control the magnitude of states under our global stability assumption (see app. D.4, Prop. D.3). Then, we decompose regret w.r.t the class of DAC policies (see app. F.1). We control two errors in the decomposition: (i) the induced counterfactual state and action deviation errors (see app. F.2) and (ii) an online gradient descent with memory regret term (see app. F.3) which requires controlling the dependence of a lipschitz constant on the number $N$ of agents. We balance the error terms to get the best dependence on both $N$ and $T$.
>
> The key in our analysis is to treat the contributions of all agents not as a total disturbance without any additional knowledge, but rather to use the structure and magnitude of these contributions to the multi-agent LDS (e.g. the fact that they derive from DAC policies with a global stability assumption) to derive our improved regret bound. Note that prior work (Agarwal et al. 2019) is restricted to the single agent setting and does not have to deal with any of these considerations involving the dependence on the number of agents. More importantly, agents use DAC policies w.r.t the original sequence of disturbances $(w_t)$. As mentioned above, modifying this sequence to include $N$ terms as a disturbance (to apply prior work) leads to a worse $N^2$ dependence on the number of agents (as we briefly comment on in l. 386-389 and app. C.3).

---

> ### Author Response · Authors · 2025-11-20
> **Rebuttal continued (updated)**
>
> Please see app. D p. 18-24 and app. F p. 24-29 for more details regarding the proofs.
>
> As for DAC policies, note that each agent does respond implicitly to other agents controls via their individual fixed matrix $K_i$ (which satisfies Assumption 4) multiplying the states (depending on the controls of other players).
>
> **3. Theorem 3.3 (Lower bound), [updated after a previously wrong first rebuttal stating that the reduction to existing lower bounds does not hold, apologies for the update after initial rebuttal post, the rest of the rebuttal is mainly unchanged]**
>
> > The authors note that Theorem 3.3 does not follow from existing online convex regret lower bounds, due to policy regret differences. But consider any LDS with (A,B) = (0,0), this reduces to an online convex game over controls, and now standard T^0.5 lower bounds apply. So, I don't see the barrier here.
>
> We thank the reviewer for pointing out the connection to standard online learning with $A = B = 0$. We agree that a $\Omega(\sqrt{T})$ lower bound on policy regret with respect to $\Pi_i^{\mathrm{lin}}$
> can follow from a reduction to classical online convex optimization lower bounds: when $A = B = 0$ in dimension 1, every linear controller $K \in \mathbb{R}$ is strongly stable for some $(\kappa,\gamma)$ (as the closed-loop matrix is $A - BK = 0$ for every gain $K$), so the comparator class coincides with $\mathbb{R}$, and a reduction yields the usual $\Omega(\sqrt{T})$ rate. One needs to be slightly careful in the reduction as the set $\Pi_i^{\mathrm{lin}}$ is unbounded but in the hard instance one can choose the sequence of cost functions so that the best fixed comparator (the minimizer over $\mathbb{R}$) lies in a bounded interval, and then the usual $\Omega(\sqrt{T})$ regret lower bounds for online convex optimization over that interval immediately translate to policy-regret lower bounds with respect to $\Pi_i^{\mathrm{lin}}$. Note though that this argument will not work for $w_t = 0$ (see details below) but works for $w_t =1$.
>
> Our Theorem 3.3 complements this observation by showing that an $\Omega(\sqrt{T})$ lower bound also holds for **nontrivial** LDS, where $B \neq 0$ and the states are not constant (see our nontrivial hard instance). We will make this clearer in the revision, correct
> the misleading statement in l. 340-342 and update Theorem 3.3 to reflect the nontriviality of the hard instance LDS.
> This setting is more relevant for our upper bounds and our control setting. This shows that the $\sqrt{T}$ upper bounds we prove cannot, in general, be improved even when one exploits the structure of the dynamics, rather than only the degenerate case where the system is static.
>
> **Details on algorithm with zero policy-regret in noiseless LDS setting with A=B=0, w_t=0**:
> Consider the one-dimensional LDS setting with $A=B=0$ and no disturbances ($w_t =0$ for all $t$), and for $N=1$. Then we can show the existence of a control algorithm such that, on any cost sequence, the policy regret against strongly-stable linear controllers is always 0, which precludes an external regret lower bound from online learning from being applied in this setting.
>
> In this setting, $x_{t+1} = Ax_t +  Bu_t + w_t = 0$ holds no matter the control sequence $\{u_t\}$ and a linear control policy yields $u^K_t = K x^K_t = 0$ since $x^K_t = 0$. Under this LDS instance, every strongly-stabilizing linear controller plays the control $u=0$ at every time step. This means that, for any $K$, the cumulative cost under this linear policy is $\sum_{t=1}^T c_t(x^K_t, u^K_t) = \sum_{t=1}^T c_t(0, 0)$ and hence the policy-regret against strongly-stable linear controllers for an algorithm $\mathcal{A}$ producing control sequence $\{u_t\}$ simplifies to:  $\text{reg}^T(\mathcal{A}, \Pi^{\text{lin}}) = \sum_{t=1}^T c_t(0, u_t) -  c_t(0, 0)$.
>
> Thus for any sequence of cost functions, the setting $A=B=0$ causes the cumulative cost of every linear policy to collapse to the same value. This means that the constant control algorithm that plays the control $u_t = 0$ at every round $t$, for every sequence of cost functions, always has 0 regret. Thus for the noiseless LDS instance with $A=B=0$, lower bound results for online learning do not directly apply in this case.

---

> ### Author Response · Authors · 2025-11-20
> **Rebuttal end**
>
> **4. Section 4 and Theorem 4.1 (Equilibrium gap guarantee)**
>
> > In contrast to the above, I really like the set of questions in Section 4. However, Theorem 4.1 provides a non-varnishing bound, thus there is no implication of convergence to an equilibrium-like notion.
>
> The bound in Theorem 4.1 depends on the variability of cost functions and disturbances. As these are time-varying games, no convergence is expected without further assumptions. After the theorem, we provide examples in which the bound is vanishing and convergence to equilibria occurs. We provide more explanations below:
>
> In section 4, we show the connection between our setting and time-varying games. In a time-varying game, payoffs are time-dependent. Therefore, equilibria are time-dependent without further assumptions, and there is no expected convergence without additional assumptions (e.g. the time-dependent payoffs converge to time-independent values). Note that time dependence here is exogenous, as it is not solely induced by the fact that other players are also updating their strategies like in standard online learning in games. In time-varying games, a natural quantity to monitor is the time-dependent equilibrium gap. Theorem 4.1 shows that the (time) average equilibrium gap is bounded by the variation of both the cost functions and disturbances. In lines 453-461, we briefly discuss conditions on the cost functions and disturbance sequences (which are part of the problem description) under which convergence to equilibria does happen. Sublinear bounds on the variability of costs and disturbances immediately translate into vanishing average equilibrium tracking gaps, and hence convergence to a policy with small equilibrium gap (e.g. for the best iterate).

---

### Author Response · Authors · 2025-12-04
**Final Author Comments**

We thank all the reviewers for their time and valuable feedback.

- We are encouraged that all reviewers highlight the relevance of our online multi-agent control setting and the importance of the questions we study. Reviewer nX9c finds that “the paper puts forward an interesting setup, and it is an interesting question as to what equilibria arise…” Reviewer tTR7 notes that “the paper tackles a challenging and relevant problem at the intersection of online non-stochastic control and online learning in games” and “provides a strong theoretical analysis.” Reviewer JSuY comments that the work “extends [Agarwal 2019] in an interesting direction” and that “analyzing regret / convergence / stability for multi-agent systems can always be challenging,” adding that we “identify nontrivial multi-agent setups in which learning can happen.” Reviewer jhEz similarly highlights that “the multi-agent setting is both natural and valuable—it extends the single-agent adversarial control framework to a broader and practically relevant domain.”

Our results establish both individual and collective performance guarantees in a dynamic, game-theoretic multi-agent control setting. We summarize each contribution in light of the reviewers’ comments and our rebuttal:

- **Theorem 3.2 (individual regret in decentralized learning)**: This theorem gives the first individual regret bound in the independent-learning setting via a reduction to single-agent online control using DAC policies with modified disturbances. As noted by the reviewers, this result quantifies the price of decentralization through its dependence on the number of agents. This contribution indeed uses a careful application of single-agent online control results from the technical viewpoint.

- **Theorem 3.3 (lower bound):** Reviewer tTR7 notes that this “confirms the optimality of the bounds with respect to $T$.” Reviewer nX9c observes that one can reduce to a trivial static system $A=B=0$ to apply known online-learning lower bounds. We clarified in the rebuttal how this reduction is possible while emphasizing that our proof shows the lower bound also holds for nontrivial LDS, where $B \neq 0$ and states evolve non-trivially. This complements the trivial reduction and establishes optimality more robustly.

- **Theorem 3.4 (individual regret with richer information):** Reviewer tTR7 notes that considering two information structures “provides a clear understanding of the value of information and the price of decentralization.” This theorem demonstrates an improved dependence on the number of agents and requires new analysis of the joint state evolution under DAC policies. Reviewer JSuY acknowledged the technical challenges after the rebuttal: “I agree that the proofs can be very involved, and one need to close certain gaps upon the analysis in [Agarwal et al.] which was already quite complicated.”

- **Theorem 4.1 (equilibrium tracking in common-interest games):** Reviewer nX9c “really likes the set of questions in Section 4,” and reviewer tTR7 finds the result “a valuable contribution.” As expanded in the rebuttal, this theorem provides a novel equilibrium-tracking guarantee in a dynamic (stateful) common-interest setting.

- **Simulations:** In response to reviewer jhEz, we included initial simulations in Section K of the revised appendix (pp. 41–43). The reviewer found these “quite interesting” and noted that they would strengthen the paper further if included in the main text. They subsequently increased their score to 8.

Finally, we reiterate the significance of our contributions:
- **First theoretical treatment:** To our knowledge, this is the first work analyzing online multi-agent control with decentralized strategic agents, individual time-varying objectives, and adversarial disturbances.
- **Bridging online control and learning in games:** Our paper connects single-agent online control with multi-agent learning in dynamic games, an underexplored intersection. Section 4 establishes equilibrium-tracking guarantees for continuous, time-varying games with state dynamics and adversarial disturbances, going beyond prior work.
- **Opening new directions:** Our framework suggests several promising extensions, including unknown or time-varying dynamics and partial-feedback settings.

---

### Meta-Review · Area_Chair_54zx · 2025-12-26

**Summary:**

The paper addresses an important and timely problem of online control in multi-agent linear dynamical systems under adversarial disturbances. The reviewers acknowledged the relevance of the setting and the technical analysis. However,  the consensus remains that the submission does not yet meet the bar for acceptance, primarily due to concerns about incremental novelty, reliance on strong assumptions, and limited empirical grounding. Although some reviewers raised the score, but the key concern of the technical novelty remains even after the rebuttal.

**Reviewer Concerns:**

A central concern is the limited technical novelty, where some results in this paper use existing results in a black-box manner. The results of regret bounds and their dependence on the number of agents are expected extensions, rather than conceptual or methodological breakthroughs.

**Reviewer Scores:**

Reviewer nX9c
Original score: 4
Likely change: No change
This reviewer’s primary concern was the limited novelty, which remains unchanged after the rebuttal.

Reviewer tTR7
Original score: 6
Likely change: Slight decrease to 5 or remain at 6
The reviewer raised substantive concerns about practicality (global stability, centralized computation, boundedness assumptions). The rebuttal addressed these points thoughtfully but did not relax the assumptions.

Reviewer JSuY
Original score: 6 (marginally above acceptance threshold)
Likely change: No change (remain at 6)

This reviewer explicitly stated after the rebuttal that they would keep their score.

Reviewer jhEz
Original score: 6
Likely change: 8
Rationale:
This reviewer was initially concerned about lack of empirical validation and incremental novelty. After the authors added simulations, the reviewer explicitly raised their score.

---

### Decision · Program_Chairs · 2026-01-26

Reject